# Out-of-distribution generalization of deep-learning surrogates for 2D PDE-generated dynamics in the small-data regime

## Abstract

Partial differential equations (PDEs) are a central tool for modeling the dynamics of physical, engineering, and materials systems, but high-fidelity simulations are often computationally expensive. At the same time, many scientific applications can be viewed as the evolution of spatially distributed fields, making data-driven forecasting of such fields a core task in scientific machine learning. In this work we study autoregressive deep-learning surrogates for two-dimensional PDE dynamics on periodic domains, focusing on generalization to out-of-distribution initial conditions within a fixed PDE and parameter regime and on strict small-data settings with at most $\mathcal{O}(10^2)$ simulated trajectories per system. We introduce a multi-channel U-Net with enforced periodic padding (me-UNet) that takes short sequences of past solution fields of a single representative scalar variable as input and predicts the next time increment. We evaluate me-UNet on five qualitatively different PDE families — linear advection, diffusion, continuum dislocation dynamics, Kolmogorov flow, and Gray–Scott reaction–diffusion—and compare it to ViT, AFNO, PDE- Transformer, and KAN-UNet under a common training setup. Across all datasets, me- UNet matches or outperforms these more complex architectures in terms of field- space error, spectral similarity, and physics-based metrics for in-distribution rollouts, while requiring substantially less training time. It also generalizes qualitatively to unseen initial conditions and, e.g., reaches comparable performance on continuum dislocation dynamics with as few as $\approx 20$ training simulations. A data-efficiency study and Grad-CAM analysis further suggest that, in small-data periodic 2D PDE settings, convolutional architectures with inductive biases aligned to locality and periodic boundary conditions remain strong contenders for accurate and moderately out-of-distribution-robust surrogate modeling.

## 1 Introduction

Scientific and engineering practice increasingly relies on numerical models and data-driven methods to understand, predict, and control complex dynamical systems. Many such systems can be viewed as the evolution of spatially distributed fields—temperatures, concentrations of chemical species, velocity and pressure fields in fluids, elastic displacement and stress in solids, electromagnetic fields, order parameters in phase-field models, probability densities in population dynamics, or quantum-mechanical wave functions—over time and space. A central goal of scientific machine learning (SCIML) (Dietrich & Schilders, 2025) is therefore to learn such field dynamics from data, either in a purely data-driven fashion or by additionally exploiting assumed or derived governing equations, for example by integrating scientific knowledge with machine learning as discussed in recent surveys such as (Willard et al., 2023).

The behavior of any real systems is extremely complex with a large number of unknown parameters and relations. Therefore, in practice, idealized mathematical models expressed as systems of equations are used as a substitute. For a large class of phenomena, these equations take the form of partial differential equations (PDEs), which encode, e.g., local conservation laws, balance relations, and constitutive assumptions. PDE models underpin applications from general relativity and quantum mechanics to fluid dynamics, elasticity,

reaction–diffusion, and continuum descriptions in materials science. Examples of famous PDEs include Einstein's equations of general relativity (Einstein, 1915), which describe how the curvature of spacetime responds to the presence of matter and energy, the Schrödinger equation (Schrödinger, 1926) for the evolution of quantum-mechanical wave functions, and the Navier–Stokes equations (Navier, 1821) governing the motion of viscous fluids. Their flexibility and reach make PDEs some of the most widely used tools for representing spatio-temporal field evolution in science and engineering, even though they always remain idealized approximations of reality. The particular choice of PDEs determines which classes of phenomena can be represented, and the broad variety of PDEs models reflects the breadth of field dynamics encountered in practice.

When high-fidelity PDE models are available, they play a dual role in SciML. First, they are scientifically important in their own right: many PDEs of interest are challenging to solve numerically, and accurate simulations can require substantial computational resources. Classical examples include large-scale cosmological simulations such as *Illustris* (Vogelsberger et al., 2014), which required more than 8000 CPU cores for several months of wall-clock time, or state-of-the-art climate simulations that run for weeks to months on leadership-class supercomputers with correspondingly large energy consumption (Randall et al., 2019). Second, PDE solvers provide a controlled and reproducible way to generate training data with known ground truth. By sampling initial and boundary conditions, various parameters, and forcing terms, one can construct benchmark datasets that probe a range of dynamical regimes while avoiding experimental noise, measurement artifacts, and uncertainty about the underlying equations. In this sense, PDE-based benchmarks are an attractive stepping stone towards the much harder problem of learning from real measurement data, where the governing equations may be only partially known or not explicitly available. Benchmark suites such as PDEBench (Takamoto et al., 2022) already provide standardized datasets and baselines for a broad range of time-dependent PDEs, primarily targeting large-data settings and broad parametric coverage. Here we instead emphasize strict small-data regimes, periodic 2D dynamics, and physics-based evaluation, aiming to mimic the constraints of many real scientific applications where generating each high-fidelity simulation is expensive.

From this perspective, PDEs in our work are not an end in themselves but a convenient and stringent testbed for models that aim to learn complex field dynamics. A SciML method that fails to robustly learn dynamics generated by well-posed PDEs—in a setting where the discretization, parameters, and numerical errors are under control—is unlikely to succeed on heterogeneous, noisy, and partially observed real-world data. Conversely, architectures that generalize well across PDE-generated datasets, in particular under limited-data and out-of-distribution (OOD) conditions, are promising candidates for deployment on experimental or observational datasets. Thus, our choice to study 2D PDE dynamics is motivated both by the intrinsic importance of PDEs in science and engineering and by their role as a controlled proxy for more complex real-world field evolution.

Deep learning (DL) surrogates fit naturally into this program. Instead of solving the governing equations on the fly for each new query, one trains a parametric model to approximate the time-advance operator that maps past field configurations (and, where applicable, control parameters) to their future evolution. Once trained, such surrogates can produce approximate multi-step forecasts at drastically reduced inference cost compared to conventional solvers, enabling tasks such as accelerated parameter studies, uncertainty quantification, or real-time control. We refer to a sequence of such predicted future fields, obtained by iteratively applying the learned time-advance operator starting from a ground-truth context, as an *autoregressive rollout*.

However, real scientific applications typically operate in *small-data regimes*, where only tens to hundreds of high-fidelity simulations or experiments are available, and where test conditions may differ markedly from those seen during training. In this regime, the central challenge is not merely to fit a training distribution, but to achieve qualitatively robust, moderately OOD generalization of field dynamics while keeping training costs within the reach of typical scientific users.

A large body of work has explored neural networks as surrogates for differential equations. Early approaches in the 1990s already exploited universal approximation results for feed-forward networks (Hornik et al., 1989) to represent solutions of ordinary and partial differential equations in collocation schemes, e.g. as in (Lee & Kang, 1990; Dissanayake & Phan-Thien, 1994; Lagaris et al., 1998). More recently, several

influential lines of research have emerged. Neural operators such as Fourier Neural Operator (FNO) (Li et al., 2020) and Deep Operator Networks (DEEPONETs) (Lu et al., 2021a) aim to learn resolution- and mesh-agnostic mappings between function spaces. They have demonstrated impressive interpolation performance on benchmarks such as parametric Navier–Stokes flows, often using on the order of $10^3$–$10^4$ simulated trajectories, each of which consists of a large number of time steps. Physics-Informed Neural Networks (PINNs) (Raissi et al., 2019; Zhu et al., 2019; Shukla et al., 2020; Zhang et al., 2020; Ren et al., 2022; 2023; Yuan et al., 2024) take a complementary route by embedding the governing equations and boundary conditions into the loss function via automatic differentiation, which allows them to exploit sparse data but typically requires solving a challenging PDE-constrained optimization problem. Related approaches have also explored neural representations of operators and data-driven discovery of PDEs, including wavelet-based architectures, meta-learning of pseudo-differential operators, and hybrid numeric-symbolic frameworks (Fan et al., 2019; Feliu-Faba et al., 2020; Khoo et al., 2021; Long et al., 2018; 2019). Most recently, foundation-style PDE models such as PROSE-PDE (Sun et al., 2025) and PDE-Transformer (Holzschuh et al., 2025) seek broad generalization across systems and geometries through large-scale multi-task pretraining on billions of tokens of simulated data.

Despite these advances, comparatively little work systematically investigates how relatively simple, carefully designed convolutional architectures perform as time-stepping surrogates for 2D field dynamics under strict small-data constraints and OOD initial conditions. Most existing benchmarks either assume access to large numbers of simulated trajectories, focus on in-distribution error metrics, or emphasize equation-informed and foundation-scale models whose training requirements are far beyond the resources of many domain scientists. Moreover, PDEs are typically treated as the ultimate target of modeling, rather than as a controlled environment in which to stress-test generalization in preparation for real measurement data.

In this work, we adopt the latter viewpoint and use PDE-generated field dynamics as a controlled testbed for studying OOD generalization of DL surrogates in realistic small-data regimes. Concretely, we consider two-dimensional, periodic field dynamics generated by five qualitatively different PDE families that cover transport, diffusion, pattern formation, and microstructure evolution. We then ask how different architectures behave when trained on at most a few dozen to one hundred simulations per PDE, and evaluated both on held-out simulations from the same distribution and on initial conditions that differ substantially from those seen during training. Our focus is on autoregressive surrogates that operate as time-stepping models, taking short sequences of past solution fields as input and predicting the next temporal increment, and on how architectural inductive biases (such as convolutional locality and periodic padding) influence data efficiency and out-of-distribution behavior.

Our contributions are as follows:

- **A small-data benchmark for 2D periodic field dynamics.** We construct ten datasets derived from five representative PDE families—linear advection, diffusion, continuum dislocation dynamics, Kolmogorov flow, and Gray–Scott reaction–diffusion—and use them as a controlled testbed for autoregressive forecasting on periodic $64 \times 64$ grids. For each system we define a strict small-data regime (at most 100 simulations) and evaluate both in-distribution rollouts and generalization to qualitatively different initial conditions within the same PDE and parameter regime.

- **A simple convolutional baseline with enforced periodicity.** We propose a multi-channel U-Net architecture with periodic padding (me-UNet) that operates as an incremental time-stepping surrogate: it takes a short sequence of past solution fields as input and predicts the next temporal increment. Under a common training protocol and fixed data budget, me-UNet consistently matches or outperforms several more complex neural-operator and transformer-based architectures (ViT, AFNO, PDE-Transformer, and KAN-UNet) in terms of field-space error and spectral similarity, while requiring substantially less training time.

- **Physics-aware evaluation of long autoregressive rollouts.** We adopt an evaluation protocol that complements standard pixel-wise and spectral metrics with physics-aware diagnostics, tracking conserved or prescribed global quantities such as mass, energy, and total dislocation density over 100-step rollouts. Across PDE families, me-UNet preserves these quantities more accurately than the competing architectures, including for moderately out-of-distribution initial conditions.

- **Data-efficiency and interpretability insights.** We perform a data-efficiency study that varies the number of simulations, time steps, and input sequence length, and we use Grad-CAM to analyze which spatial regions and U-Net blocks contribute most strongly to the predictions. These analyses indicate that convolutional inductive biases aligned with spatial locality and periodicity allow me-UNet to reach low-error regimes with as few as $\approx 20$ training simulations in continuum dislocation dynamics and to allocate representational capacity in a physically meaningful way.

Taken together, our results highlight that, in small-data, periodic 2D field-dynamics settings, carefully designed convolutional baselines remain strong contenders for accurate and moderately out-of-distribution-robust surrogate modeling. More broadly, our study underscores the importance of evaluating SciML architectures under realistic data constraints and physics-aware metrics, and supports the use of PDE-generated benchmarks as a rigorous stepping stone toward models that can robustly learn from real experimental and observational data.

The remainder of this paper is organized as follows. Sections 2.1 and 2.2 introduce the PDE models and simulation datasets, and Section 2.4 describes the training setup. Section 2.3 describes the me-UNet and baseline architectures. Section 3 presents our empirical results, and Section 4 discusses implications for scientific machine learning and future work on real measurement data.

## 2 Mathematical model and neural network architecture

### 2.1 Mathematical model

In this work we consider six mathematical models of increasing complexity, listed in Tab. 1. They differ not only in the type of dynamics (transport, diffusion, pattern formation, turbulence, microstructure evolution) but also in the number and coupling of state variables. We organize them into three groups:

- **PDE-1–2: single-field linear transport and diffusion.** These are classical scalar advection and diffusion equations on periodic domains, used as basic test cases for forecasting and spectral fidelity.

- **PDE-3–4: continuum dislocation dynamics (CDD)-based models.** These models describe the transport and evolution of possibly curved line-segments via various continuum densities. PDE-3a and PDE-3b are reduced continuum dislocation dynamics (CDD) systems that capture expansion and motion of dislocation loops in a simplified setting (with a smaller number of active fields and, in PDE-3a and 3b, curvature initialized to zero), whereas PDE-4 is the full three-field CDD model considered in Appendix 2.1 with an additional curvature density.

- **PDE-5–6: multi-field fluid and reaction–diffusion systems.** PDE-5 is a Navier–Stokes system in Kolmogorov-flow configuration, evaluated in terms of the scalar vorticity field, and PDE-6 is the Gray–Scott reaction–diffusion system with two reacting species.

Example simulations from all six models are shown in Fig. 1. Together they cover a range from simple scalar transport and diffusion to more complex systems such as Kolmogorov flow, Gray–Scott pattern formation, and the statistical evolution of systems of curved lines in CDD. Detailed descriptions of each model and the chosen parameters are provided in the Appendix.

To keep comparisons uniform and to mimic partial observability in real measurements, we train all models on rollouts of a single representative scalar field per system (Table 1), even when the underlying PDE couples multiple state variables.

### 2.2 Simulation datasets and training samples

The dataset abbreviations are listed in Tab. 1: for PDE-1, 2, 4, 5 we obtain datasets DS-1, 2, 4 and 5; for PDE-3 we consider 2 parameter settings, denoted DS-3a and DS-3b; for PDE-6 we consider four different parameter settings, denoted DS-6a, DS-6b, DS-6c, and DS-6d. Each of the resulting 10 datasets consists

Table 1: Overview of the six PDE models and resulting datasets used in this work. The column "Fields" refers to the number and type of continuous fields used for the numerical solution ("simulation"). The training dataset uses only *a single field* of that simulation as input to the ML model, cf. section 2.3.

| No. | Mathematical model | Simulation | | Training | |
|---|---|---|---|---|---|
| | | Equation | Fields | Datasets | Features |
| 1 | Advection of a distribution of blobs | PDE-1 | 1 (scalar) | DS-1 | 1 (concentration) |
| 2 | Diffusion of a distribution of blobs | PDE-2 | 1 (scalar) | DS-2 | 1 (concentration) |
| 3 | Expansion and diffusion of a distribution of loops (reduced CDD) | PDE-3 | 3 (1 scalar, 1 vector) | DS-3a, DS-3b | 1 (total dislocation density) |
| 5 | Statistical evolution of systems of curved lines (CDD) | PDE-4 | 4 (2 scalars, 1 vector) | DS-4 | 1 (total dislocation density) |
| 6 | Kolmogorov flow (Navier–Stokes, vorticity formulation) | PDE-5 | 2 (1 vector) | DS-5 | 1 (vorticity) |
| 7 | Reaction–diffusion (Gray–Scott model) | PDE-6 | 2 (scalar) | DS-6a, DS-6b, DS-6c, DS-6d | 1 (concentration) |

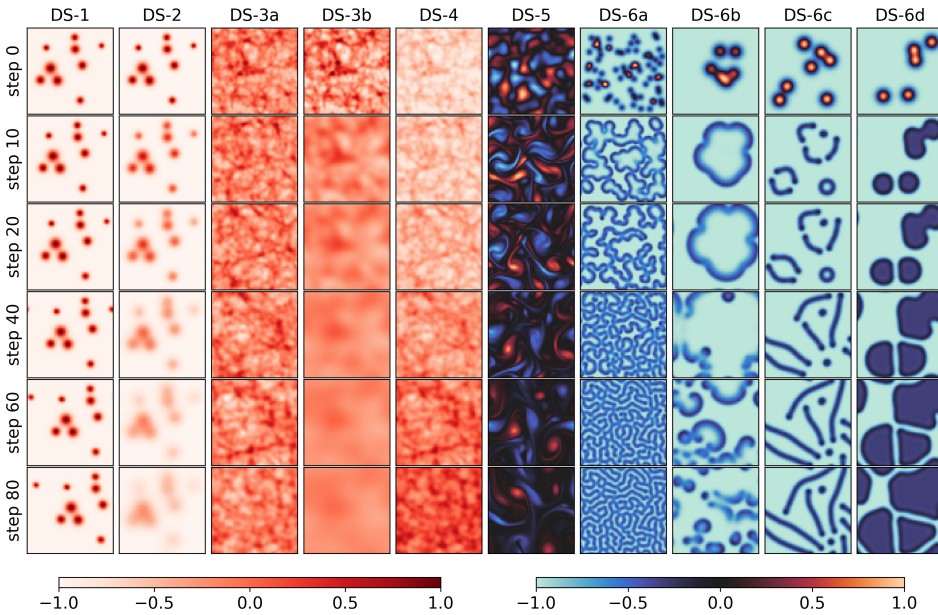

Figure 1: Examples of simulation results from all mathematical models. Datasets DS-1 and 2 are obtained from simple convection and diffusion equations, while the complexity of the PDEs and the number of involved state variables increases towards the right: DS-3a, 3b, and 4 are obtained from CDD PDEs, DS-5 from Navier-Stokes equations, while the variants of DS-6 are obtained from the Gray-Scott model.

of 110 simulations. The PDEs are simulated on higher-resolution grids to ensure numerical accuracy and stability of the reference solutions. These fields are then downsampled to 64×64 before being used for training and evaluation. The downsampling is performed using standard interpolation, which preserves the large-scale structures and is consistent with the effective resolution of the model. Examples from each dataset are shown in Fig. 1, and detailed descriptions are given in the Appendix. For each dataset, 100 simulations are used for training, and the remaining 10 are kept untouched for validation and testing. Each simulation contains 100 time steps.

For autoregressive training, the input data are constructed by sampling a random time index $n$ with $7 \leq n \leq 99$ (using one-based indexing for time steps). For each such $n$, we stack the 7 consecutive fields

$$u^{n-6}, u^{n-5}, \ldots, u^n \tag{1}$$

along the channel dimension to form a multi-channel input image. The corresponding target is an *increment field* that represents the change between time steps $n$ and $n+1$,

$$\Delta u^{n+1} = u^{n+1} - u^n. \tag{2}$$

Thus, our models are trained to predict $\Delta u^{n+1}$ given the past sequence $\{u^{n-6}, \dots, u^n\}$, rather than the absolute field $u^{n+1}$. During rollout, the next state is obtained via the residual update

$$\hat{u}^{n+1} = \hat{u}^n + \Delta \hat{u}^{n+1}, \tag{3}$$

where hats denote model predictions. Focusing on local temporal increments stabilizes long autoregressive rollouts: the network only needs to learn short-term corrections instead of reconstructing the full state, and it can concentrate on the dynamics of the system rather than on absolute field values. This residual parameterization empirically leads to substantially more stable long-horizon predictions, which is consistent with the fact that many PDE solutions change gradually between adjacent time steps.

All data values are scaled to the interval $[-1, 1]$ separately for each simulation and state variable. For a given state variable, the minimum and maximum over all time steps within a simulation are used to linearly map the values in each frame to the required range; scaling is not recomputed per frame. This per-simulation, per-variable normalization follows common practice in PDE surrogate modeling, but it can have drawbacks in multi-field systems, which we discuss later in Appendix E. All images are resized to $64 \times 64$ pixels.

**In-distribution vs out-of-distribution initial conditions**  For each PDE family we fix the governing equations, physical parameters, numerical scheme, and spatial resolution. Training and test simulations always share this fixed configuration. The only source of distribution shift we study in this work is the distribution of initial conditions: "in-distribution" (ID) test rollouts use the same random initial-condition generator as training, whereas "out-of-distribution" (OOD) test rollouts use qualitatively different initial-condition families within the same PDE and parameter regime (e.g., sparse loops or line-like structures for CDD, or structured line/blob perturbations for Gray–Scott). Thus, throughout this paper, OOD refers specifically to shifts in the initial-condition distribution under a fixed PDE and parameter setting.

## 2.3  Neural network architecture

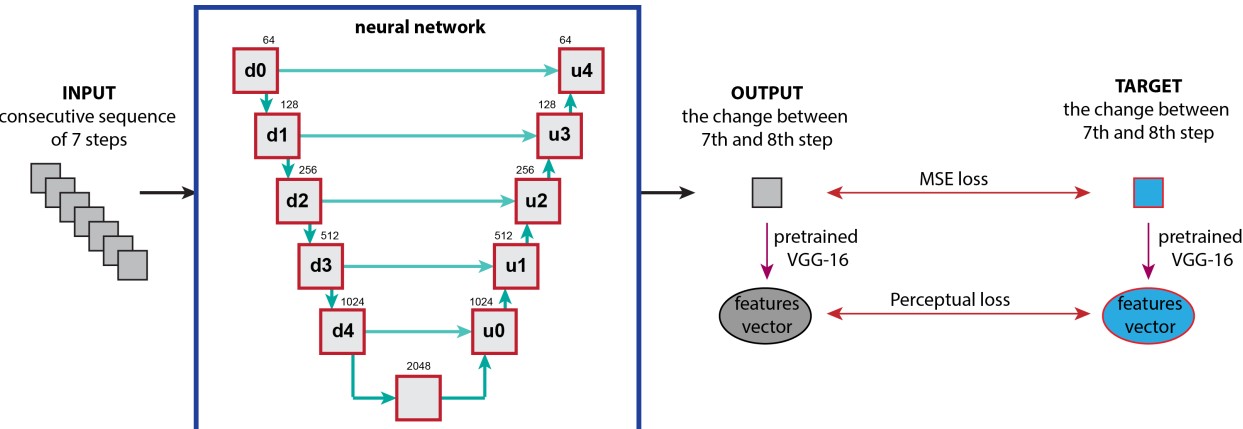

Figure 2: Illustration of the me-UNet architecture, showing the encoder–decoder structure, periodic padding, and multi-channel input/output setup for autoregressive time-stepping.

Our U-Net architecture is an extension of the original architecture by Ronneberger et al. (Ronneberger et al., 2015) and consists of two parts: an encoder (contracting path) that captures the spatial context of the input fields, and a decoder (expanding path) that enables precise localization of features. The input

consists of a sequence of images, i.e., even though some of the original PDEs consist of multiple or vectorial field quantities, we only use a single, scalar field. The encoder comprises five stages, each with a double convolutional module followed by an average-pooling operator for downsampling. The decoder also contains five stages, each with a transposed convolution operator for upsampling followed by a double convolutional module. Between encoder and decoder we place a double convolutional module acting as a bottleneck.

Each double convolutional module consists of two convolutional layers, each followed by batch normalization and a nonlinear activation function. A final convolutional layer is added at the end of the network to reduce the number of channels to match the number of output state variables of the considered problem. In total, the network has 23 convolutional layers, five average-pooling operators for downsampling, and five transposed convolution operators for upsampling. Skip connections between encoder and decoder stages allow low-level features from earlier layers to be combined with higher-level features in the decoder. This feature fusion helps preserve spatial detail and improves accuracy in tasks that require precise localization.

The first substantial modification in our approach is the use of *periodic padding*. In order to enforce periodic boundary conditions during training, we apply periodic padding in all convolutional layers whenever the data have periodic boundary conditions (Qu et al., 2022; Rao et al., 2023). In this way, prior knowledge about the boundary behavior is encoded directly into the network architecture so that the trained model cannot ignore periodicity. Concretely, instead of using the default zero padding in the `Conv2d` layers, we pad by copying values from the opposite side of the image: the leftmost columns are copied to the right, the rightmost columns to the left, and analogously for the top and bottom rows, yielding a fully periodic domain. This can also be implemented using the PyTorch built-in `CircularPad2d` operator.

We also adjust the downsampling operator by using average pooling with stride 2 instead of max pooling. According to (Zhao & Zhang, 2024), average pooling is a spatially invariant operation whose output does not depend on the precise position of features within the pooling window, whereas max pooling can neglect weaker but still important features. In our experiments, average pooling leads to more stable rollouts and better preservation of fine-scale structures.

A second design aspect is the ability to adjust the number of input and output channels. In general, the architecture can handle multiple state variables and multiple time steps as separate channels. In our setup we use an input sequence of $L = 7$ past time steps of a single representative state variable per PDE, so the input has seven channels. For systems with multiple state variables (e.g., the CDD model involves four fields), the architecture could in principle output several channels; however, for the cross-dataset comparisons in this work we restrict training and evaluation to one state variable per PDE to keep the setup consistent across datasets. The choice of sequence length $L$ is a hyperparameter that controls how much temporal context the model sees relative to the intrinsic time scale of the dynamics. For the datasets considered here, the simulation time steps were chosen such that the evolution over six consecutive intervals exhibits nontrivial but still smooth changes; for systems with slower or faster dynamics, one would typically adapt $L$ or sub-sample time steps accordingly.

We denote by $f_\theta$ the parametric mapping implemented by a given architecture, with trainable parameters $\theta$. All architectures considered in this paper, including me-UNet and the baselines, are trained in an *incremental* or residual form. Given an input sequence of $L$ past fields

$$u^{n-L+1}, \ldots, u^n \in \mathbb{R}^{H \times W}, \tag{4}$$

the network does not predict the next field $u^{n+1}$ directly. Instead, it predicts a temporal update

$$\Delta u^{n+1} = f_\theta\big(u^{n-L+1}, \ldots, u^n\big), \tag{5}$$

and the next state is obtained as

$$u^{n+1} = u^n + \Delta u^{n+1}. \tag{6}$$

Thus, for $L = 7$ the output image represents the change in the field between the 7th and 8th time step. This residual-style parameterization empirically stabilizes long autoregressive rollouts and encourages the network to focus on learning short-time dynamics rather than absolute field values.

To enable a fair comparison with other architectures, we keep the input sequence length (seven consecutive time steps), the incremental output representation in equation 6, and the loss function fixed across models. Only the internal architecture of the mapping $f_\theta$ is changed. Specifically, in addition to me-UNet we adapt four alternative architectures: a vision transformer (ViT) (Dosovitskiy et al., 2020), an adaptive Fourier neural operator (AFNO), which is the core architecture of FourCastNet (Pathak et al., 2022), the PDE-Transformer (PDE-T) (Holzschuh et al., 2025), and a KAN-UNet in which the double convolutional layers of U-Net are replaced by KAN convolutional layers (Gao, 2024).

During training we minimize a combination of a pixel-wise loss $\mathcal{L}_{\mathrm{MSE}}$ and a perceptual loss $\mathcal{L}_{\mathrm{pc}}$. The total loss $\mathcal{L}$ for a predicted field $\hat{y}$ and reference field $y$ is

$$\mathcal{L}(\hat{y}, y) = \mathcal{L}_{\mathrm{MSE}}(\hat{y}, y) + \lambda_{\mathrm{pc}}\, \mathcal{L}_{\mathrm{pc}}(\hat{y}, y), \tag{7}$$

where $\mathcal{L}_{\mathrm{MSE}}$ is the mean-squared error defined in eq. (10), and $\mathcal{L}_{\mathrm{pc}}$ is the perceptual loss based on feature maps $\Phi_i$ of a pre-trained VGG-16 network as in eq. (11). In all experiments we keep the VGG-16 weights fixed and use feature maps from layer `relu2_2`, and we set $\lambda_{\mathrm{pc}} = 1$. We provide an ablation isolating the effect of $\mathcal{L}_{\mathrm{pc}}$ in Appendix J; in brief, the perceptual term is essentially neutral on simple datasets but substantially improves the preservation of extended geometric structures (e.g., the Gray–Scott worm regime, DS-6c) over long rollouts.

## 2.4 Training setup

For each PDE dataset and each architecture, we train a separate model using the same training protocol. Unless otherwise stated, we use the Adam optimizer with an initial learning rate of $1 \times 10^{-4}$, weight decay $1 \times 10^{-5}$, and a batch size of 40. Each model is trained for the same fixed budget of 1000 epochs without patience-based early stopping, and the final model used for evaluation is the checkpoint with the lowest validation loss within that budget. We use this protocol — rather than architecture-dependent early stopping — so that every architecture sees an identical training budget, which we view as the most honest setup for a comparative study of architectures.

For me-UNet, the channel widths at the five encoder levels are $[64, 128, 256, 512, 1024]$ and mirrored in the decoder. For the baselines, we follow standard configurations adapted to our $64 \times 64$ input size: for ViT we use a patch size of 8, embedding dimension 768, depth 12, and 8 attention heads; for AFNO we use a patch size of 4, embedding dimension 768, depth 12, MLP ratio 4, and 16 AFNO blocks; for PDE-Transformer we use the PDE-B configuration of (Holzschuh et al., 2025); and for KAN- UNet we mirror the me-UNet channel widths in encoder and decoder, replacing the `Conv2d` layers in the double-convolution blocks by `FastKANConvLayer` modules with Chebyshev basis functions. All other architectural details are kept as close as possible to the original implementations referenced above.

Periodic boundary conditions are implemented in most of the mentioned architectures: in AFNO, periodic padding is used through an additional 2D convolutional layer (Pathak et al., 2022); PDE-Transformer mimics periodic boundary conditions by rolling the tokens along the x- and y-axis when shifting the attention windows (Holzschuh et al., 2025); KAN-UNet implements periodic padding similar to me-UNet. Only our implementation of the vision transformer use the vanilla architecture.

## 2.5 Evaluation metrics and physics-aware measures

We evaluate all models using three complementary criteria: (i) field-space error, (ii) spectral similarity, and (iii) physics-aware metrics that monitor conserved or prescribed global quantities.

**Field-space error** For each rollout we compute the root mean square error (RMSE) between the predicted fields $\hat{y}$ and reference fields $y$, averaged over spatial grid points, output channels, and time steps:

$$\mathrm{RMSE} = \sqrt{\frac{1}{NTC} \sum_{t=1}^{T} \sum_{c=1}^{C} \sum_{i=1}^{N} \left(\hat{y}_{t,c,i} - y_{t,c,i}\right)^2}, \tag{8}$$

where $N$ is the number of spatial grid points, $C$ the number of state variables, and $T$ the number of time steps in the rollout.

**Spectral similarity**  To compare spatial frequency content we compute the azimuthally averaged power spectral density (PSD) for each snapshot and channel similar to what is done in (Koehler et al., 2024; Nguyen et al., 2024). Let $P_{t,c}(k)$ and $\hat{P}_{t,c}(k)$ denote the PSD of the reference and predicted fields, respectively, as a function of the radial wavenumber $k$ (obtained by averaging $|\mathcal{F}[y_{t,c}]|^2$ and $|\mathcal{F}[\hat{y}_{t,c}]|^2$ over angular directions in Fourier space). For each time step $t$ and channel $c$ we then compute the cosine similarity

$$\text{cos\_sim}_{t,c} = \frac{\sum_k \hat{P}_{t,c}(k)\, P_{t,c}(k)}{\sqrt{\sum_k \hat{P}_{t,c}(k)^2}\, \sqrt{\sum_k P_{t,c}(k)^2}}. \tag{9}$$

The reported spectral similarity is the mean of $\text{cos\_sim}_{t,c}$ over channels and time steps.

**Physics-based metrics**  In addition to these generic metrics we track physically meaningful global quantities for each PDE. All such quantities are defined as spatial integrals (or sums over grid points) of appropriate functions of the state variables. For Kolmogorov flow we monitor the kinetic energy $E(t)$ of the velocity field (obtained from the vorticity field) and compare the temporal evolution of $E(t)$ between reference and prediction. For the CDD model we monitor the spatial integral of the total dislocation density $\rho_t(t)$, which is known to follow a prescribed linear evolution according to the governing equations, and we assess how closely the learned surrogate reproduces this relationship. For Gray–Scott systems we track the total mass of the reactants (spatial integrals of $u$ and $v$), which should remain approximately conserved up to reaction terms, and compare their evolution under the learned models. For the advection equations we track conservation of total mass, which should remain constant during transport up to numerical errors.

For each architecture and dataset we report summary statistics of these diagnostics over the rollout horizon in section 3. The normalized absolute errors of the monitored quantities are visualized in Fig. 12.

## 2.6  Software and implementation details

All neural-network models are implemented in PyTorch (Paszke et al., 2019) using Python 3.10.12. Experiments are run on a workstation equipped with an NVIDIA RTX A6000 GPU with 48 GB of memory and a multi-core CPU, using CUDA 12.8 and cuDNN 91002. The me-UNet architecture and training loops are implemented from scratch, while the baseline models (ViT, AFNO, PDE-Transformer, KAN-UNet) build on publicly available reference implementations that we adapt to our input–output format and loss functions.

The synthetic datasets are generated with finite-element and finite-volume solvers implemented in C++/Python using FEniCS (Logg et al., 2012) for PDE-1–PDE-4, JAX-CFD (Kochkov et al., 2021) for PDE-5, and exponax (Koehler et al., 2024) for PDE-6. For each PDE dataset we fix the numerical scheme, time step, and resolution as described in sections 2.1 and 2.2. Random initial conditions are sampled using NumPy 1.26.4 with a fixed random seed, and we use consistent seeds across models to ensure that training and evaluation splits are identical.

All configuration files specifying all hyperparameters (learning rates, batch sizes, channel widths, etc.) are provided in the accompanying code repository. Where possible we also release pre-trained model checkpoints and scripts to regenerate all figures and tables from this paper, in order to facilitate reproducibility and further comparison.

## 3  Results

We now evaluate the performance of the different architectures on the PDE-generated datasets described in sections 2.1 and 2.2. Our goal is not to obtain the lowest possible error on a single benchmark by combining a very large model with massive training data. Instead, we focus on a realistic small-data regime with autoregressive rollouts of $T = 100$ time steps and ask the following questions:

1. How do convolutional, operator-based, and transformer-based architectures compare for *in-distribution* rollouts across five qualitatively different PDE families?

2. How robust are these models to *out-of-distribution* (OOD) initial conditions within a fixed PDE and parameter regime?

3. How does performance depend on the amount of available data and temporal context (number of simulations, time steps, and input sequence length)?

4. What do the models actually learn, and how do architectural inductive biases manifest in their internal activations?

All models are trained with the common setup described in section 2.4. For each dataset and architecture we train a separate model and generate autoregressive predictions starting from seven ground-truth snapshots; these seven frames form the input sequence, and the model predicts increments for the subsequent $T = 100$ time steps.

### 3.1 In-distribution prediction

We first consider in-distribution prediction, where training and test simulations are drawn from the same distribution of initial conditions for each PDE. The datasets are organized along a difficulty spectrum: DS-1 (linear advection) and DS-2 (linear diffusion) serve as sanity-check baselines for which any reasonable architecture should perform well, while DS-3 to DS-6 introduce progressively more nonlinear and multiscale dynamics. For each dataset and architecture, we generate rollouts of length $T = 100$ and compute the RMSE and spectral similarity defined in section 2.5.

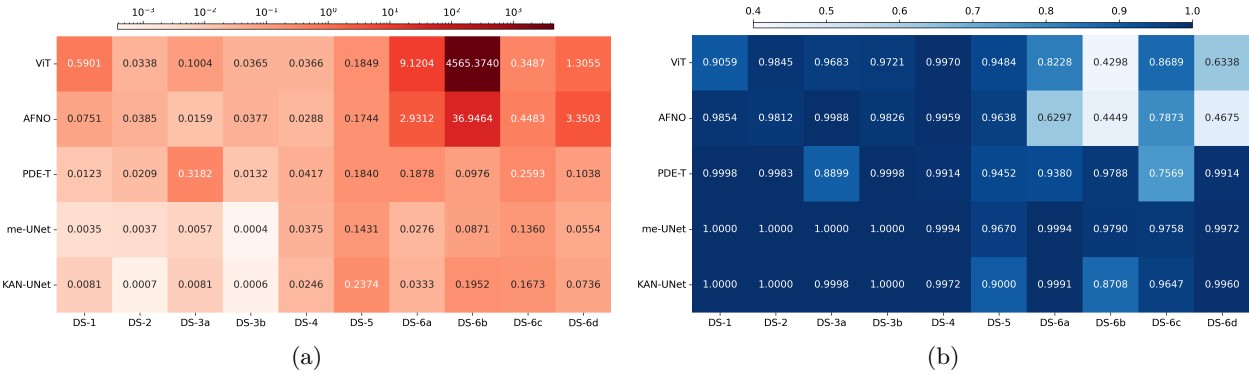

Figure 3: Comparison of average values over $T = 100$ time steps: (a) RMSE and (b) cosine-similarity scores of the power spectral density (PSD). Rows correspond to architectures and columns to datasets; colors are shown on a logarithmic scale for RMSE.

Figure 3 and figure 10a presents an overview of in-distribution performance across all datasets: panel (a) shows the average RMSE over $T = 100$ prediction steps, and panel (b) shows the corresponding average PSD cosine similarity. Rows correspond to the investigated neural network architectures, and columns to the datasets derived from the mathematical models in section 2.1. Lower RMSE indicates better field-space accuracy, while cosine similarity values closer to 1 indicate a closer match of the spatial frequency content.

Overall, me-UNet achieves the lowest RMSE on almost all datasets. In particular, its average RMSE on DS-1, DS-3a, DS-3b, DS-5, DS-6a, DS-6b, DS-6c, and DS-6d is $0.0035\pm0.0002$, $0.0057\pm0.0007$, $0.0004\pm0.00005$, $0.1431\pm0.02$, $0.0276\pm0.005$, $0.0871\pm0.03$, $0.1360\pm0.08$, and $0.0554\pm0.006$, respectively. The corresponding predicted snapshots show good agreement with the ground truth (see the me-UNet rows in Fig. 15a, Fig. 16a, Fig. 17a, Fig. 18a, Fig. 5a, Fig. 19a, Fig. 21a, and Fig. 20a). KAN-UNet achieves the lowest RMSE on DS-2 and DS-4 with $0.0007 \pm 0.00002$ and $0.0246 \pm 0.0125$, respectively (see the KAN-UNet rows in Fig. 15a and Fig. 4a).

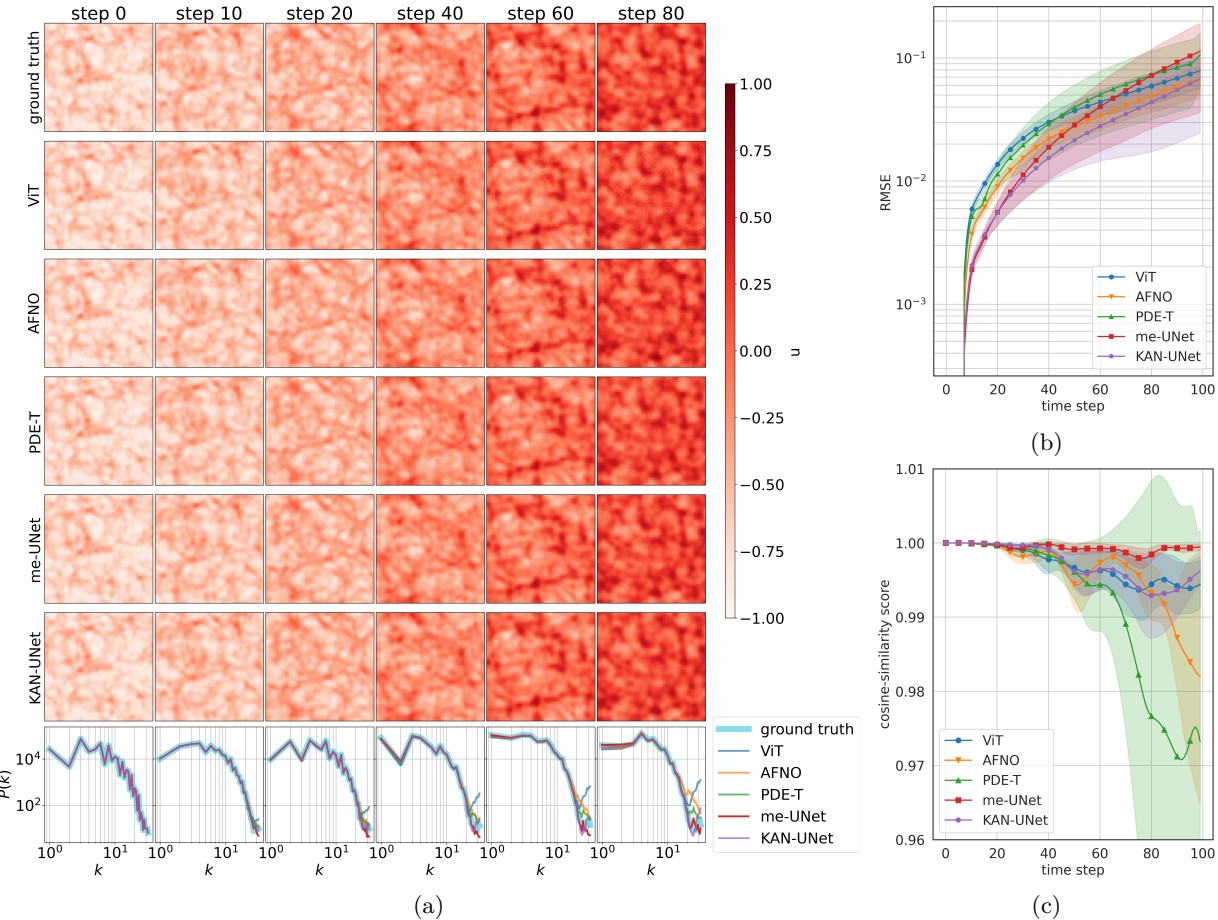

Figure 4: Performance of the different neural network architectures on the CDD dataset (DS-4): (a) example autoregressive rollouts; (b) per-time-step RMSE; and (c) cosine similarity of the PSD curves between prediction and reference.

In contrast, ViT attains the highest RMSE on several datasets (e.g., $0.5901 \pm 0.2$ on DS-1, $9.1204 \pm 18.96$ on DS-6a, and $4565.3740 \pm 1860.8349$ on DS-6b), which is reflected in visibly degraded predictions (see the first rows of Fig. 14a, Fig. 5a, and Fig. 19a). PDE-Transformer performs worst on DS-3a ($0.3182 \pm 0.06$, see Fig. 16a), while AFNO shows the largest errors on DS-6c ($0.4483 \pm 0.1384$) and DS-6d ($3.3502 \pm 2.2589$).

Spectrally, me-UNet also consistently achieves the highest PSD cosine similarity, with values close to 1.0 across all datasets. KAN-UNet is typically the second-best model, with cosine similarity near 1.0 except on DS-5, where it attains $0.9 \pm 0.07$. ViT and AFNO exhibit the lowest spectral similarity on the most challenging Gray–Scott datasets (e.g., $0.4298 \pm 0.02$ and $0.4449 \pm 0.02$ on DS-6b, and $0.4675 \pm 0.08$ on DS-6d), indicating substantial distortion of the spatial frequency content over long rollouts.

Figures 4 and 5 show representative in-distribution rollouts for the CDD dataset (DS-4) and the Gray–Scott dataset DS-6a. Each figure contains snapshot sequences, per-time-step RMSE, and PSD cosine similarity for all architectures. For CDD, me-UNet preserves both the large-scale microstructure and the fine dislocation patterns over $T = 100$ time steps, with RMSE remaining low and PSD similarity close to 1. Some baselines show accumulation of high-frequency artifacts and microstructure degradation over time. For Gray–Scott, me-UNet again produces stable pattern evolution with accurate spectral content, whereas several heavier architectures exhibit pattern drift and loss of small-scale structures.

Additional in-distribution rollouts for all datasets are provided in Appendix C, together with the corresponding PSD curves and cosine similarity plots.

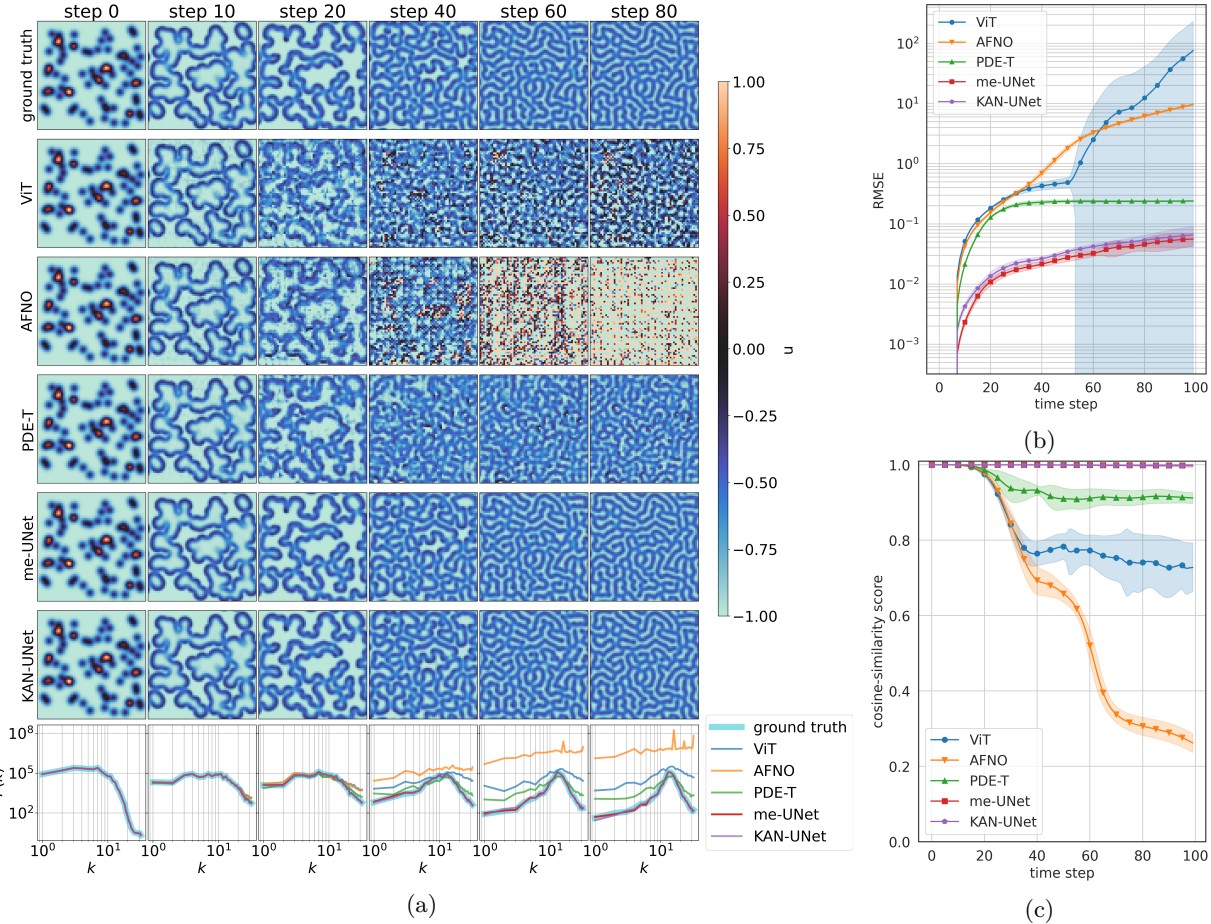

Figure 5: Performance of the different neural network architectures on the Gray–Scott dataset DS-6a: (a) example autoregressive rollouts; (b) per-time-step RMSE; and (c) cosine similarity of the PSD curves between prediction and reference.

Training and validation losses converge smoothly for me-UNet under the common protocol; ViT and AFNO show less stable training dynamics (see Fig. 11).

## 3.2 OOD prediction

We now investigate OOD prediction performance on the CDD dataset (DS-4) and the Gray–Scott dataset DS-6a, for which the corresponding in-distribution results are shown in Fig. 4 and Fig. 5. In both cases, the underlying PDE and parameters are kept fixed, but the initial conditions are qualitatively very different from those seen during training.

For CDD, we train on simulations with a very large number of superimposed blob- and loop-like initial patterns, resulting in images where the individual blobs or loops can no longer be identified. Testing is then performed on configurations with a very small number of loops or with mixed line and loop arrangements, which are visually very different from the training data. Figures 7a and 7b illustrate two such OOD cases. The main physical phenomenon—expansion of dislocation loops throughout the domain—is captured by most trained networks, and the predicted patterns remain close to the reference for me-UNet and KAN-UNet. The PSD curves in the last rows of Fig. 7 show a good match across a wide range of frequencies, with only small deviations at high frequencies for the best-performing models. AFNO, however, exhibits unstable behavior as time progresses (see the third rows of Fig. 7), leading to the highest RMSE values of 0.2353 and 0.2071 in the two OOD tests, as well as the lowest cosine similarities (0.9485 and 0.8907, see

Fig. 6a and Fig. 22a). The fact that me-UNet can extrapolate from densely populated microstructures to sparse, geometrically simple initial conditions strongly suggests that it has learned the underlying dislocation dynamics rather than merely memorizing typical training configurations.

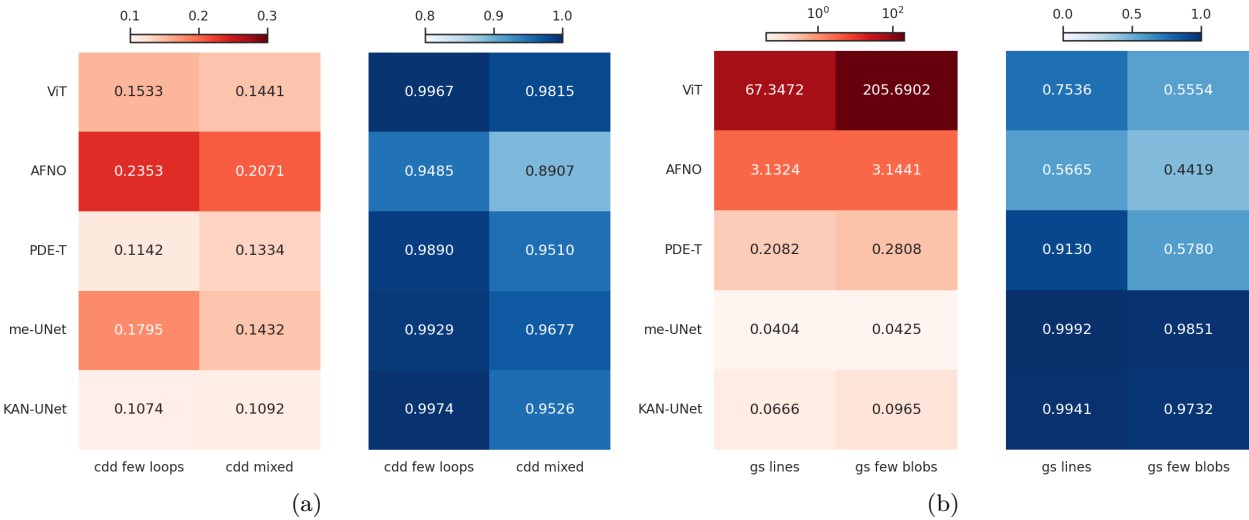

Figure 6: Comparison of average RMSE (red color scheme) and cosine-similarity score (blue color scheme) for OOD prediction on (a) the CDD dataset (DS-4) and (b) the Gray–Scott dataset DS-6a.

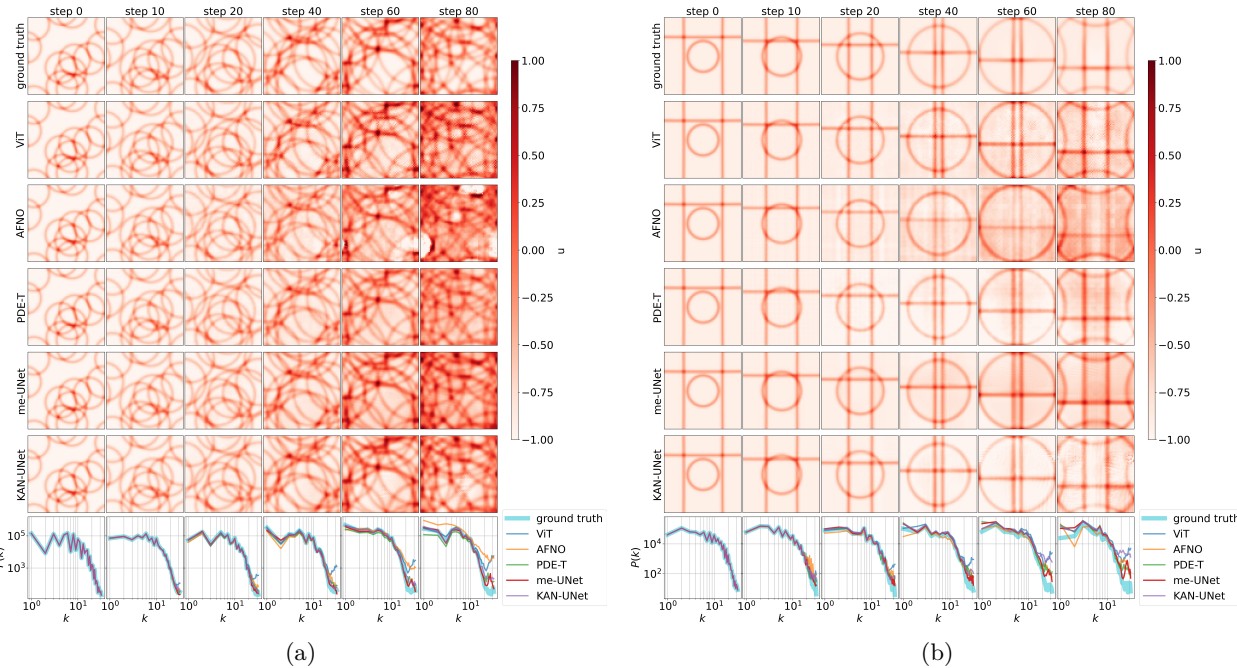

Figure 7: Example OOD prediction results of the different neural networks on the CDD model: (a) initial condition with several loops; (b) initial condition with three lines and one loop. The training data was visually very different but still contained the same underlying dynamical behavior that is also observed here.

For the Gray–Scott system, we train on randomly perturbed initial states and test on structured line and blob initial conditions that differ strongly from the training distribution. Figures 8a and 8b show OOD rollouts for line and blob initial patterns, respectively (compare the first columns of Fig. 5 and Fig. 8). Transformer-based models such as ViT, AFNO, and PDE-Transformer already struggle to train in-distribution on this

dataset and therefore perform poorly on OOD initial conditions; their predictions are very noisy and quickly diverge from the ground truth (see the second, third, and fourth columns of Fig. 8).

In contrast, me-UNet and KAN-UNet produce qualitatively correct OOD dynamics, capturing the reaction–diffusion behavior and generating patterns that remain similar to the reference (fifth and sixth columns of Fig. 8a). For me-UNet, the RMSE is lowest among all models (0.0404 and 0.0425 for the two OOD cases), and the cosine similarity is highest (0.9992 and 0.9851, see Fig. 6b and Fig. 22b). Some noise appears at later time steps due to accumulated prediction errors, but the overall patterns remain stable. ViT attains the highest RMSE (67.3472 and 205.6902), and AFNO reaches the lowest cosine similarities (0.5665 and 0.4419), indicating severe breakdown of OOD generalization for these heavy architectures.

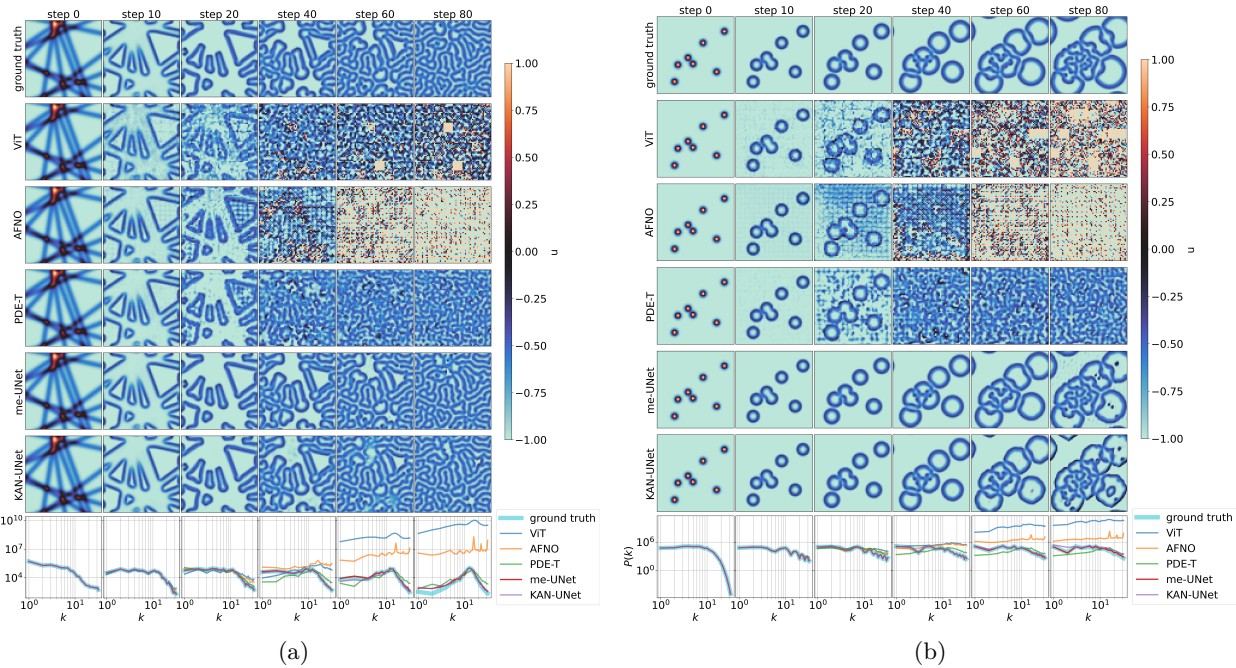

Figure 8: Example OOD prediction results of the different neural networks on the Gray–Scott model (DS-6a): (a) line-like initial condition; (b) blob-like initial condition.

An overview of OOD performance across architectures is given in Fig. 6, which summarizes average RMSE and PSD cosine similarity for the CDD and Gray–Scott OOD tests. me-UNet exhibits the most consistent behavior, with relatively small degradation from in-distribution to OOD initial conditions, while several baselines show substantial increases in error or spectral mismatch on at least one of the two systems. Physics-aware metrics, such as energy, mass, and total dislocation density, further corroborate these trends: Fig. 12 shows that me-UNet preserves these quantities more accurately than the baselines, especially under OOD conditions.

## 3.3 Data efficiency

Here we study data efficiency on the CDD dataset, which serves as a representative benchmark for evaluating how the architectures behave under data limitations. We do not repeat these ablations on all other datasets, but we expect qualitatively similar trends to hold.

We vary three factors: (i) the number of time steps per simulation used during training, (ii) the number of training simulations, and (iii) the input sequence length $L$ (number of past time steps provided as context). For each setting we retrain all architectures with the same protocol as in section 2.4 and evaluate in-distribution rollouts.

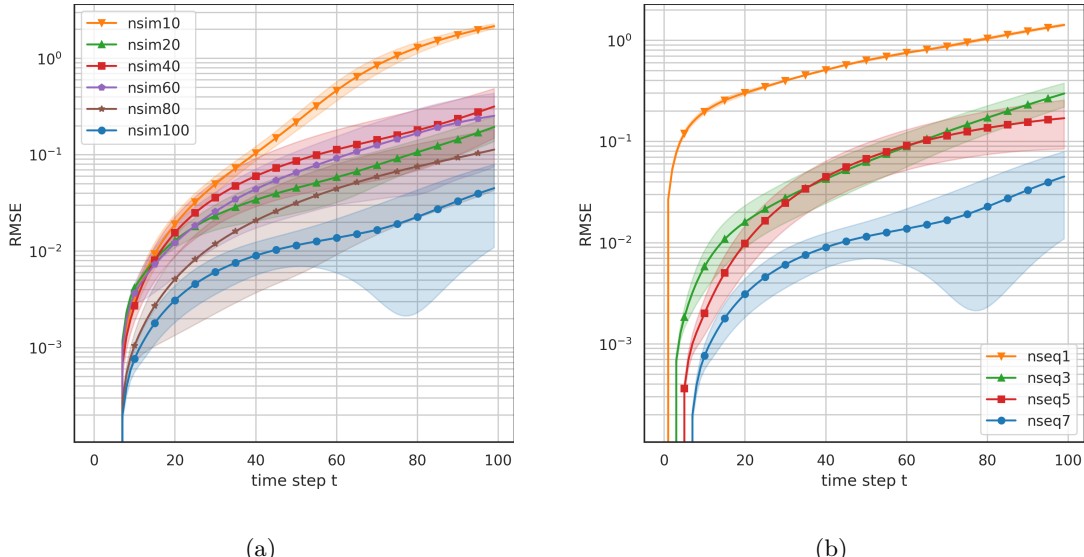

(a)              (b)

Figure 9: Data-efficiency study on the CDD dataset (DS-4): RMSE over 100-step rollouts as a function of (a) the number of training simulations, and (b) the input sequence length $L$ (number of past frames used as context).

Figure 9 summarizes the RMSE obtained under these variations. In panel (a), labels nsim10, nsim20, nsim40, nsim60, nsim80, and nsim100 indicate the number of simulations used for training (again with $L = 7$ and $T = 100$ time steps available). Here, me-UNet reaches a low-error regime with as few as 20 training simulations, with RMSE already close to the values obtained with 40, 60, or 80 simulations. In contrast, the transformer- and operator-based models benefit more noticeably from additional simulations, indicating that their data requirements are higher in this setting.

Finally, panel (b) varies the input sequence length $L \in \{1, 3, 5, 7\}$ while keeping the number of simulations and time steps fixed at 100. We find that using $L = 5$ or $L = 7$ input frames yields substantially lower RMSE than using a single-frame context ($L = 1$), confirming that limited temporal history is important for accurate next-step prediction. Performance saturates between $L = 5$ and $L = 7$, consistent with the choice of time step in the simulations: the dynamics change appreciably over a window of about six intervals, and longer contexts bring limited additional benefit.

Overall, these data-efficiency experiments show that, in the small-data regime, the convolutional inductive biases of me-UNet allow it to make more effective use of limited training data than the heavier baselines. They also provide practical guidance for selecting the number of simulations, time steps, and input sequence length in future studies on similar systems.

## 3.4 Studying the network behavior

In this section we analyze the internal behavior of me-UNet during training on the different mathematical models. We use Grad-CAM to visualize which spatial regions each block of the network attends to over the course of training. The resulting visualizations are shown in Figs. 23 to 32. The horizontal axis in these figures corresponds to training epochs, while the vertical axis indexes the U-Net blocks.

We denote the five encoder (downsampling) blocks by d0, d1, d2, d3, and d4, and the five decoder (upsampling) blocks by u0, u1, u2, u3, and u4. These blocks are annotated in Fig. 2. The spatial resolutions of the feature maps at d0–d4 are $64 \times 64$, $32 \times 32$, $16 \times 16$, $8 \times 8$, and $4 \times 4$, respectively, and the decoder resolutions are mirrored. For visualization purposes we show Grad-CAM maps every 40 epochs over the total of 1000 training epochs, since the early stages of training exhibit the most pronounced changes.

For each block, we compute Grad-CAM by multiplying the backward gradients with the forward activations, averaging over channels, and scaling the resulting values to the range $[0, 1]$. This highlights the spatial regions that contribute most strongly to the loss for each block at a given epoch, and therefore reveals which structures the network focuses on during learning.

We observe that image sequences from different datasets affect the attention patterns of me-UNet in characteristic ways. In the blocks closest to the input (d0, d1) and output (u3, u4), the model initially responds to a broad set of features, but as training progresses the activations concentrate on the occupied regions of the domain (e.g., dislocation lines, vortices, or reaction fronts) in most datasets (see Fig. 23a, Fig. 24a, Fig. 25a, Fig. 26a, Fig. 27, Fig. 28, and Fig. 29a). In some datasets (e.g., Gray–Scott DS-6b, DS-6c, and DS-6d), the model already focuses on low-level features from the first epochs (see Fig. 30a, Fig. 32a, and Fig. 31a).

In deeper encoder blocks (d2, d3, d4) and the corresponding decoder blocks (u0, u1, u2), the model concentrates on more global, semantic features. This is particularly evident for the advection dataset DS-1, where vertical streaks in the Grad-CAM maps reflect the horizontal motion of the advected structures (see Fig. 23). For the Gray–Scott datasets (DS-6a, DS-6b) and Kolmogorov flow (DS-5), deeper decoder blocks remain active throughout training (see Fig. 29, Fig. 30, and Fig. 28), reflecting the multi-scale nature of the pattern dynamics.

Taken together, these observations suggest that me-UNet leverages its convolutional and periodic inductive biases to allocate representational capacity in a physically meaningful way: shallow blocks capture local, fine-scale details, while deeper blocks encode global structures and long-range interactions. This aligns with the strong empirical performance of me-UNet in both in-distribution and OOD prediction tasks.

## 4 Discussion

Our experiments provide a comparative picture of how different deep-learning architectures behave as surrogates for 2D periodic PDEs in a small-data regime. Across all five PDE families, the proposed me-UNet achieves consistently low field-space error and high spectral similarity for in- distribution rollouts and remains stable over hundreds of time steps.

In the same setting, neural-operator and transformer-based models (AFNO, PDE- Transformer, ViT, KAN-UNet) display more heterogeneous behavior: some of them perform competitively on selected PDEs, but they are typically less robust across all datasets and more prone to the accumulation of high-frequency artifacts in long rollouts. When evaluated with physics-based metrics, me-UNet also tends to preserve conserved quantities such as energy (see Fig. 12c) and mass (see Fig. 12a and Fig. 12d) more accurately than the competing architectures, which suggests that local convolutions with periodic padding constitute a favorable inductive bias for these periodic PDEs.

More concretely, me-UNet performs well across all datasets DS-1–DS-6d, which span models ranging from simple diffusion to more complex systems such as Navier–Stokes (Kolmogorov flow) and continuum dislocation dynamics, using a single set of hyperparameters. In contrast, the performance of the neural- operator and transformer-based baselines varies strongly between datasets when trained with the same hyperparameters. For example, ViT and AFNO are competitive on the CDD (DS-4, Fig. 4) and Kolmogorov flow (DS-5, Fig. 18) datasets, but their performance degrades substantially on the Gray–Scott variants DS-6a (Fig. 5), DS-6b (Fig. 19), DS-6c (Fig. 21) and DS-6d (Fig. 20), where they tend to produce noisy or unstable patterns.

Beyond in-distribution performance, our results indicate that purely data- driven models can capture qualitative aspects of the dynamics even for initial conditions that differ substantially from those seen during training. For example, in the CDD OOD experiments shown in Fig. 7, the learned models reproduce the expansion and motion of individual loops or lines, although the training data (Fig. 4) contain only highly crowded superpositions in which the motion of individual objects is not visually apparent. The fact that me-UNet generalizes from these visually complex initial states to much simpler line- and loop-like configurations strongly suggests that it has internalized the underlying dislocation dynamics rather than merely memorizing typical microstructure textures.

Viewed through the lens of model complexity versus data complexity, our results suggest that, in the considered setting, additional architectural expressiveness does not automatically translate into better surrogates. The training datasets are relatively small, and the underlying dynamics, while nonlinear and sometimes chaotic, are governed by smooth fields on periodic domains. In this regime, a convolutional architecture with strong inductive biases—local filters, translation equivariance, multi-scale feature aggregation, and periodic padding—appears to be well matched to the data complexity, whereas higher-capacity models such as AFNO, PDE-Transformer, and ViT may be harder to optimize and more prone to overfitting or the build-up of high-frequency artifacts. We do not claim a universal relationship between model capacity and performance for PDEs, but our empirical study highlights that carefully chosen inductive biases can be more important than sheer architectural complexity when data are scarce.

Our notion of out-of-distribution generalization is deliberately restricted to initial-condition shifts within a fixed PDE and parameter regime on periodic domains, as defined in Section 2.2. We do not claim generalization across PDE families, parameter ranges, domain geometries, boundary conditions, or resolutions; extending the benchmark and architectures to such forms of distribution shift (parameter OOD, PDE-family OOD, and domain OOD) is an important direction for future work.

Additionally, me-UNet works well with a relatively small amount of data. Based on the experimental results shown in Fig. 9b, sufficient performance on CDD is obtained with only 20 training simulations. While many works require a substantially larger number of simulations for training—for example, (Oommen et al., 2022) use 2000 simulations to train DEEPONET, and the smallest number of simulations in (Kontolati et al., 2024) is more than 260—our results suggest that, at least for CDD, a modest number of simulations can suffice when using an architecture with appropriate inductive biases. In terms of data scale, our setting sits between physics-informed and hybrid solver–ML approaches on one hand — which can train with sparse or zero labeled trajectories by leveraging the governing equations (Raissi et al., 2019; Ren et al., 2022; Kochkov et al., 2021) — and asymptotic operator-learning regimes that typically rely on $10^3$ to $10^4$ trajectories (Li et al., 2020; Lu et al., 2021a). The question we ask is complementary: how far do architecture-level inductive biases alone substitute for large-$N$ sample averaging in this intermediate regime? Our results suggest that locality and periodic padding go a substantial way, while neither replacing equation-level supervision nor matching the asymptotic guarantees of large-$N$ operator learning. Furthermore, the time required for training me-UNet is much lower than that of the state-of-the-art architectures considered here. Training from scratch for 1000 epochs with a batch size of 40 on an NVIDIA RTX A6000 GPU takes roughly 5 hours for me-UNet, while ViT, AFNO, PDE-Transformer, and KAN-UNet take approximately 6, 42, 12, and 9 hours, respectively.

Notably, several of the benchmarks are only partially observed during training (one scalar field per system; cf. Table 1). The strong performance of me-UNet suggests that short temporal context can provide an effective data-driven closure for the chosen observable in these periodic small-data regimes.

Moreover, Grad-CAM provides a qualitative window into the representations learned by me-UNet during training. It highlights which spatial regions and U-Net blocks most influence the prediction loss and how this focus evolves over epochs (see Appendix I). Because the Grad-CAM maps are aggregated over batches that span all simulation time steps, they do not resolve individual physical times; nevertheless, they reveal dataset-dependent signatures such as streak- or cross-shaped activations in transport-dominated systems and more localized responses around reaction fronts or vortex cores for the Gray–Scott and Kolmogorov-flow datasets. These patterns indicate that me-UNet learns internal representations that reflect the dominant transport and pattern-formation mechanisms in each benchmark, consistent with its strong in-distribution and OOD performance.

This benchmark is deliberately scoped to highlight small-data autoregressive forecasting on 2D periodic grids, which leaves several important extensions open. First, all experiments are carried out on two-dimensional, structured, periodic grids with a fixed spatial resolution. While this setting is representative of many canonical benchmarks, it does not cover unstructured meshes, complex geometries, or adaptive resolution. Second, the architectures are trained separately for each PDE family rather than as a single multi-task model, and we have not attempted extensive per-architecture hyperparameter tuning beyond a common training protocol. Third, although we incorporate physics-based metrics that quantify the conservation of selected invariants and show that me-UNet can preserve them reasonably well, we only *encode periodic*

*boundary conditions* explicitly, via periodic padding; we do not impose any hard conservation constraints in the architecture or loss. Future work could combine the periodic U-Net backbone with physics-informed loss terms or hard constraints to more strictly enforce mass, energy, or other invariants, and could explore similar inductive biases within neural-operator architectures designed for more general domains.

In addition, me-UNet, like most traditional CNN-based approaches, is naturally suited to image-like datasets where values are distributed on a structured grid, because it relies on fixed filter kernels (e.g., a $3 \times 3$ kernel for most 2D convolutional operators). Such architectures struggle to handle unstructured grids with irregular shapes and non-uniform node distributions. To address this, it may be necessary either to combine me-UNet with techniques that transform an unstructured grid into a structured grid, such as space-filling curves (Heaney et al., 2024), or to use graph convolutional networks (Jiang et al., 2019) to perform convolution-like operations on irregular data.

Furthermore, the current version of me-UNet is still limited to 2D simulation results; however, there is room to extend the architecture to 3D problems, which will be the next step following this work. The possible grid size used as input for me-UNet is not limited to $64 \times 64$, but can be increased to higher-resolution images (e.g., $128 \times 128$, $256 \times 256$, $512 \times 512$, or higher), depending on the datasets produced, the availability of GPU hardware, and the purpose of the research. Also, the datasets produced for this study all have periodic boundary conditions in both the horizontal and vertical directions for comparison purposes. Our neural network architecture is in principle able to train with different types of boundary conditions (e.g., Dirichlet boundaries) by changing the padding technique, as discussed in section 2.3. The generalization ability across boundary conditions is another important aspect to be addressed in future work. From a theoretical standpoint, recent studies have established in-distribution generalization guarantees for ML-based PDE solvers using tools from statistical learning theory (Lu et al., 2021c;b), while extending these results to out-of-distribution settings remains largely open; combining such analyses with emerging OOD generalization frameworks that emphasize invariance and stability across varying PDE configurations (Ye et al., 2021), may provide a principled explanation of cross-condition generalization, which we leave for future investigation.

Finally, we emphasize again that in this work PDEs serve both as scientifically important models and as controlled, interpretable generators of high-dimensional spatio-temporal data. A surrogate model that cannot robustly learn dynamics generated by well-posed PDEs under controlled numerical conditions is unlikely to succeed on heterogeneous, noisy measurement data where the governing equations are only approximately known. Conversely, architectures that perform well under the small-data and OOD-initial-condition regime studied here are promising candidates for subsequent evaluation on real experimental and observational datasets.

Several extensions are natural follow-ups to the present scope. First, varying the PDE parameters at training time — e.g., propagation speeds, diffusion coefficients, Reynolds numbers — would test parameter-OOD generalization, which is closer to many practical scientific scenarios than the initial-condition-OOD setting we study here. Second, systematically varying propagation speeds and time steps for a fixed PDE would probe the relationship between effective Courant number and learnability, particularly in the small-data regime. Third, richer initial-condition distributions (mixtures of smooth and discontinuous cases, parametric families of geometric structures) would more thoroughly stress-test the OOD-generalization claim. Each of these axes opens a substantial sub-study — e.g., parameter-conditioned architectures, multi-task training across regimes — that we view as the basis for follow-up work building on the architectural baseline established here.

## 5 Conclusion

We have presented a systematic empirical study of autoregressive deep-learning surrogates for two-dimensional periodic PDEs in a small-data regime. Across five representative PDE families, we compared the proposed me-UNet to several recent neural-operator and transformer-based architectures under a common training protocol. Using field-space error, spectral similarity, and physics-based metrics to quantify the preservation of invariants such as energy and mass, we found that me-UNet consistently provides accurate and stable rollouts for in-distribution test cases while requiring substantially less training time than the competing architectures.

Within the considered setting, me-UNet also exhibits robust generalization to out-of-distribution initial conditions whose spatial structure differs markedly from the training data, including cases where the training set consists of highly crowded microstructures and the tests involve only a few well-separated objects. Together with the data-efficiency experiments, this supports the view that strong inductive biases in convolutional U-Nets—local filters, translation equivariance, multi-scale feature aggregation, and periodic padding that matches the boundary conditions—can be more beneficial than additional model complexity when data are scarce and the underlying dynamics are smooth on periodic domains.

At the same time, our objective is not to maximize the intrinsic complexity of the underlying PDE systems, but rather to benchmark and compare the behavior of different neural architectures in controlled dynamical settings. Therefore, our study is limited to 2D, structured, periodic grids at a fixed resolution, and to initial-condition OOD within a fixed PDE and parameter regime. Extending the benchmark and models to parameter and geometry shifts, non-periodic boundary conditions, unstructured meshes, and three-dimensional problems, and combining the periodic U-Net backbone with physics-informed loss terms or hard constraints, are important directions for future work. More broadly, PDEs in this work serve as controlled, interpretable generators of high-dimensional spatio-temporal data, and we expect that the same architectural and evaluation principles will also be relevant for surrogate modeling and forecasting in other domains where the dynamics are described by, or well approximated by, continuum models and related dynamical systems.

## Competing interests

The authors declare no competing interests.

## Data availability

All training datasets used in this study, together with scripts to generate them from the underlying PDE solvers, are provided in anonymized form (`https://drive.proton.me/urls/7DH6NBT8A0#2C8h9BmFRgdx`) as part of the supplementary material for review and will be released in a public repository upon acceptance.

## Code availability

All code required to reproduce the experiments, including model implementations, training scripts, and configuration files, is provided in anonymized form (`https://drive.proton.me/urls/7DH6NBT8A0#2C8h9BmFRgdx`) as part of the supplementary material for review and will be released in a public repository upon acceptance.

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

## A  Loss function

### A.1  MSE loss

The mean-squared error (MSE) loss is widely used in the machine-learning community. It computes the average squared difference between the predicted and true values:

$$\mathcal{L}_{\mathrm{MSE}} = \frac{1}{N} \sum_{i=1}^{N} (\hat{y}_i - y_i)^2, \tag{10}$$

where $N$ is the number of data points, $y_i$ is the true value, and $\hat{y}_i$ is the predicted value for the $i$-th sample.

### A.2  Perceptual loss

Perceptual loss was introduced by Johnson et al. (Johnson et al., 2016); it is computed based on features extracted from a pre-trained network rather than directly from pixel-wise differences. Perceptual loss captures more abstract and global image qualities that are often more relevant for human perception, whereas pixel-based losses such as MSE penalize local differences and can lead to overly smooth or blurry images:

$$\mathcal{L}_{\mathrm{pc}} = \sum_{i} \left\| \Phi_i(\hat{y}) - \Phi_i(y) \right\|^2, \tag{11}$$

where $\Phi_i$ denotes the feature map from the $i$-th layer of the pre-trained network, and $y$ and $\hat{y}$ are the true and predicted images, respectively. In this work, we use VGG-16 as the pre-trained network to obtain feature maps.

## B  Detailed description of the mathematical models

**Advection equation**  The advection (or transport) equation is a fundamental mathematical model used to describe a variety of physical phenomena in fields such as biology, chemistry, and materials science. It predicts how a quantity is transported over time due to the motion of particles or a background flow.

The general form of the advection equation in our setting is

$$\frac{\partial u}{\partial t} = -(v_x \frac{\partial u}{\partial x} + v_y \frac{\partial u}{\partial y}), \tag{12}$$

where $u$ is the scalar quantity being transported (e.g., temperature or concentration), and $v_x = 1$, $v_y = 0$ are the velocity components in the $x$- and $y$-directions.

The initial values consist of "blobs" introduced into the system as a Gaussian exponential function,

$$c = \exp\left(-a\left((x - x_0)^2 + (y - y_0)^2\right)\right). \tag{13}$$

The number of these blobs is randomly chosen within the range $(1, 10)$, and the blobs are randomly positioned inside the quadratic domain with edge length $l_x = l_y = 10$. For the Gaussian distribution the length scale parameter was chosen as $a = 10$. The spatial mesh is $100 \times 100$.

The simulation is implemented in FEniCS, an open-source computing platform for solving PDEs with the finite element method. Each simulation is run for 100 time steps, and the field is saved at every step. The time step size is 0.01, and a Streamline-Upwind Petrov–Galerkin (SUPG) stabilization term (Brooks & Hughes, 1982) is used to prevent numerical oscillations and instabilities in this convection-dominated problem. We use the default "Sparse LU" solver provided by FEniCS.

**Diffusion equation**   The diffusion equation is another fundamental mathematical model used to describe a wide range of physical processes, including heat conduction and the spreading of chemical species. It predicts how a quantity spreads out over time due to random motion of particles.

The general form of the diffusion equation is

$$\frac{\partial u}{\partial t} = c_x \frac{\partial^2 u}{\partial x^2} + c_y \frac{\partial^2 u}{\partial y^2}, \tag{14}$$

where $u$ is the diffusing quantity (e.g., temperature), and $c_x = c_y = 1$ are the diffusion coefficients in the $x$- and $y$-directions.

As before, the number of initial blobs is randomly chosen in the range $(1, 10)$ and the blobs are randomly distributed inside the domain with $l_x = l_y = 10$. Each blob again is again obtained from a Gaussian function eq. (13) with $a = 10$. The spatial mesh is $100 \times 100$.

The simulation is implemented in FEniCS and run for 100 time steps with the field saved at every step. The time step size is 0.01, and we use the default "Sparse LU" solver for this diffusion equation.

**Reaction–diffusion equation (Gray–Scott model)**   A two-dimensional Gray–Scott system (Pearson, 1993) describes the behavior of two reacting and diffusing substances by

$$\frac{\partial u}{\partial t} = -uv^2 + c_u \Delta u + f(1 - u), \tag{15}$$

$$\frac{\partial v}{\partial t} = \quad uv^2 + c_v \Delta v - (f + k)v, \tag{16}$$

where $u$ and $v$ are the concentrations of the two reacting species $U$ and $V$, $c_u$ and $c_v$ are their diffusion coefficients, and $f$ and $k$ are feed and kill rates.

We use the exponax package (Koehler et al., 2024) to simulate a set of Gray–Scott systems with different parameter choices corresponding to DS-6a, DS-6b, DS-6c, and DS-6d; the parameters are given in Tab. 2. The spatial mesh sizes are either $128 \times 128$ or $256 \times 256$, as listed in the table. Each simulation is run for 200 time steps, and the field values are saved every 2 steps, resulting in 100 stored snapshots per simulation.

Table 2: Gray–Scott model simulation parameters.

|        | Feed rate $f$ | Kill rate $k$ | # Gaussian blobs | Spatial mesh |
|--------|---------------|---------------|------------------|--------------|
| DS-6a  | 0.03          | 0.0565        | 50–200           | $256 \times 256$ |
| DS-6b  | 0.018         | 0.051         | 5–10             | $128 \times 128$ |
| DS-6c  | 0.058         | 0.065         | 5–10             | $128 \times 128$ |
| DS-6d  | 0.082         | 0.059         | 5–10             | $128 \times 128$ |

The four configurations are drawn from qualitatively different regions of the Pearson phase diagram: DS-6a corresponds to the self-replicating maze / labyrinth regime, DS-6b to the $\alpha$ regime, DS-6c to the "worm"

regime characterized by extended one-dimensional structures, and DS-6d to the bubble (spot-replication) regime.

**Navier–Stokes equation (Kolmogorov flow)**    The Navier–Stokes equations (Kolmogorov) describe the motion of incompressible fluid flows:

$$\frac{\partial \boldsymbol{u}}{\partial t} = -\nabla \cdot (\boldsymbol{u} \otimes \boldsymbol{u}) + \frac{1}{\text{Re}} \nabla^2 \boldsymbol{u} - \frac{1}{\rho} \nabla p + \boldsymbol{f}, \tag{17}$$

$$\nabla \cdot \boldsymbol{u} = 0, \tag{18}$$

where $\boldsymbol{u}$ is the velocity field, $\boldsymbol{f}$ is an external forcing, $\rho$ is the density, Re is the Reynolds number, and $p$ is the pressure. In two dimensions, the scalar vorticity field

$$\omega = \partial_x u_y - \partial_y u_x$$

is often used to characterize incompressible flows.

We simulate Kolmogorov flow for decaying turbulence using the open-source JAX-CFD code (Kochkov et al., 2021), which employs finite-volume/finite-difference discretizations on a staggered grid (pressure at cell centers and velocity components on corresponding faces). The spatial grid size is $256 \times 256$, and each simulation runs for 100 time steps with the vorticity field saved at every step. The density is set to $\rho = 1$, the Reynolds number to Re = 1000, and the Kolmogorov forcing terms are kept at the default values provided in the package examples. Kolmogorov flow is known to be unstable and chaotic, so small numerical errors can accumulate over time and cause trajectories to diverge.

**Continuum dislocation dynamics (CDD) model**    Continuum dislocation dynamics (CDD) (Hochrainer et al., 2014) is a set of PDEs that describe the transport and evolution of curved lines under a given velocity field. Lines can be represented as quasi-discrete, single lines or even as "smeared-out" continuum-like bundles of many lines (because the PDEs are the result of a statistical field theory).

In the 2D setting considered here, we assume a velocity $\boldsymbol{v}$ with constant magnitude $|\boldsymbol{v}| = \text{const}$ that is by construction always perpendicular to the local line tangent. As a consequence, a circular dislocation loop will expand, whereas a straight line segment will experience only pure translation. The key difference between these two extreme geometrical objects is that expanding loops increase the total line length in the system, while straight segments merely move without changing the total length.

This purely geometric behavior is—in the field of materials science—central to dislocation plasticity, where the motion and evolution of line-like defects inside metals and semiconductors govern plastic deformation, as demonstrated for example in Sandfeld and Po (Sandfeld & Po, 2015). There, the purely geometrical aspects described above are central aspects.

In this work we consider a CDD model following the formulation in (Hochrainer et al., 2014). The state is represented by three fields: the total density $\rho_{\text{t}}$ (whose volume integral gives the total line length inside the volume), a vector of geometrically necessary dislocation density $\boldsymbol{\varrho} = [\varrho_{\text{s}}, \varrho_{\text{e}}]$ (corresponding to screw and edge dislocations), and a curvature density $q_{\text{t}}$ (whose volume integral gives the number of closed loops as a multiple of $2\pi$). The temporal evolution is governed by transport-type equations

$$\begin{aligned} \partial_t \rho_{\text{t}} &= -\nabla \cdot (v \boldsymbol{\varrho}^{\perp}) + v q_{\text{t}}, \\ \partial_t \boldsymbol{\varrho} &= -\nabla(v \rho_{\text{t}}), \\ \partial_t q_{\text{t}} &= -\nabla \cdot \left(-v \boldsymbol{Q}^{(1)} + \boldsymbol{A}^{(2)} \cdot \nabla v\right), \end{aligned} \tag{19}$$

where a dot denotes the scalar product and $\boldsymbol{\varrho}^{\perp} = [\varrho_{\text{e}}, -\varrho_{\text{s}}]$ is the 90°-rotated GND density vector. Following (Monavari et al., 2014), we assume

$$\begin{aligned} \boldsymbol{Q}^{(1)} &= -\boldsymbol{\varrho}^{\perp} \frac{q_{\text{t}}}{\rho_{\text{t}}}, \\ \boldsymbol{A}^{(2)} &= \frac{\rho_{\text{t}}}{2} \left[(1 + \Psi) \, \boldsymbol{l}_{\varrho} \otimes \boldsymbol{l}_{\varrho} + (1 - \Psi) \, \boldsymbol{l}_{\varrho^{\perp}} \otimes \boldsymbol{l}_{\varrho^{\perp}}\right], \end{aligned} \tag{20}$$

where $\boldsymbol{l}_\varrho = \boldsymbol{\varrho}/|\boldsymbol{\varrho}|$ is the average line direction and $\boldsymbol{l}_{\varrho^\perp}$ is the unit vector perpendicular to $\boldsymbol{l}_\varrho$. The anisotropy parameter $\Psi$ is approximated as

$$\Psi \approx \frac{(|\boldsymbol{\varrho}|/\rho_{\rm t})^2 \left(1 + (|\boldsymbol{\varrho}|/\rho_{\rm t})^4\right)}{2}. \tag{21}$$

We implement the CDD model in FEniCS. For simplicity, we assign a constant velocity $v$, corresponding to dislocation glide on a fixed slip system, and neglect interactions between dislocation loops (e.g., annihilation, cross slip). The number of circular dislocation loops is chosen randomly in the range $[50, 500]$ with radius $r_0 = 400b$. The spatial mesh is $100 \times 100$, and each simulation runs for 1000 time steps with fields saved every 10 time steps (resulting in 100 stored snapshots). We consider an ensemble of dislocation loops with the same Burgers vector $\boldsymbol{b}$, and a domain of size $l_x = l_y = 2000b$ with $b = 0.256\,\text{nm}$, $v = 1 \cdot 10^5\,\text{nm}\,\text{s}^{-1}$, $dt = 1 \cdot 10^{-5}\,\text{s}$. Periodic boundary conditions are used in both directions so that loops leaving the domain on one side re-enter on the opposite side. Further details and parameter choices can be found in (Sandfeld & Po, 2015).

**Reduced CDD model**   To produce the datasets DS-3a and DS-3b, containing advection-like or diffusion-like motion of loop distributions, we retain only the first two equations in eq. (19), drop the curvature-density term and add the diffusion terms. The equation becomes,

$$\partial_t \rho_{\rm t} = -\nabla \cdot (v\boldsymbol{\varrho}^\perp) + D\nabla^2 \rho_{\rm t},$$
$$\partial_t \boldsymbol{\varrho} = -\nabla(v\rho_{\rm t}) + D\nabla^2 \boldsymbol{\varrho}, \tag{22}$$

All parameters for solving this equation are similar to that of the CDD model. We assume a velocity $\boldsymbol{v}$ with constant magnitude $|\boldsymbol{v}| = \text{const}$ and diffusion coefficient $D = 0$ for dataset DS-3a; $|\boldsymbol{v}| = 0$ and $D = \text{const}$ for DS-3b.

## C   Comparison of average values over $T - 100$ time steps

Figure 10 presents an overview of in-distribution and OOD performance across all datasets: panel (a) shows the average RMSE over $T = 100$ prediction steps, and panel (b) shows the corresponding average PSD cosine similarity. Lower RMSE indicates better field-space accuracy, while cosine similarity values closer to 1 indicate a closer match of the spatial frequency content.

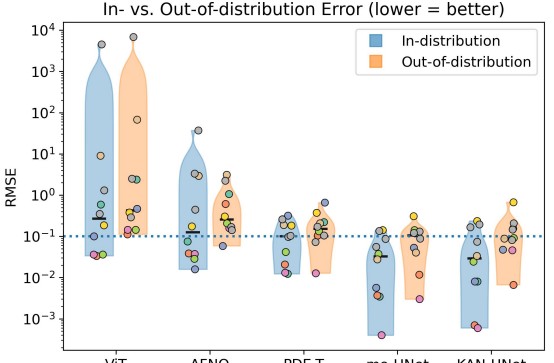 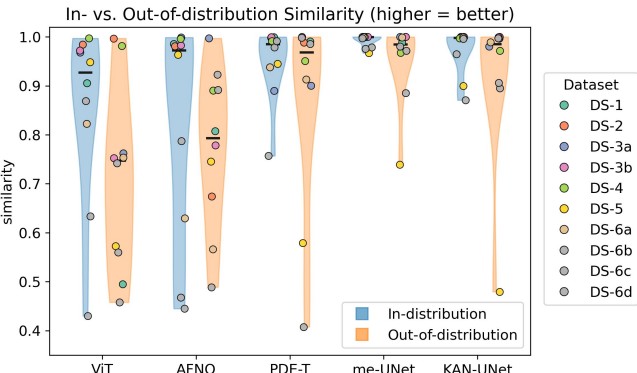

Figure 10: Comparison of average values over $T = 100$ time steps: Left panel visualize RMSE and right panel visualize cosine-similarity scores of the power spectral density (PSD).

# D   Training behaviour

Figure 11 shows the convergence behavior of training and validation loss for the CDD and Gray–Scott DS-6a datasets. me-UNet converges faster and more smoothly than the other architectures, without evidence of overfitting, while ViT and AFNO have difficulty training with the limited data. PDE-Transformer converges on DS-4 but struggles on DS-6a, consistent with its weaker predictive performance on the latter.

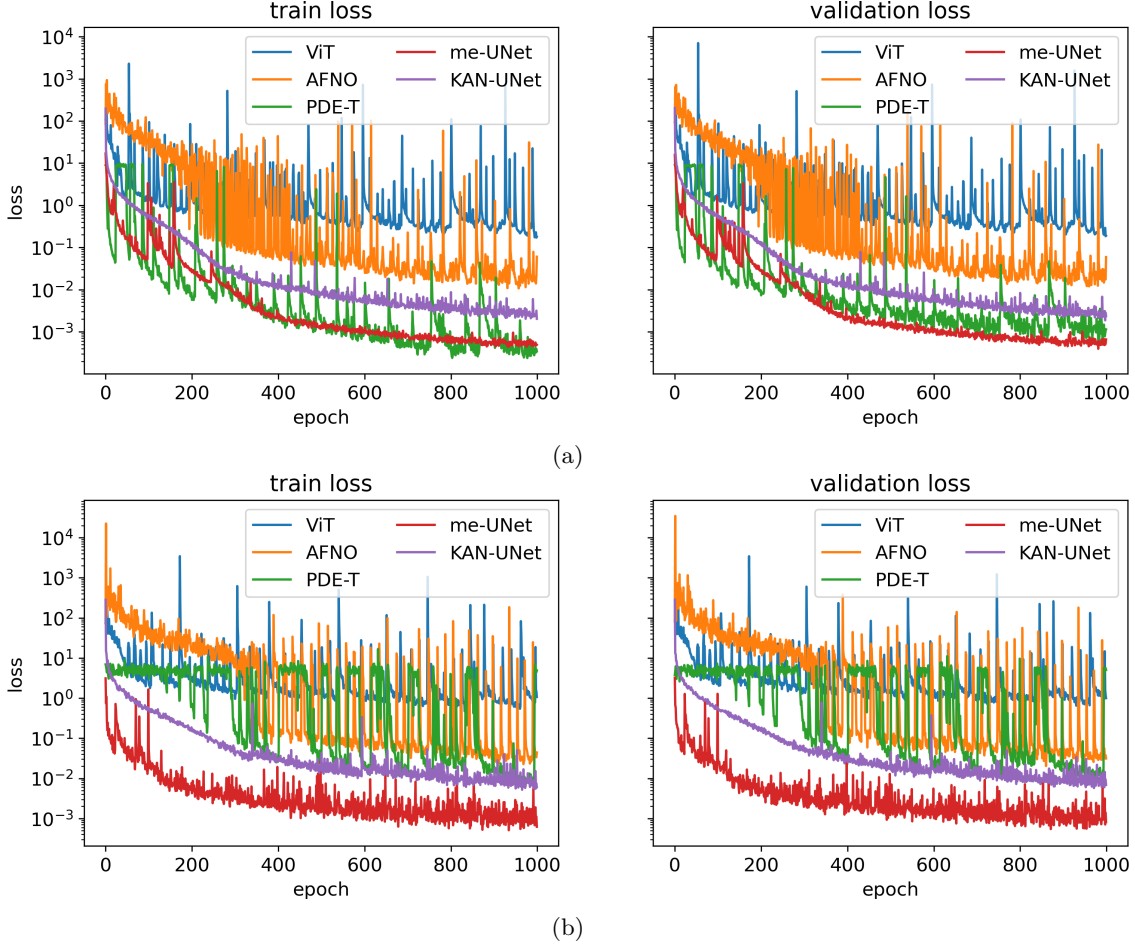

Figure 11: Training and validation loss curves (MSE + perceptual loss) for the different architectures on (a) the CDD dataset (DS-4) and (b) the Gray–Scott dataset DS-6a. me-UNet converges reliably and shows no signs of overfitting under the common training setup, whereas ViT and AFNO struggle in the small-data regime.

# E   Visualization of physics-based metrics across architectures

In this section we provide additional plots of the physics-based metrics introduced in section 2.5 for in-distribution rollouts. Each panel in Fig. 12 shows the normalized absolute error of a conserved or approximately conserved quantity (e.g., mass, energy, total dislocation density) as a function of time for all architectures. Across the four datasets, me-UNet generally yields the smallest deviation from the reference curves over 100 time steps, with KAN-UNet often second-best, and the transformer- and operator-based models showing larger drift.

The accumulated error for DS-4 (Fig. 12b, Fig. 13a and Fig. 13b) may arise from the large range of values between the minimum and maximum of the original dataset (before scaling to $[-1, 1]$). Since the number of

loops used as initial values spans a wide range (50–500 loops), the number of simulations may be relatively small compared to the variability in total line length. This can result in a systematic shift of the spatial integral of the state variable (see Fig. 13a,b). However, the main physical behavior of localization (i.e., the overall patterns) is still captured by the trained models (see Fig. 7).

For the other datasets, such as DS-6a (Fig. 12d, Fig. 13c and Fig. 13d), the minimum and maximum values of the original data for all simulations, both in-distribution and OOD, lie in a much narrower range. In these cases, the physical phenomena are well captured, both in terms of localization and generalization behavior (see Fig. 8, Fig. 13c,d).

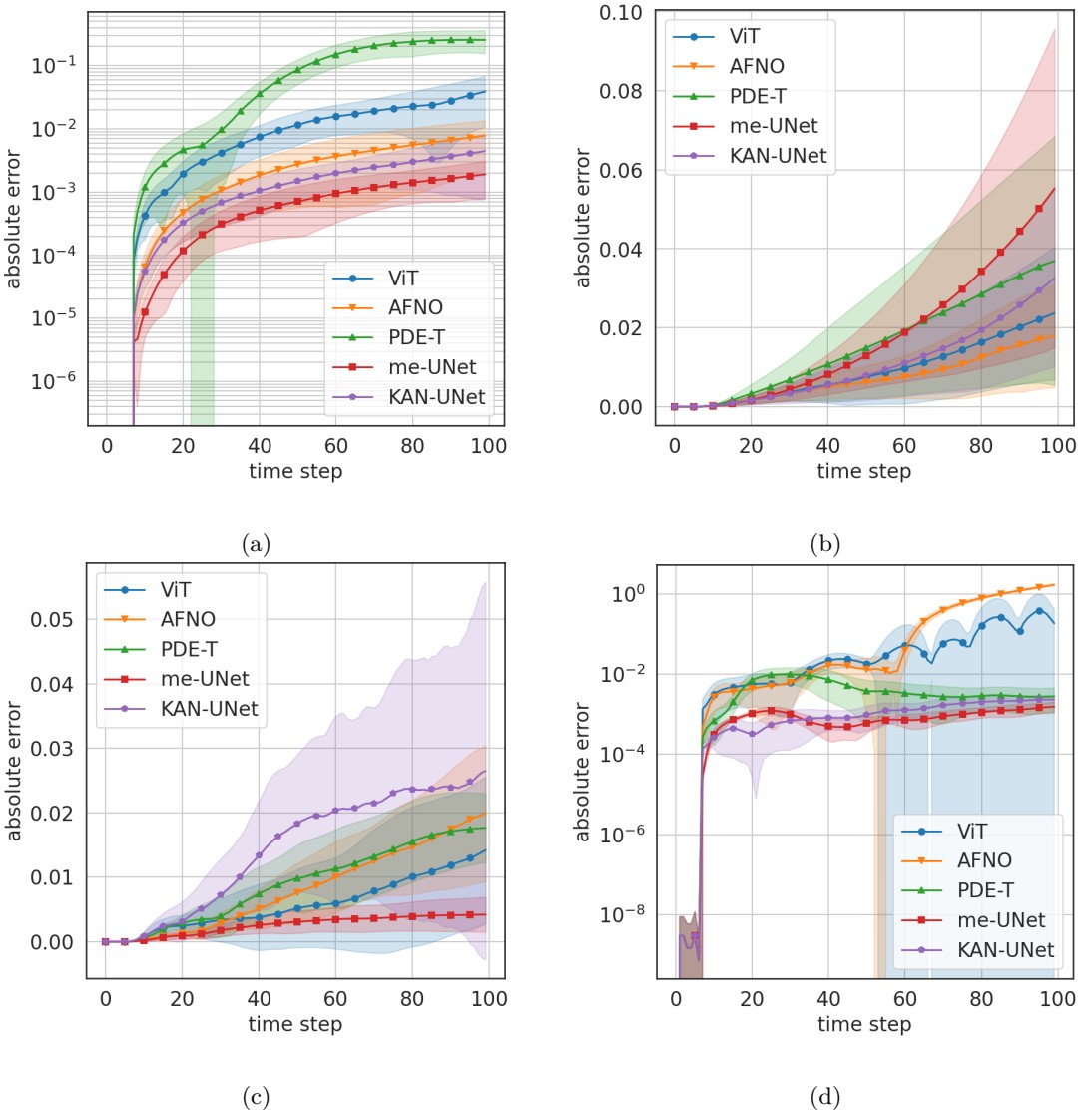

Figure 12: Normalized absolute error of physics-based metrics for (a) DS-3a, (b) DS-4, (c) DS-5, and (d) DS-6a.

# F  Visualization of physics-based metrics for OOD initial conditions

In this section we show the corresponding physics-based metrics for the OOD initial-condition experiments described in section 3. Figure 13 displays spatial integrals of the monitored quantities for CDD and Gray–Scott rollouts under different OOD initial states. As in the in-distribution case, me-UNet tracks the reference integrals most closely over time, while several baselines exhibit noticeable drift, particularly for the more challenging line- and blob-type initial conditions.

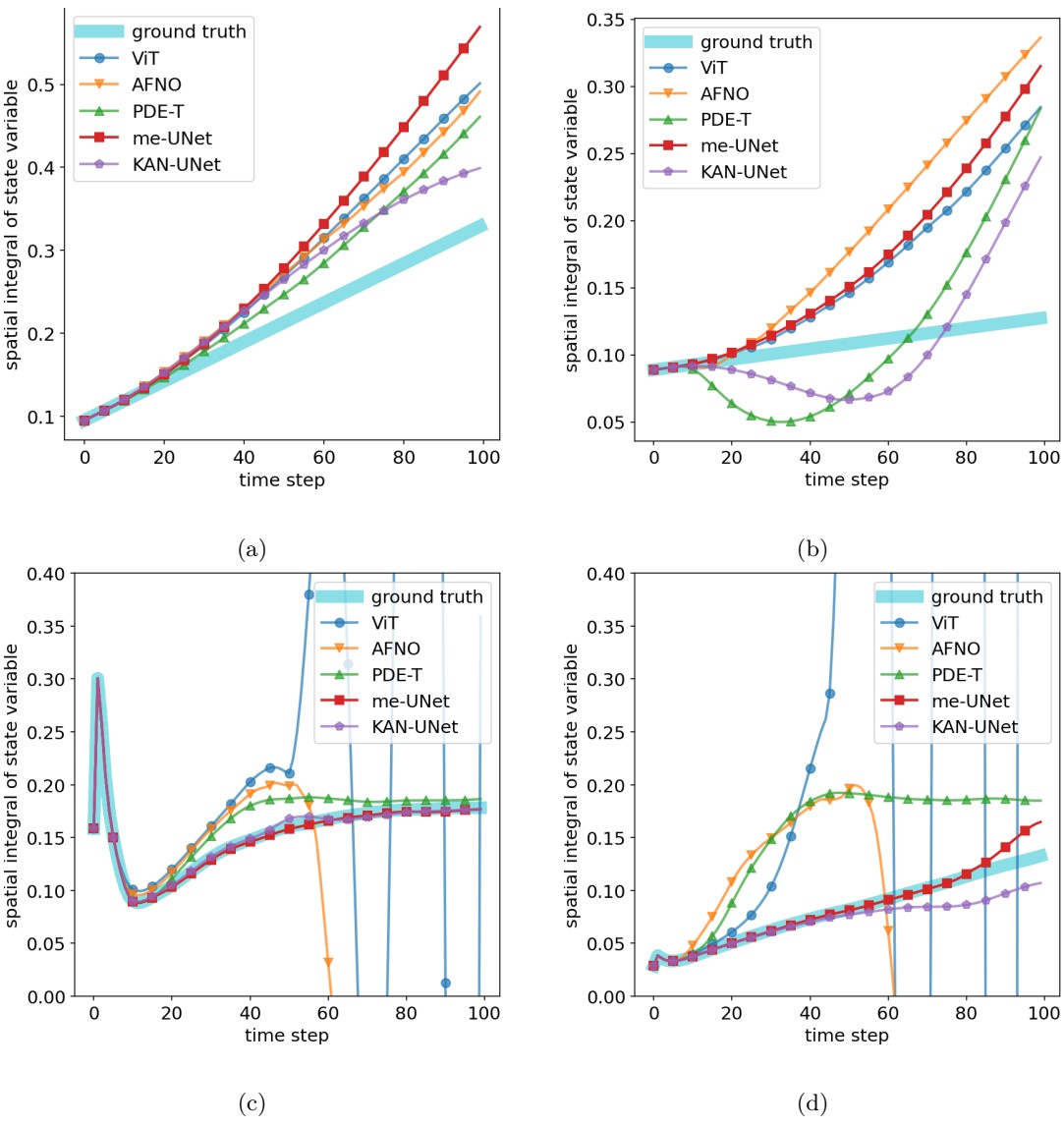

Figure 13: Spatial integrals of the monitored quantities for OOD rollouts on (a) DS-4 with initial few loops, (b) DS-4 with mixed lines and loops, (c) DS-6a with line-like initial conditions, and (d) DS-6a with blob-like initial conditions.

# G    Visualization of prediction performance across architectures

This section contains additional qualitative rollouts and metric curves for all datasets, complementing the examples in section 3. Each figure shows, for a fixed dataset, autoregressive predictions of all architectures over 100 time steps together with per-time-step RMSE and PSD cosine similarity. These plots illustrate in more detail the trends discussed in the main text: me-UNet and, to a lesser extent, KAN-UNet produce stable pattern evolution with low error and high spectral similarity, whereas several of the heavier architectures either smooth out fine-scale structure or develop high-frequency artifacts over long rollouts.

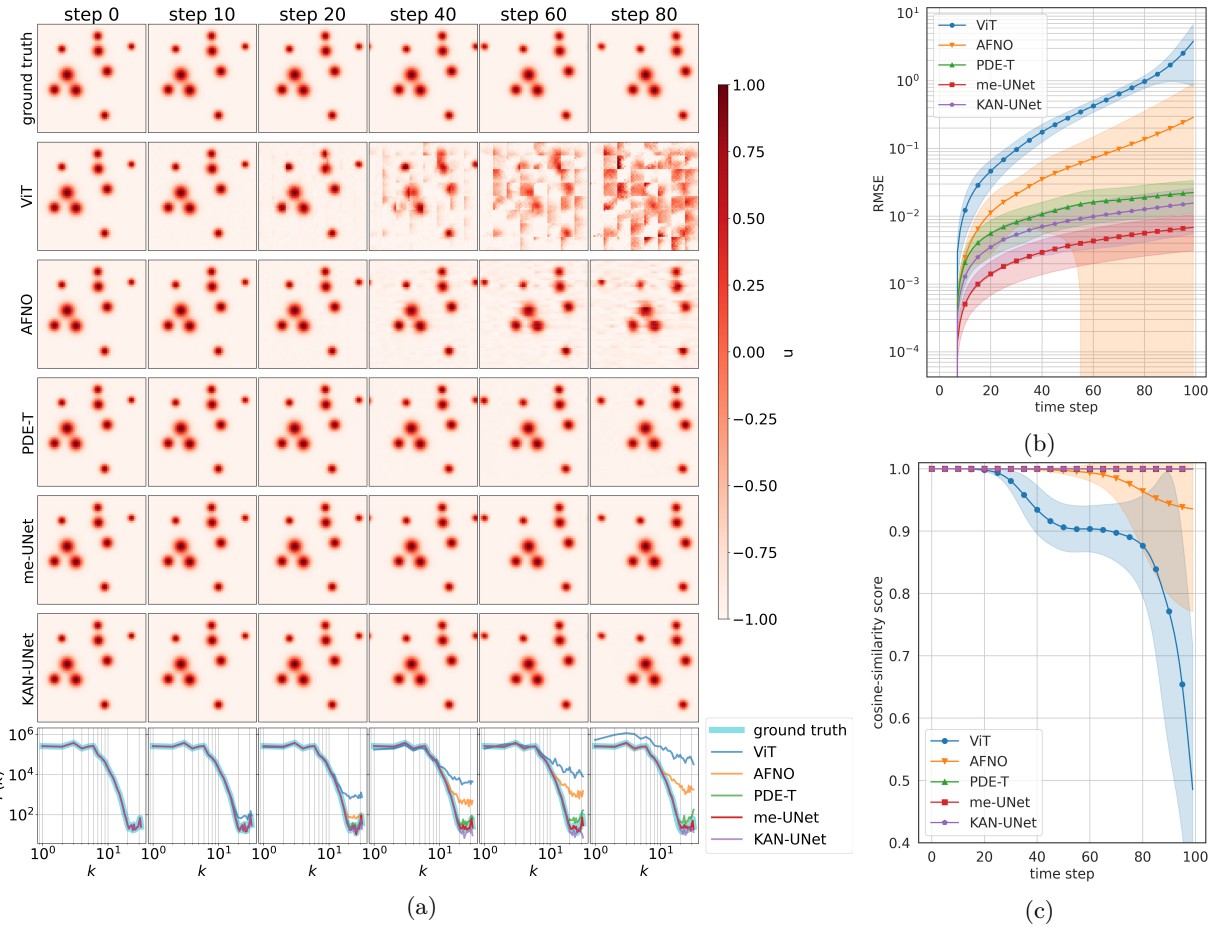

Figure 14: Visualization of the performance of the different neural networks on the DS-1 dataset: (a) autoregressive prediction results; (b) RMSE; and (c) cosine similarity of the PSD curves.

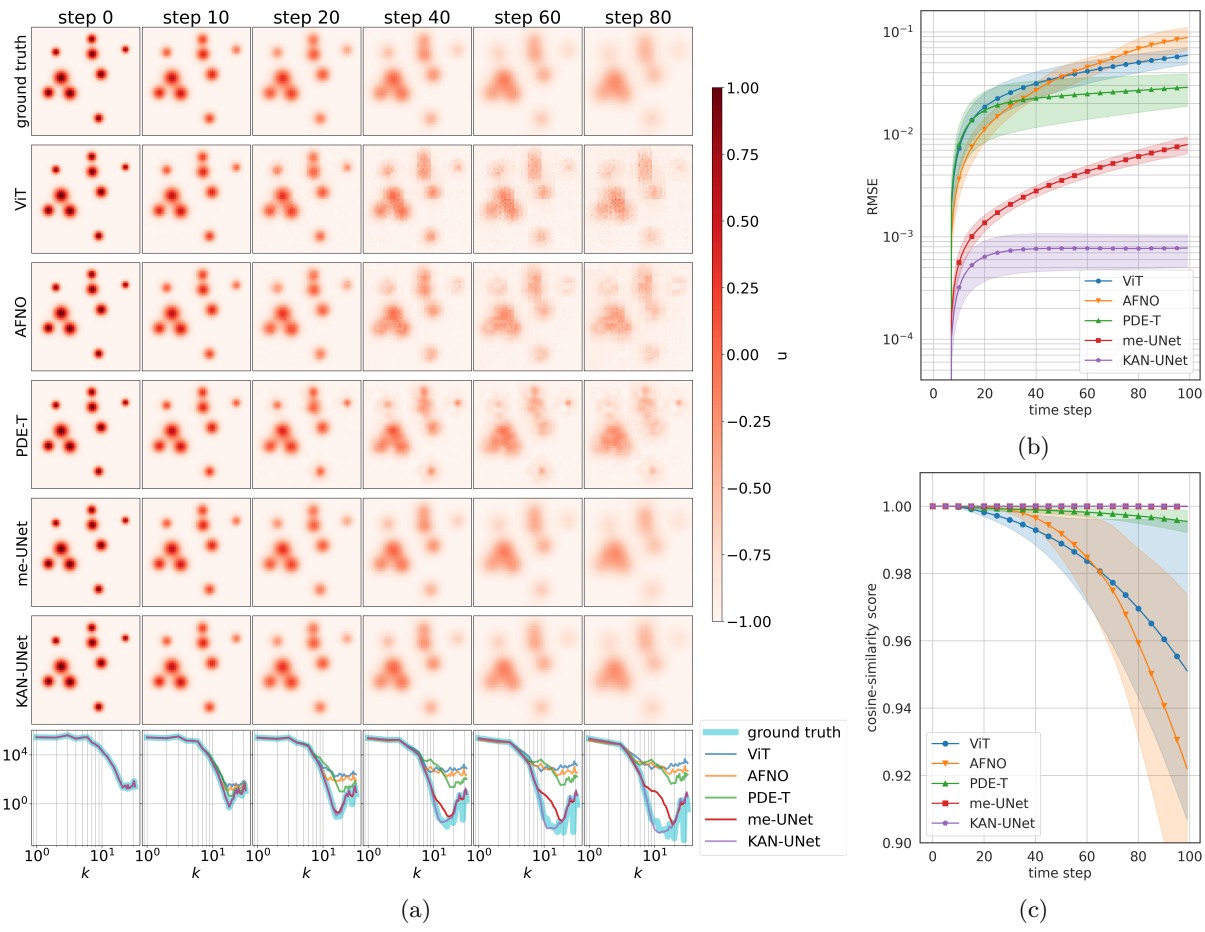

Figure 15: Visualization of the performance of the different neural networks on the DS-2 dataset: (a) autoregressive prediction results; (b) RMSE; and (c) cosine similarity of the PSD curves.

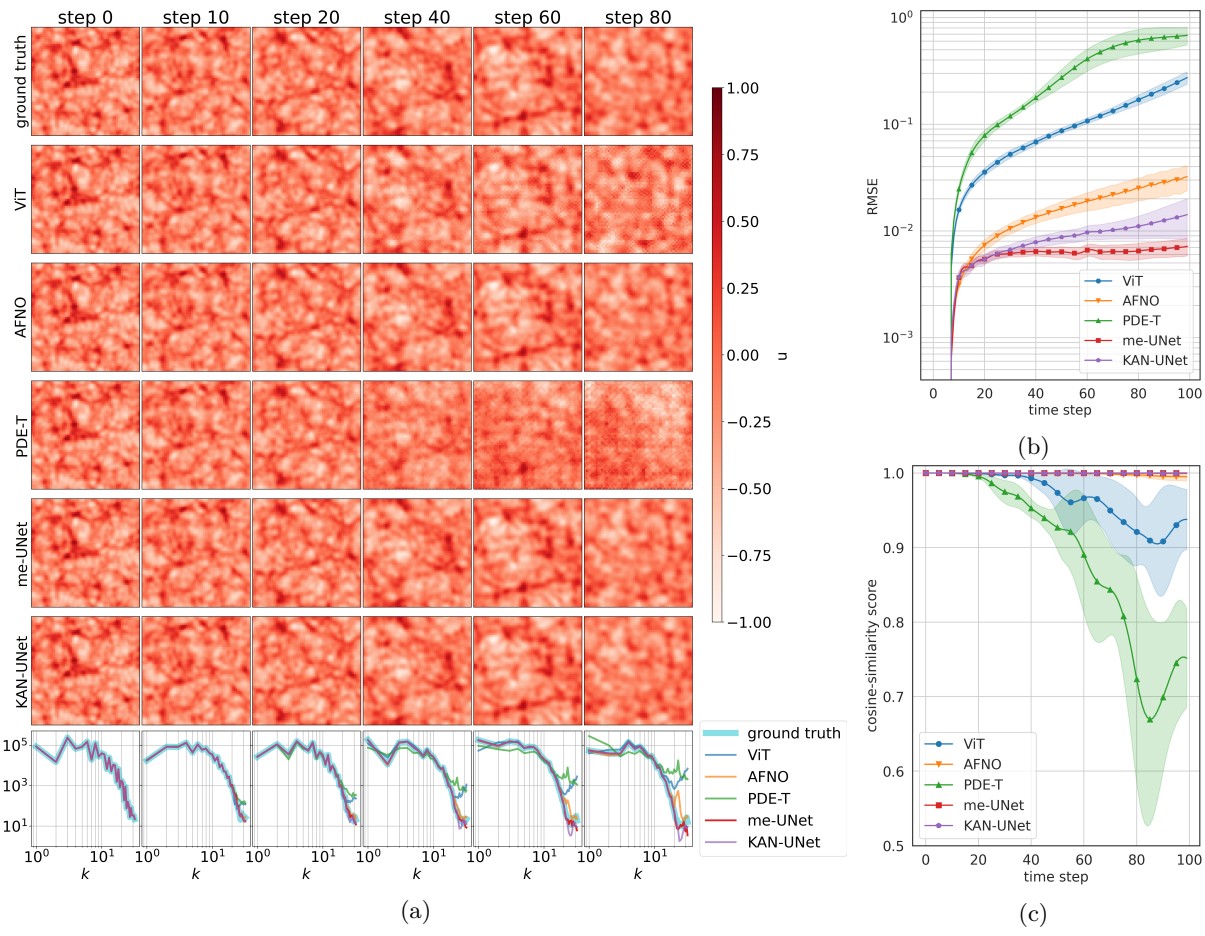

Figure 16: Visualization of the performance of the different neural networks on the DS-3a dataset: (a) autoregressive prediction results; (b) RMSE; and (c) cosine similarity of the PSD curves.

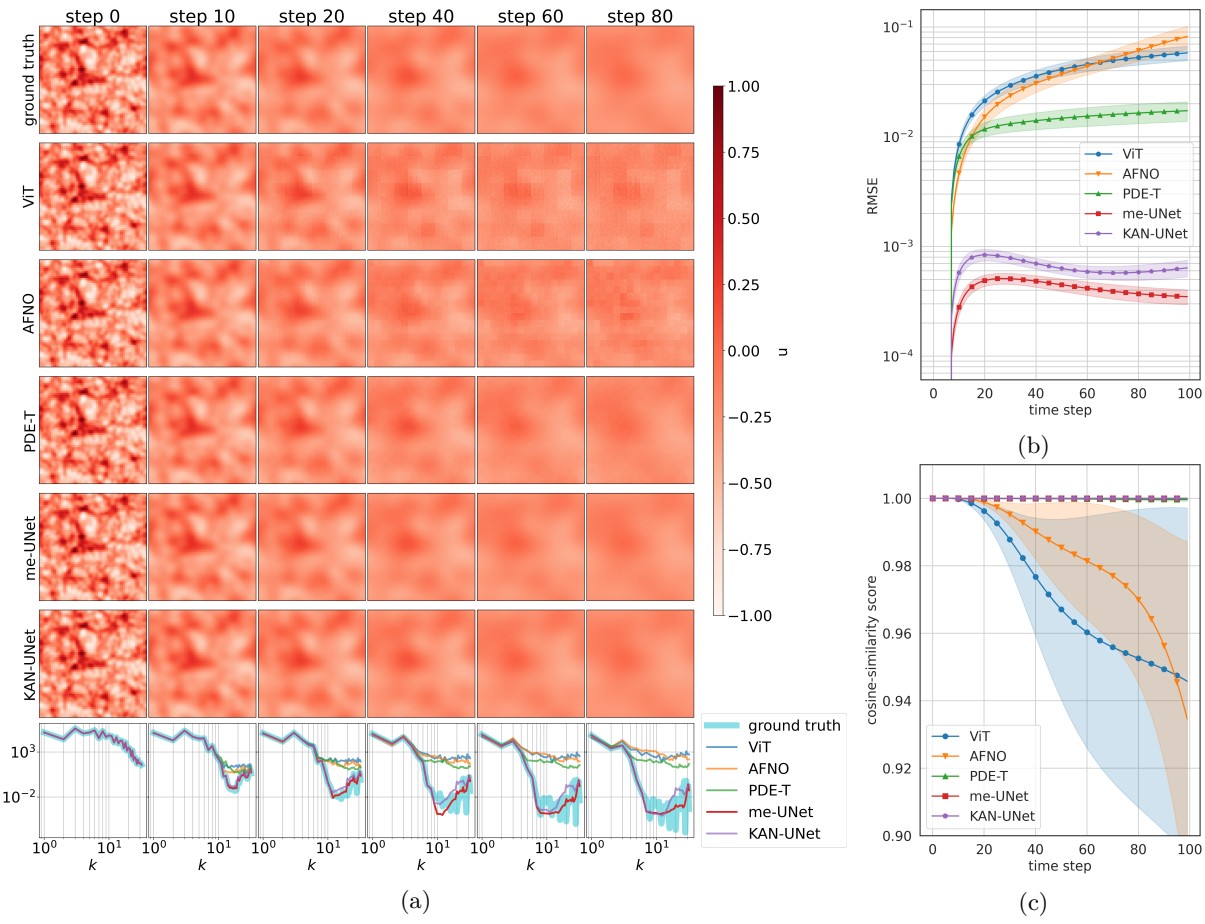

Figure 17: Visualization of the performance of the different neural networks on the DS-3b dataset: (a) autoregressive prediction results; (b) RMSE; and (c) cosine similarity of the PSD curves.

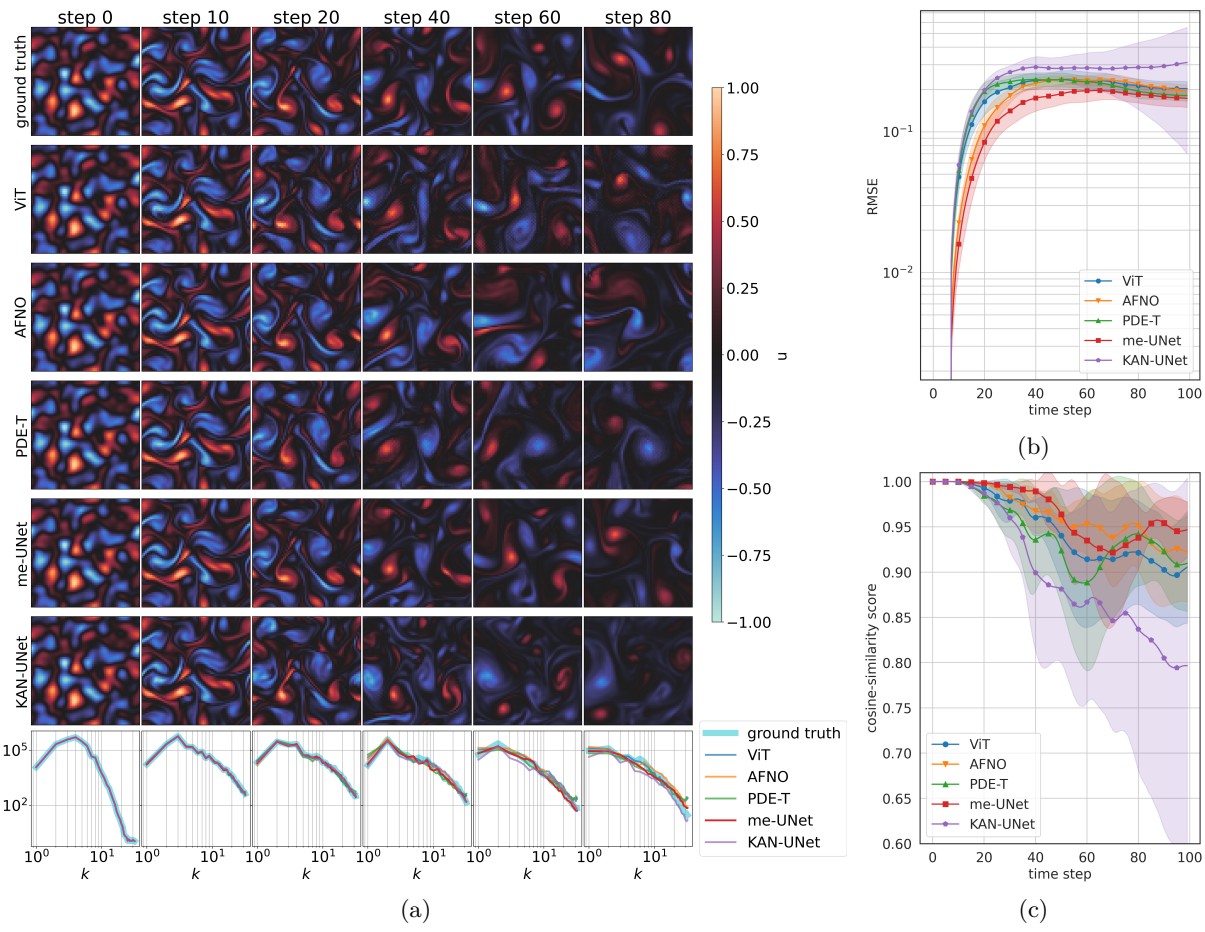

Figure 18: Visualization of the performance of the different neural networks on the DS-5 dataset: (a) autoregressive prediction results; (b) RMSE; and (c) cosine similarity of the PSD curves.

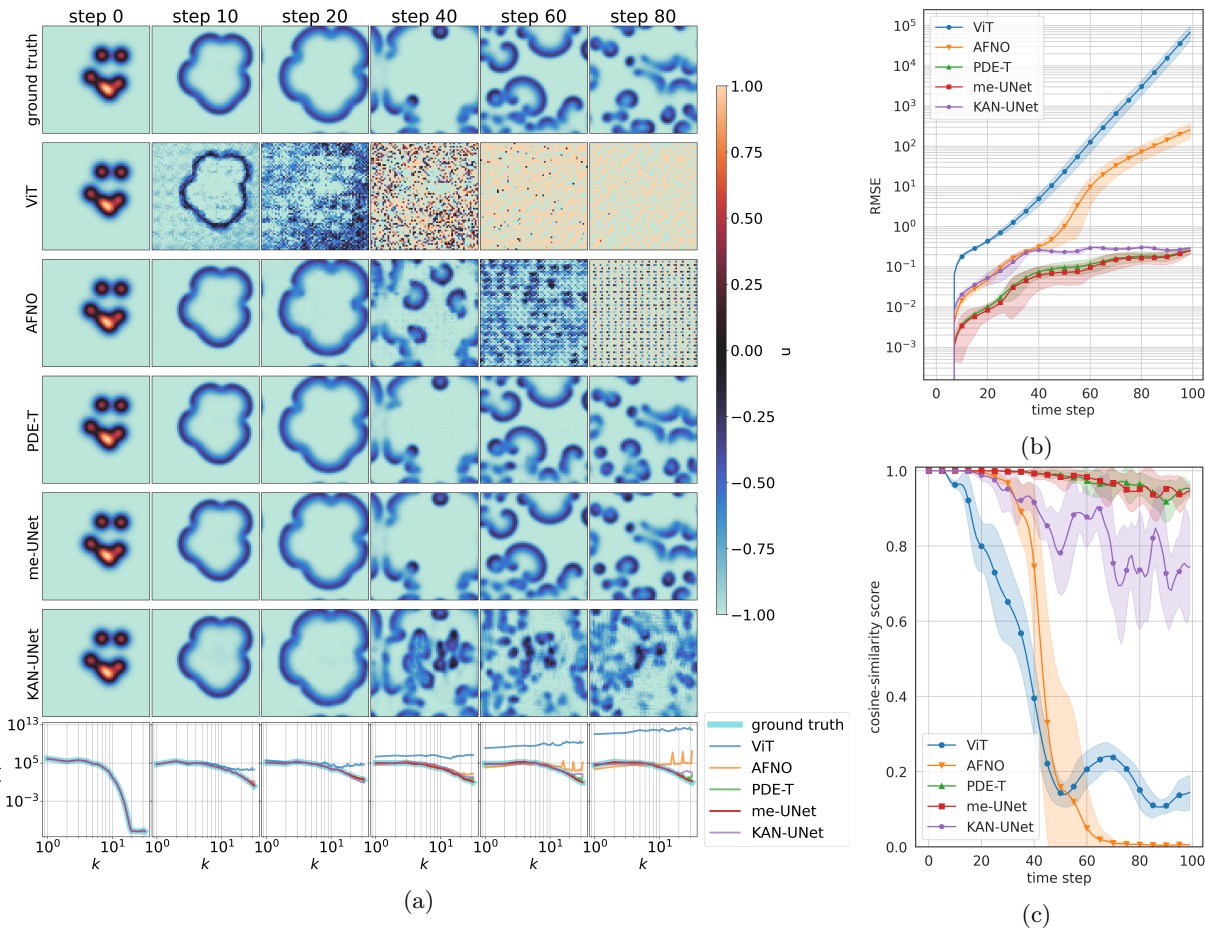

Figure 19: Visualization of the performance of the different neural networks on the DS-6b dataset: (a) autoregressive prediction results; (b) RMSE; and (c) cosine similarity of the PSD curves.

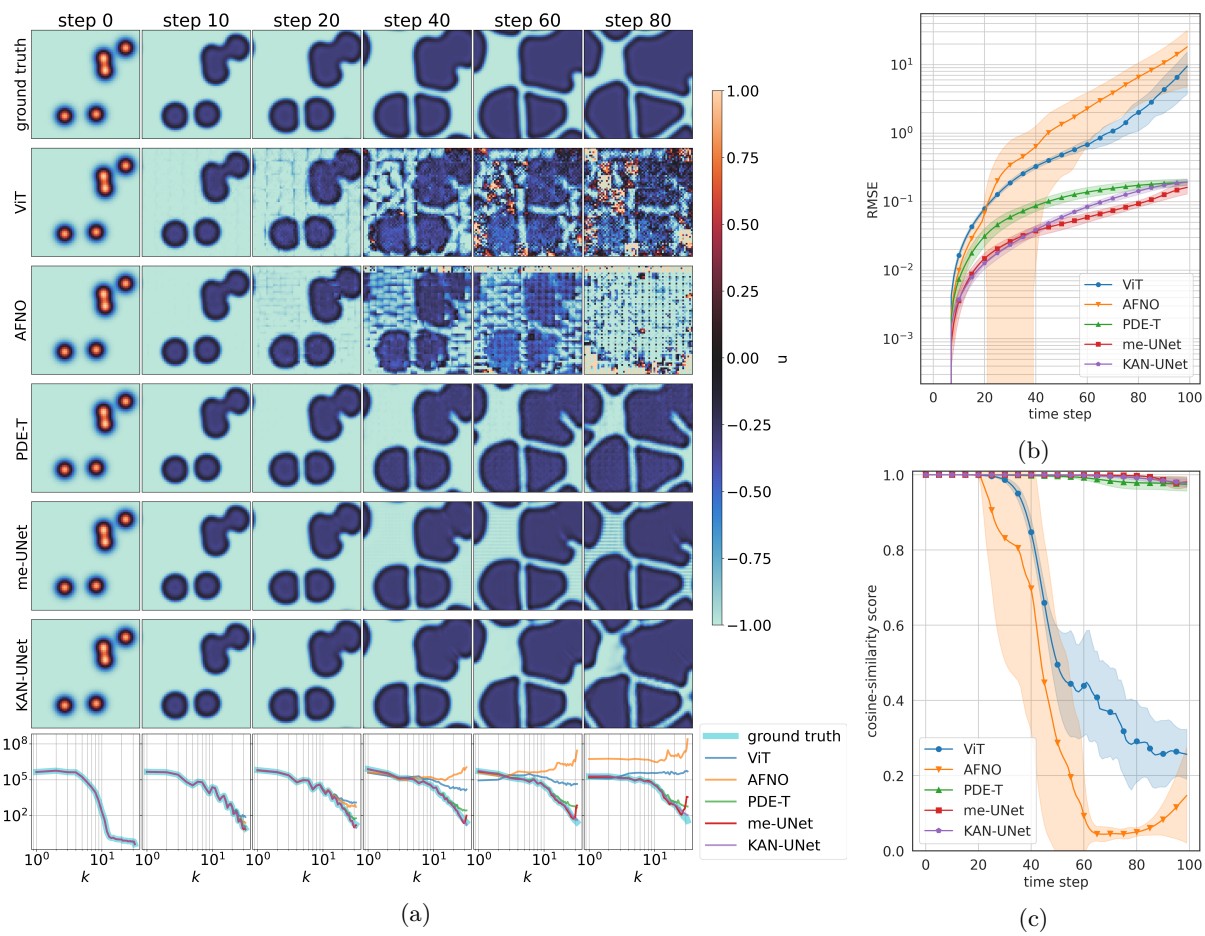

Figure 20: Visualization of the performance of the different neural networks on the DS-6c dataset: (a) autoregressive prediction results; (b) RMSE; and (c) cosine similarity of the PSD curves.

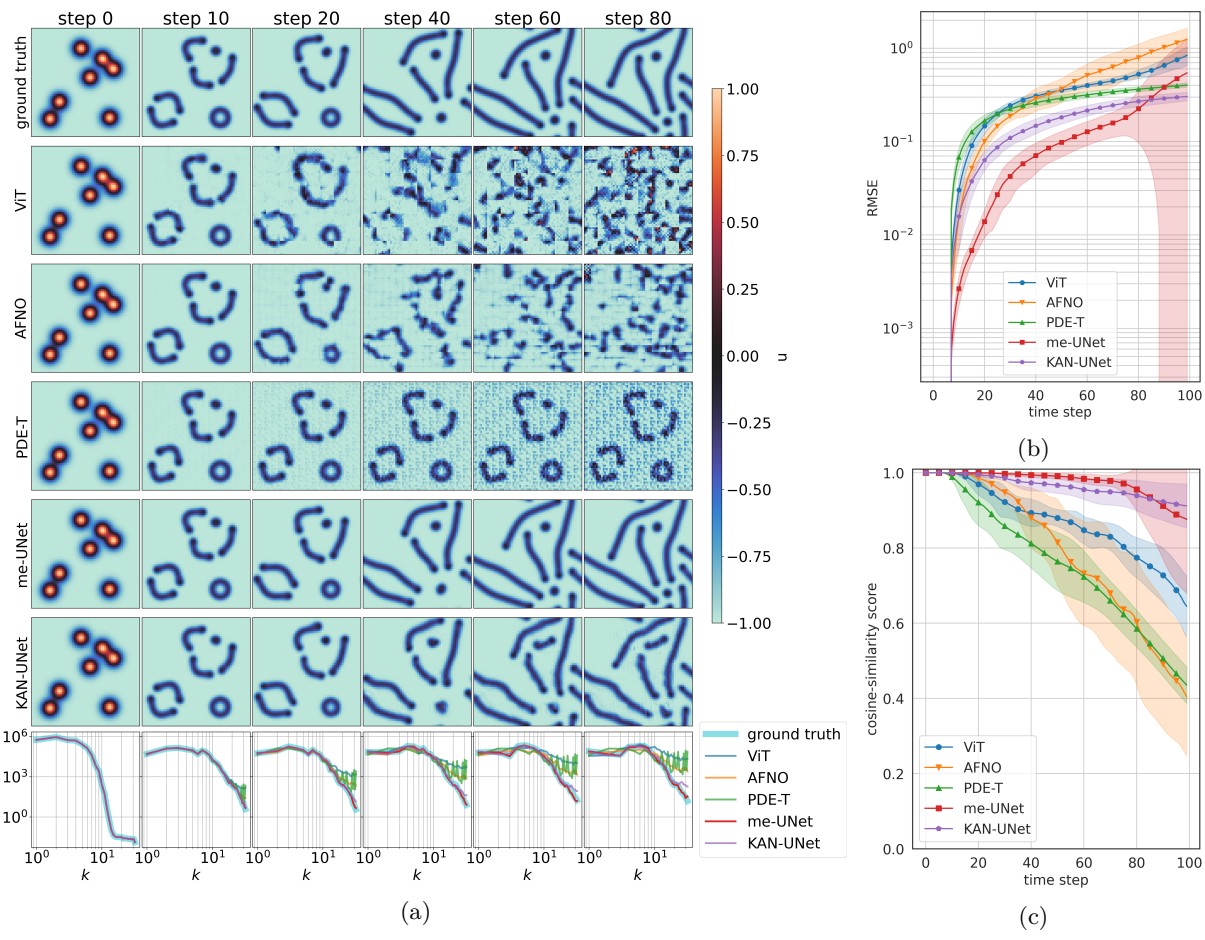

Figure 21: Visualization of the performance of the different neural networks on the DS-6d dataset: (a) autoregressive prediction results; (b) RMSE; and (c) cosine similarity of the PSD curves.

# H   Visualization of RMSE and cosine similarity for OOD predictions

As expected from the main results in section 3, me-UNet lies near the bottom of the error curves and near the top of the cosine-similarity curves in both panels, while ViT and AFNO show much larger degradation under OOD initial conditions. Fig. 22 shows these measures for all models and all simulation time steps.

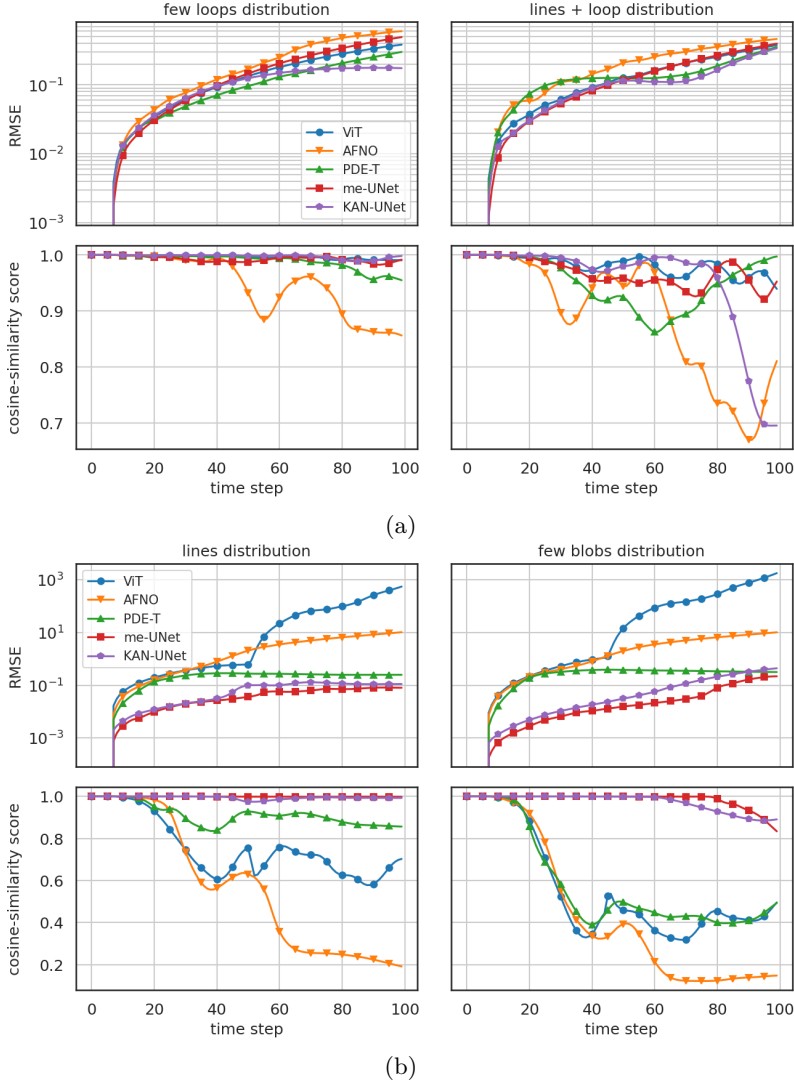

Figure 22: OOD performance of the different neural network architectures on (a) the CDD model and (b) the Gray–Scott model (DS-6a), in terms of RMSE and PSD cosine similarity.

# I  Visualization of me-UNet behavior during training (Grad-CAM)

In what follows we provide full Grad-CAM visualizations for all datasets, complementing the qualitative discussion in section 3. For each dataset, the top panel shows Grad-CAM maps for all encoder/decoder blocks over training epochs, and the bottom panel shows the average Grad-CAM value per block. These plots illustrate how attention concentrates on physically relevant regions over the course of training and how different datasets activate different parts of the U-Net. Note that during each training epoch *all* simulation steps are simultaneously considered by the model; therefore, the early epochs *cannot* be correlated with the early simulation time steps.

**DS-1 (Fig. 23)**   Across the 1000 training epochs, block layers d0, d1, u1, u2, u3, and u4 remain active, whereas layers d2, d3, d4 and u0 become inactive at epochs 440, 400, 360 and 360, respectively. This dataset exhibits purely advective behavior in the x-direction. The Grad-CAM visualizations reveal a pronounced and consistent vertical streak structure, most clearly visible in u1 and u2, corresponding to the transport of objects along the x-axis. If, instead, we would have had transport along the y-axis, horizontal streaks would occur (not shown here).

**DS-2 (Fig. 24)**   Throughout 1000 training epochs, layers d0, d1, u3, and u4 remain active, while layers d2, d3, d4, u0, u1, and u2 deactivate at epochs 360, 280, 200, 200, 280 and 360, respectively. This dataset exhibits purely diffusive behavior. However, diffusion is a type of transport process where, e.g., a concentration is moved along both directions. Initially, this transport phenomenon is particularly pronounced, producing characteristic cross-shaped patterns in u1 and u2. As training progresses, the model transitions to alternative feature representations, reflecting that diffusive transport is slowing down such that only minimal state changes occur.

**DS-3a (Fig. 25)**   During the 1000-epoch training, layers d0, d1, u3, and u4 remain active, while layers d2, d3, d4, u0, u1, and u2 deactivate at epochs 520, 440, 400, 400, 440 and 520, respectively. This dataset contains bidirectional transport. Consistent with this, the Grad-CAM maps display cross-pattern structures across multiple scales of the network.

**DS-3b (Fig. 26)**   Across the 1000 epochs, only layers d0 and u4 remain consistently active. Layers d1–d4 and u0–u3 deactivate at epochs 440, 360, 320, 240, 240, 320, 360 and 440, respectively. As in DS-2, the presence of transport in both directions leads to cross-pattern activations in u1 and u2.

**DS-4 (Fig. 27)**   Throughout training, layers d0 and u4 stay active, while layers d1–d4 and u0–u3 deactivate at epochs 880, 480, 400, 280, 280, 400, 480 and 840, respectively. Here, bidirectional transport is intrinsic to the governing equations, yielding pronounced and persistent cross-patterns in layers u1, u2, and u3.

**DS-5 (Fig. 28)**   All block layers remain active over the 1000 training epochs. Although the dataset contains complex transport phenomena, the Grad-CAM maps still exhibit cross-pattern structures, particularly in u1 and u2, albeit embedded within more complex feature combinations. From the gradCAM visualization we can clearly see that a number of different physical phenomena are active throughtout the whole time range of the simulation.

**DS-6a (Fig. 29)**   Layers d0–d3 and u2–u4 stay active during training, while d4, u0, and u1 deactivate at epochs 200, 200 and 920, respectively. The dataset contains bidirectional transport, reflected in the maze-like growth dynamics of the simulation. Correspondingly, cross-pattern features appear in u1 and u2.

**DS-6b (Fig. 30)**   All block layers remain active throughout the 1000 epochs. Early in the simulation, expanding blob structures induce transport in both spatial directions, resulting in early-epoch cross-pattern signatures in the Grad-CAM maps before other features become dominant at later stages. As opposed to the previous dataset, the model is still undergoing changes also for later epochs.

**DS-6c (Fig. 32)**   Layers d0, d1, u3, and u4 stay active, while d2, d3, d4, u0, u1, and u2 deactivate at epochs 640, 440, 280, 440 and 640, respectively. Bidirectional transport is again present in this dataset. Cross-patterns appear only locally within the Grad-CAM visualizations, indicating a more spatially heterogeneous transport behavior.

**DS-6d (Fig. 31)**   Layers d0, d1, u2, u3, and u4 remain active, while layers d2–d4 and u0–u1 deactivate at epochs 840, 800, 400, 400, and 800, respectively. After an "incubation period" the dataset contains strong bidirectional transport dynamics, yielding clearly identifiable cross-patterns in the Grad-CAM maps, with layer u2 even converging to a stable cross-pattern representation. The absense of these patterns during the early phase of training suggests that initially, it was beneficial to focus the training on non-transport aspects.

Overall, there are common transport activities in almost all presented datasets, which result in the appearance of streak- or crossed-pattern, depending on which direction this movement is dominated. Their appearance depends on the amount of training data which contains these behaviors. If a large part of the simulation clearly shows transport activities (e.g., DS-1, DS-3a, or DS-4), then in the grad-CAM visualization these patterns appear often and may even converge (Fig. 23). If only a small part of the simulation that the transport activities are clearly observed (e.g., DS-2, DS-3b, DS-6b, or DS-6c), the crossed patterns appear at the beginning or only in some places during the training process, then other patterns dominate.

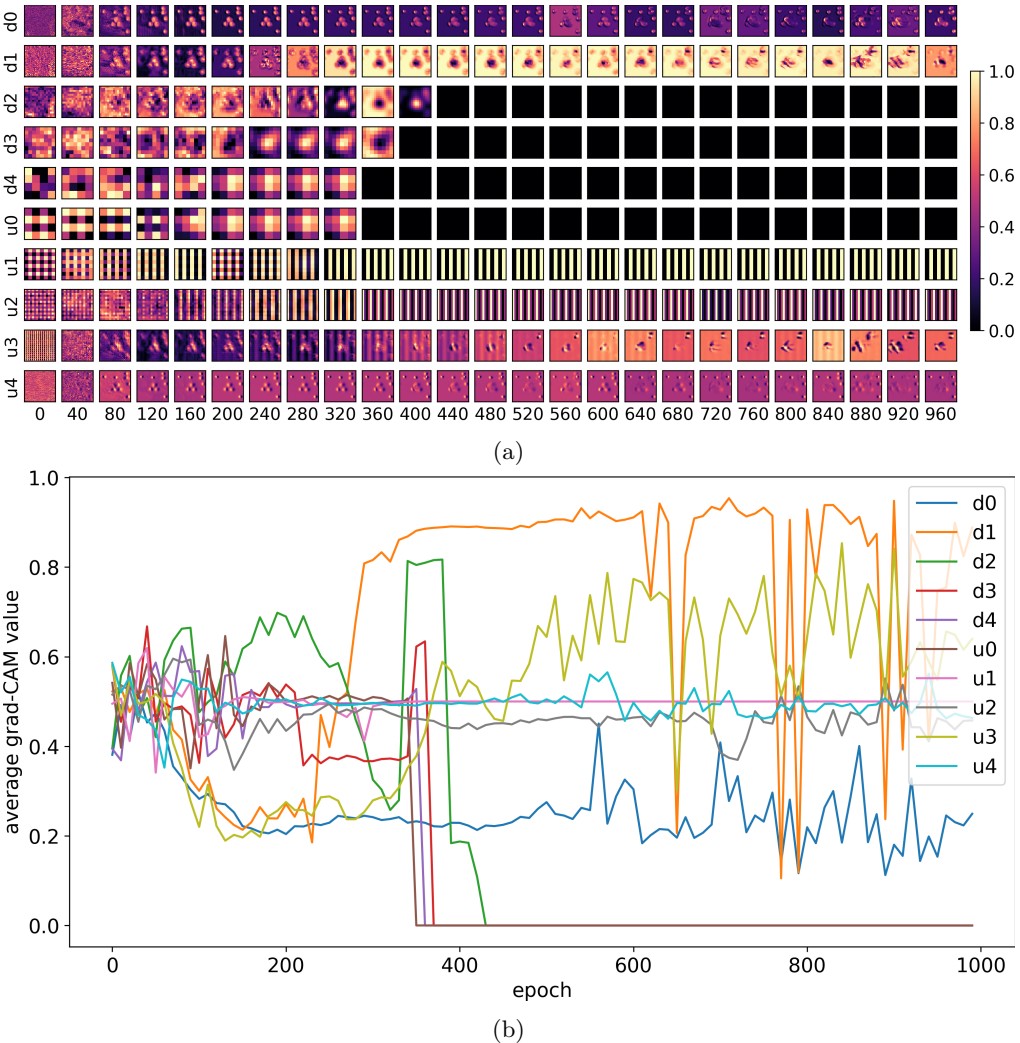

Figure 23: Behavior of me-UNet during training on the DS-1 dataset: (a) Grad-CAM maps over training epochs; (b) average Grad-CAM value per block as a function of epoch.

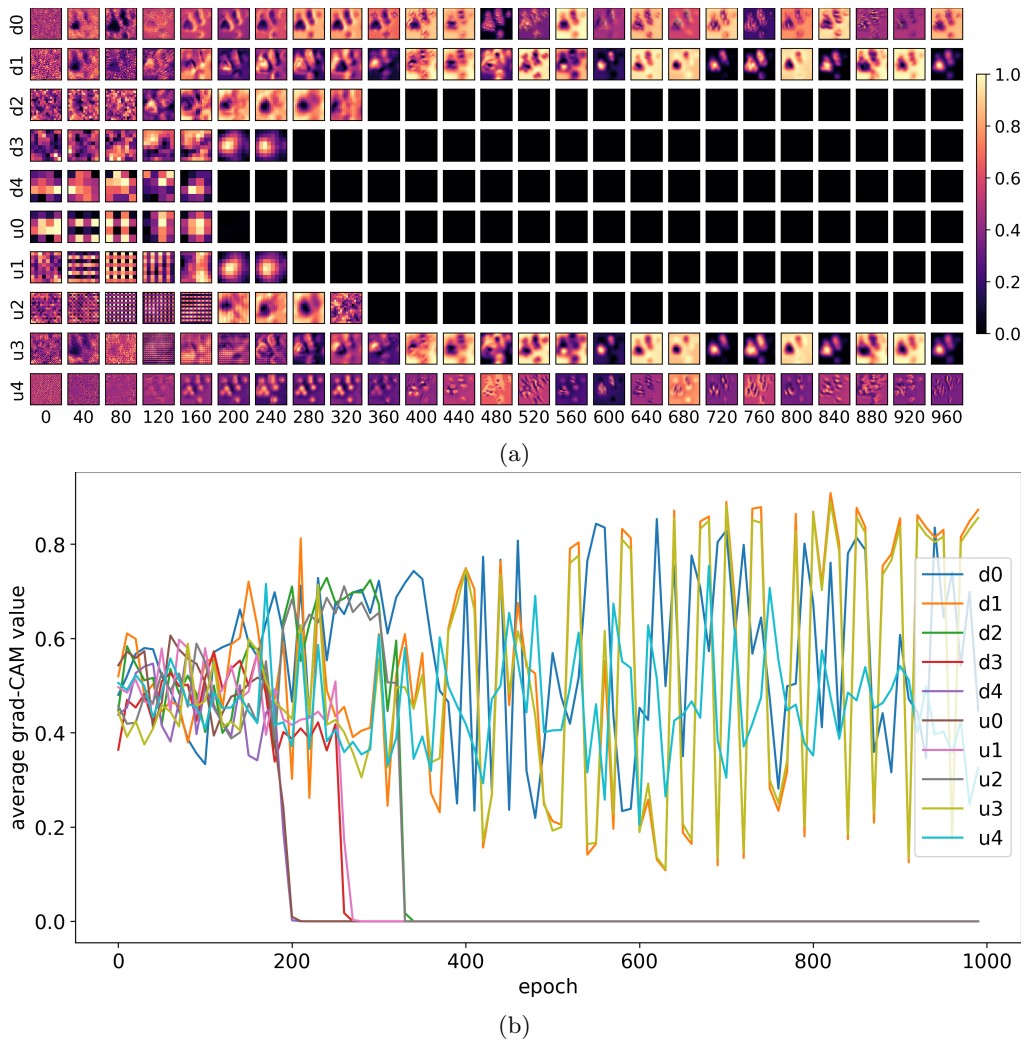

(a)

(b)

Figure 24: Behavior of me-UNet during training on the DS-2 dataset: (a) Grad-CAM maps; (b) average Grad-CAM value per block over epochs.

## J  Ablation: effect of the perceptual loss term

To assess the role of the VGG-16 perceptual loss $\mathcal{L}_{\mathrm{pc}}$ introduced in Section 2.3 (Eq. 8), we conduct a controlled ablation in which me-UNet is retrained from scratch with the loss

$$\mathcal{L}_{\mathrm{abl}}(\hat{y}, y) = \mathcal{L}_{\mathrm{MSE}}(\hat{y}, y), \tag{23}$$

i.e. with $\lambda_{\mathrm{pc}} = 0$, while keeping all other settings (optimizer, learning rate, weight decay, batch size, training-budget of 1000 epochs, random seed, data splits, and per-simulation normalization) identical to the main protocol of Section 2.4. We run the ablation on representative datasets spanning the difficulty spectrum.

On the simpler datasets (linear advection DS-1 and linear diffusion DS-2), the with- and without-$\mathcal{L}_{\mathrm{pc}}$ rollouts are practically indistinguishable: average RMSE and PSD cosine similarity differ by less than the run-to-run variability, and qualitative snapshots show no visible difference at any time step. In this regime the perceptual term is therefore essentially neutral.

The picture changes on geometrically structured datasets. Figure 33 shows the comparison on DS-6c, the Gray–Scott worm regime, which is the most geometrically demanding of our ten datasets in terms of preserving the topology and curvature of extended one-dimensional structures over long rollouts. The top three

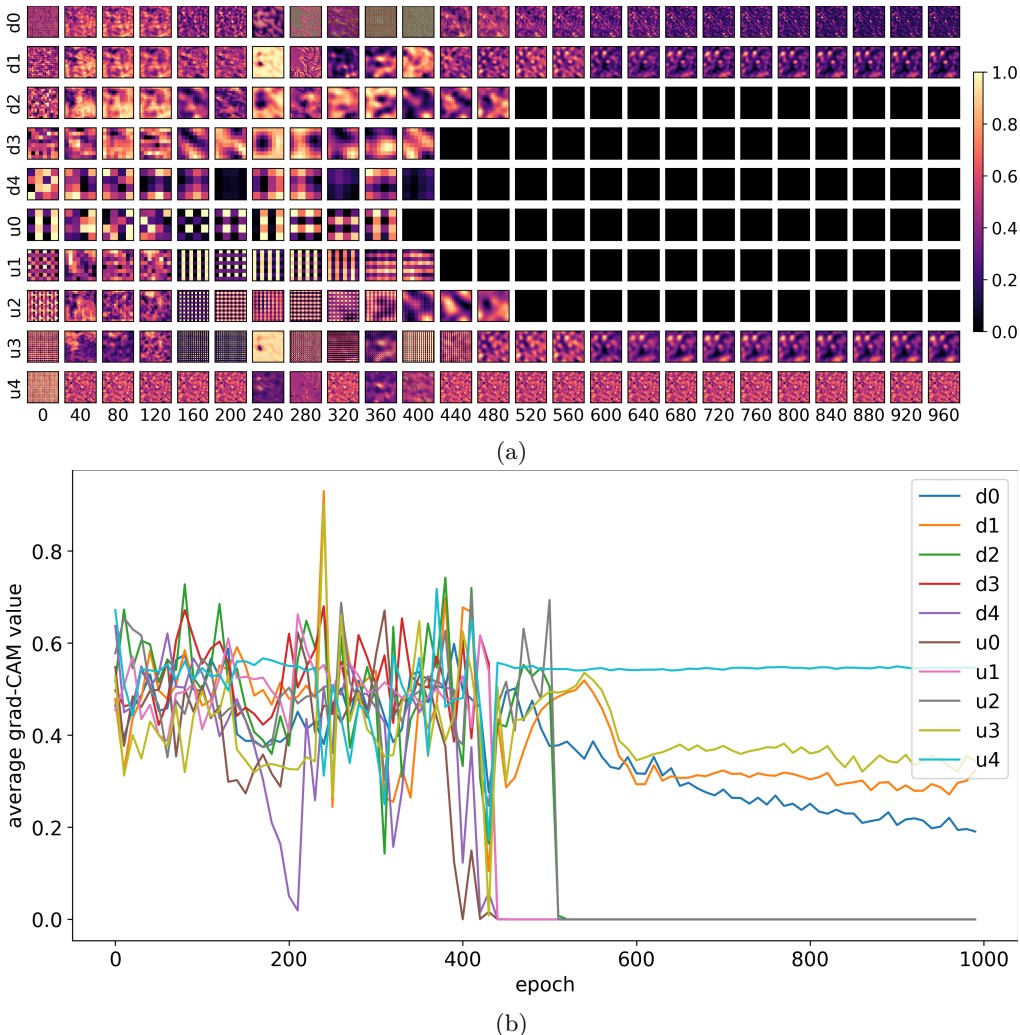

Figure 25: Behavior of me-UNet during training on the DS-3a dataset: (a) Grad-CAM maps; (b) average Grad-CAM value per block over epochs.

rows show the ground-truth rollout, the with-$\mathcal{L}_{\mathrm{pc}}$ prediction, and the without-$\mathcal{L}_{\mathrm{pc}}$ prediction. The bottom two rows show the corresponding pixelwise absolute-error maps.

Two observations are worth highlighting:

1. The without-$\mathcal{L}_{\mathrm{pc}}$ model still places the worms in roughly the correct positions and reproduces their average pixel intensities, but it fails to preserve their *shape*: by step 80, kinks and curved segments present in the ground truth are flattened into straighter pieces, and the worm bodies fragment in places. The with-$\mathcal{L}_{\mathrm{pc}}$ model preserves the curvature and connectivity of the same structures.

2. The absolute-error maps make this geometric failure mode quantitatively visible. The errors of the without-$\mathcal{L}_{\mathrm{pc}}$ model are not spatially diffuse but concentrate along the worm bodies themselves — exactly where geometric coherence is required — whereas the with-$\mathcal{L}_{\mathrm{pc}}$ errors are visibly weaker and more uniformly distributed.

We interpret this as evidence that, in this setting, the early-layer VGG features (`relu2_2`) act as a regularizer for the geometric integrity of extended objects under repeated autoregressive updates. This complements — rather than duplicates — the pointwise MSE term, which is local by construction and provides little

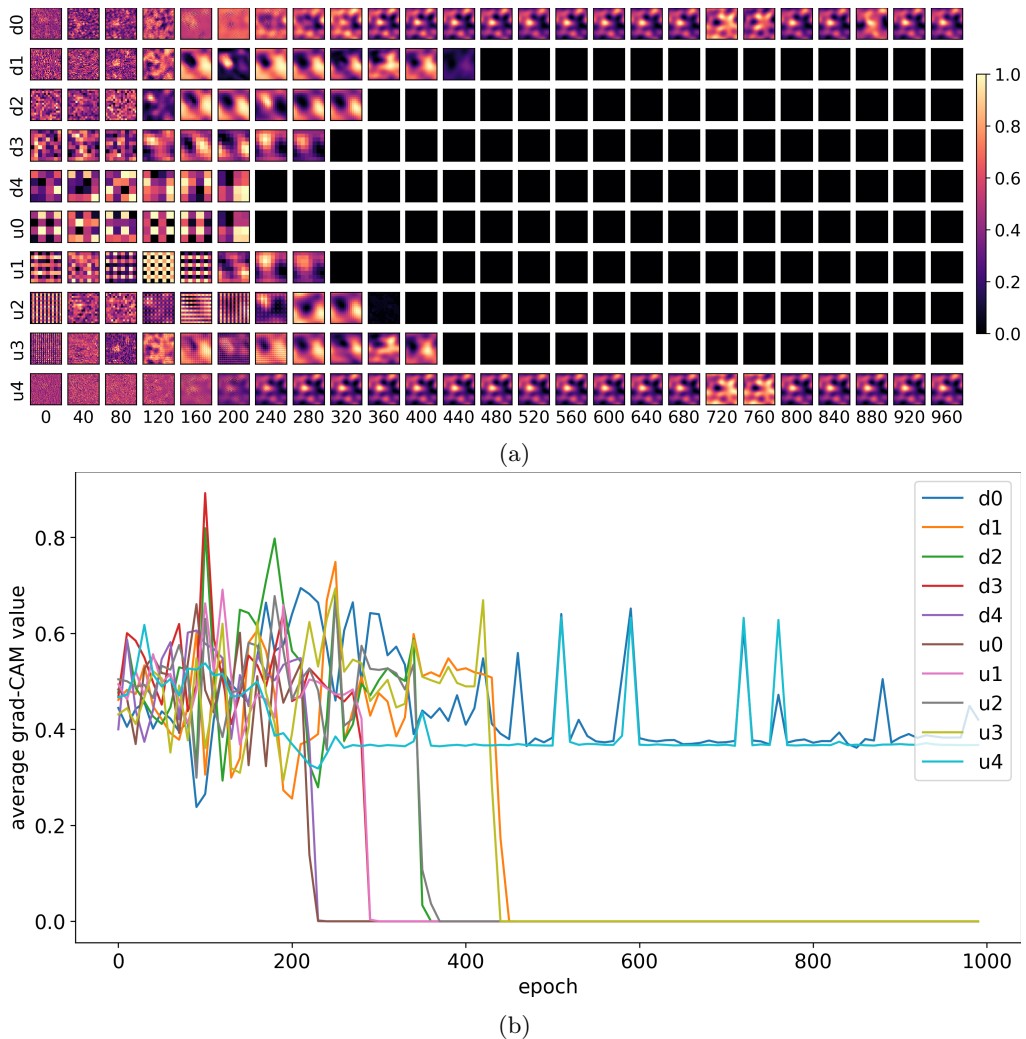

(a)

(b)

Figure 26: Behavior of me-UNet during training on the DS-3b dataset: (a) Grad-CAM maps; (b) average Grad-CAM value per block over epochs.

signal about the coherence of extended geometric structures. The role of $\mathcal{L}_{\mathrm{pc}}$ in our setup is therefore not to make predictions *look better* (a "visual appeal" effect commonly associated with perceptual losses in image generation), but to encourage shape-coherent rollouts on datasets where simple pointwise losses underconstrain extended structures.

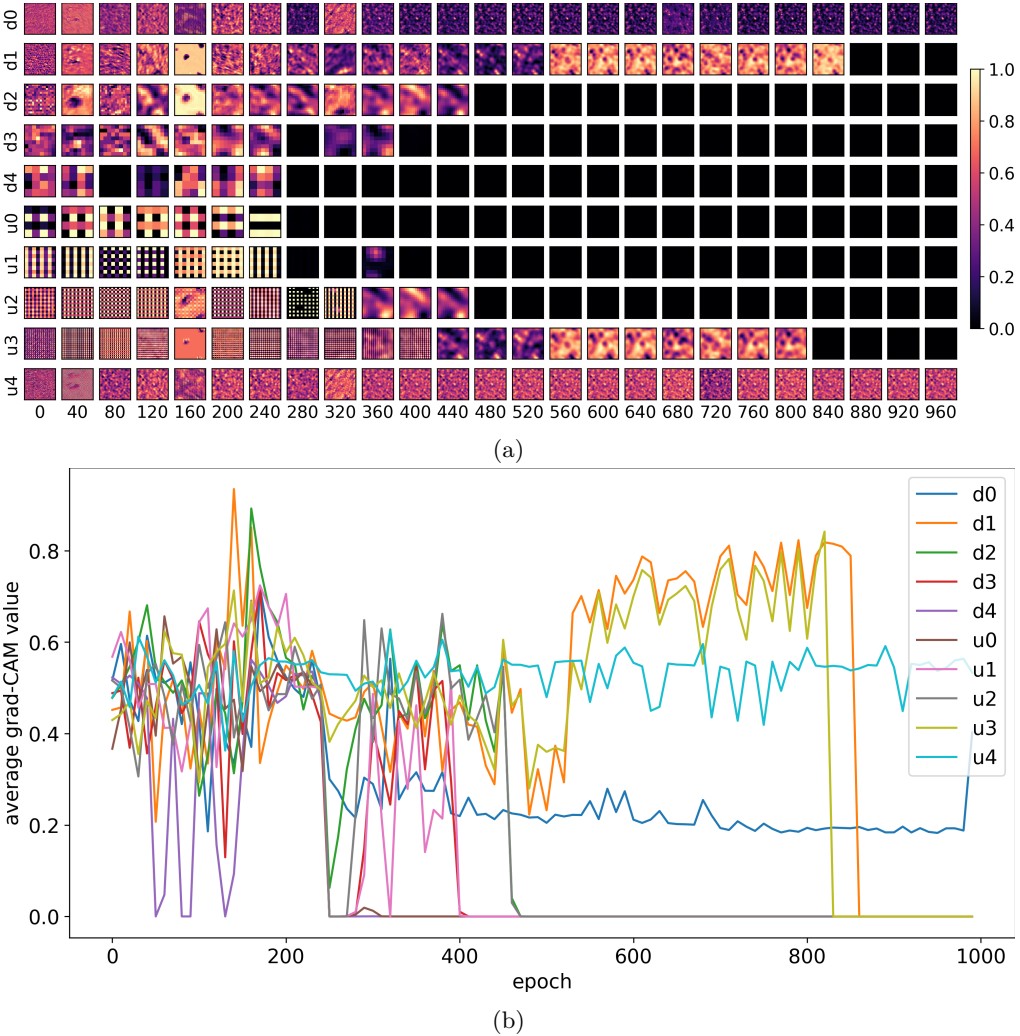

(a)

(b)

Figure 27: Behavior of me-UNet during training on the DS-4 dataset: (a) Grad-CAM maps; (b) average Grad-CAM value per block over epochs.

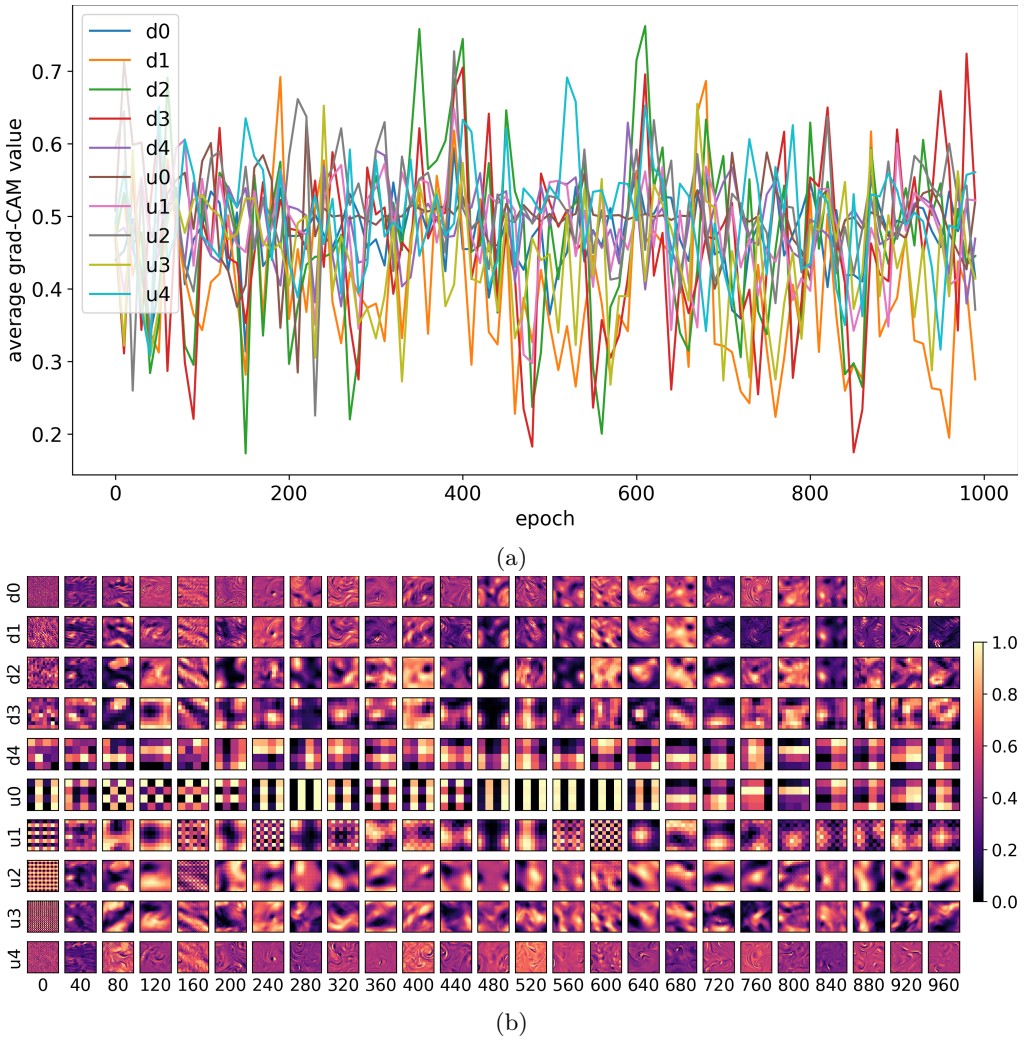

(a)

(b)

Figure 28: Behavior of me-UNet during training on the DS-5 dataset: (a) Grad-CAM maps; (b) average Grad-CAM value per block over epochs.

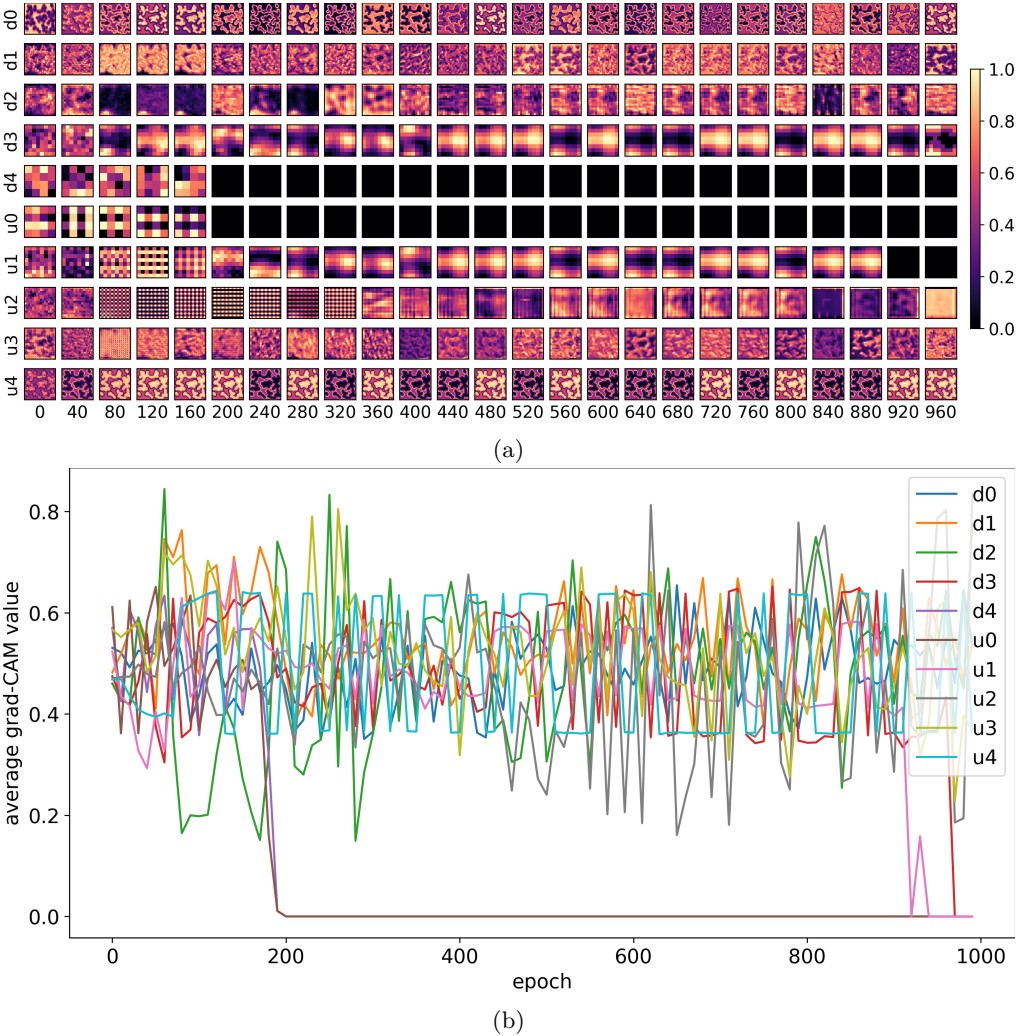

(a)

(b)

Figure 29: Behavior of me-UNet during training on the DS-6a dataset: (a) Grad-CAM maps; (b) average Grad-CAM value per block over epochs.

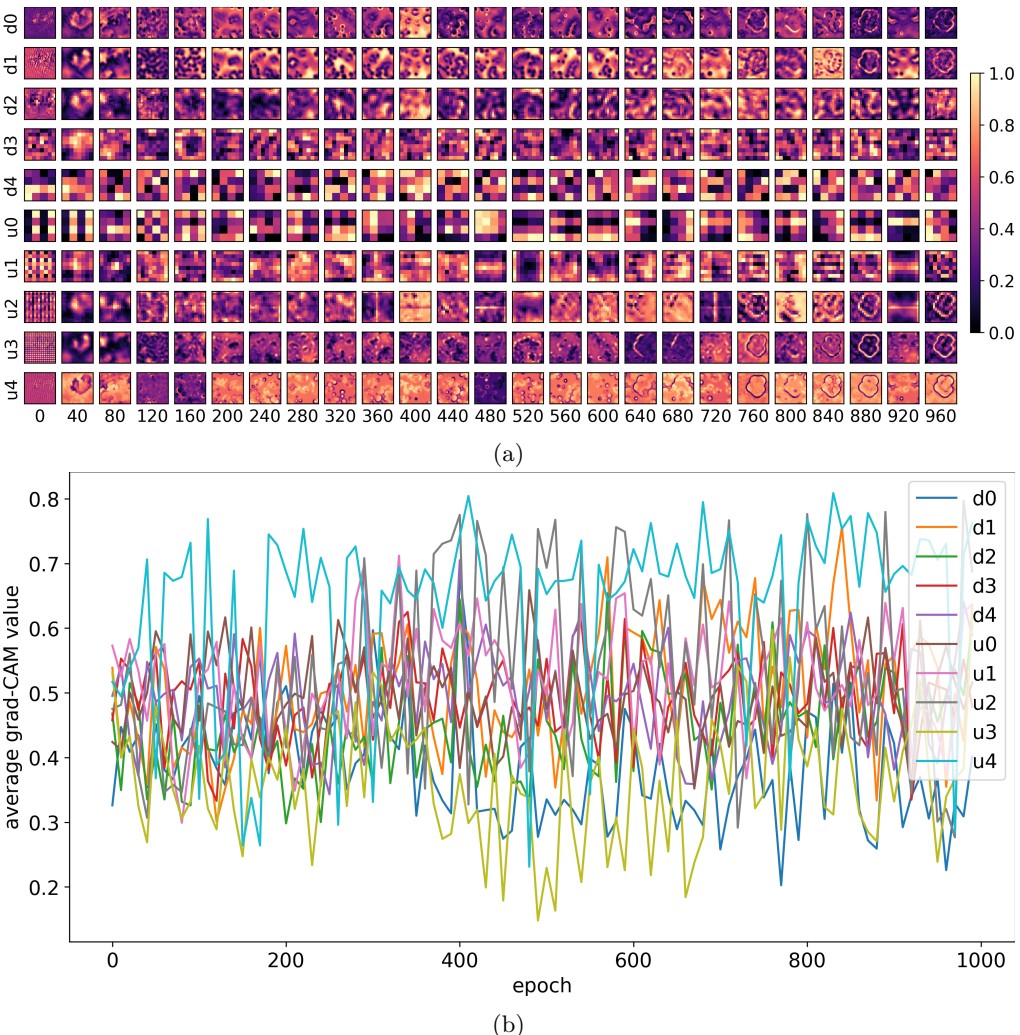

(a)

(b)

Figure 30: Behavior of me-UNet during training on the DS-6b dataset: (a) Grad-CAM maps; (b) average Grad-CAM value per block over epochs.

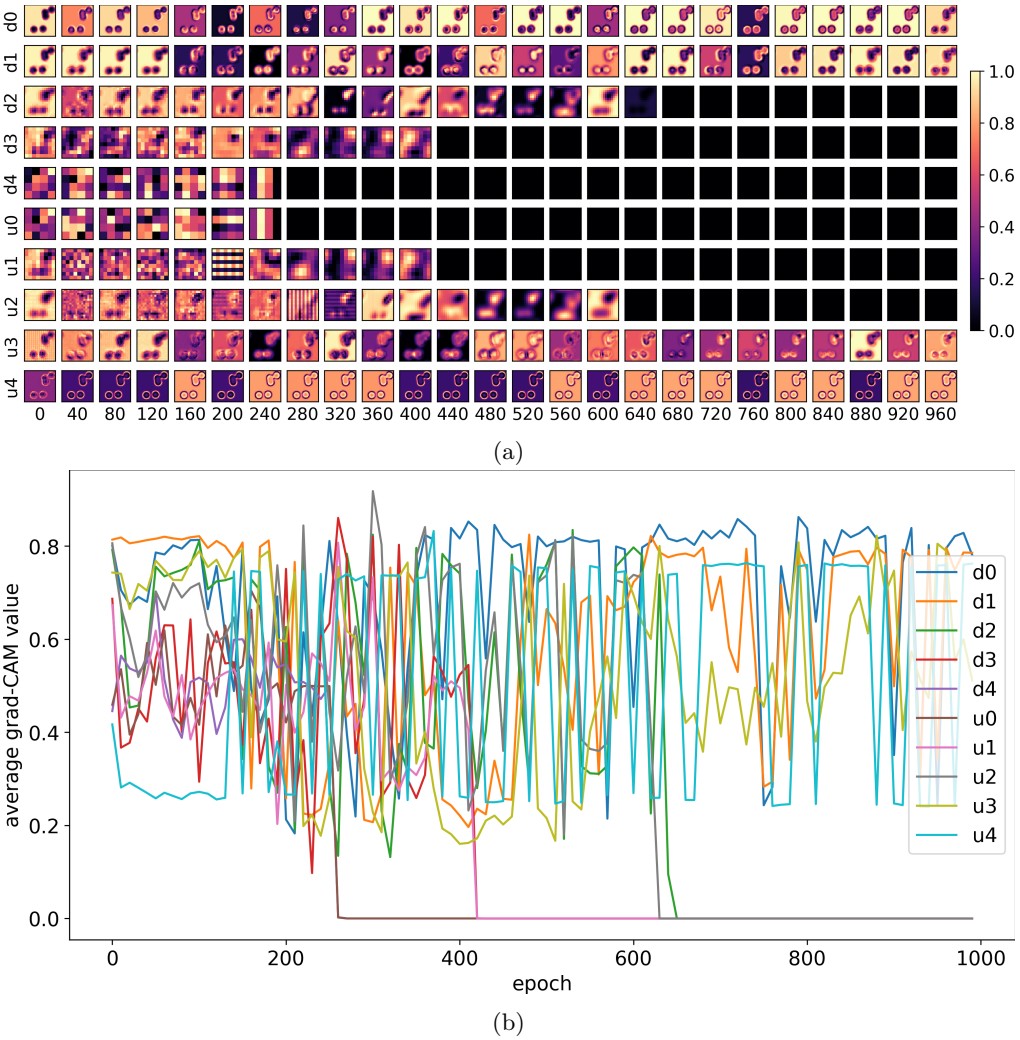

(a)

(b)

Figure 31: Behavior of me-UNet during training on the DS-6c dataset: (a) Grad-CAM maps; (b) average Grad-CAM value per block over epochs.

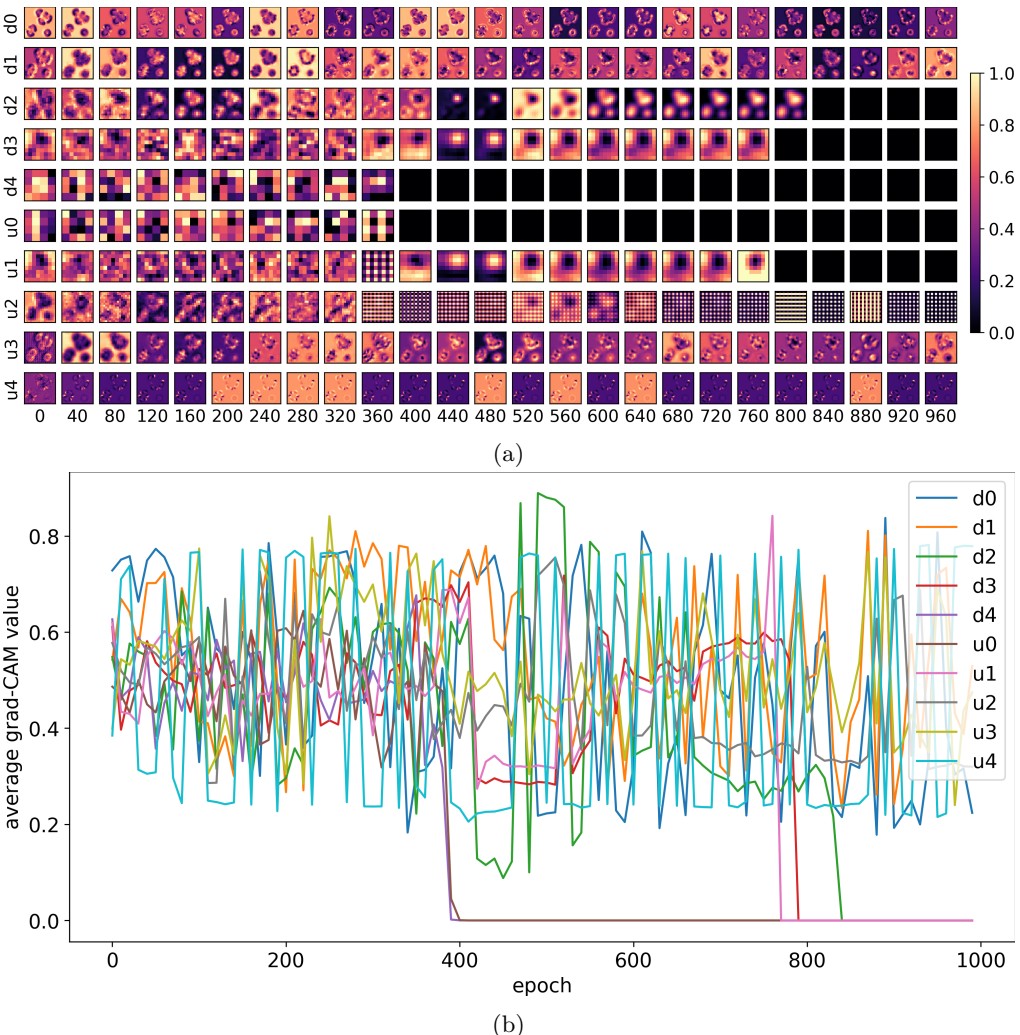

(a)

(b)

Figure 32: Behavior of me-UNet during training on the DS-6d dataset: (a) Grad-CAM maps; (b) average Grad-CAM value per block over epochs.

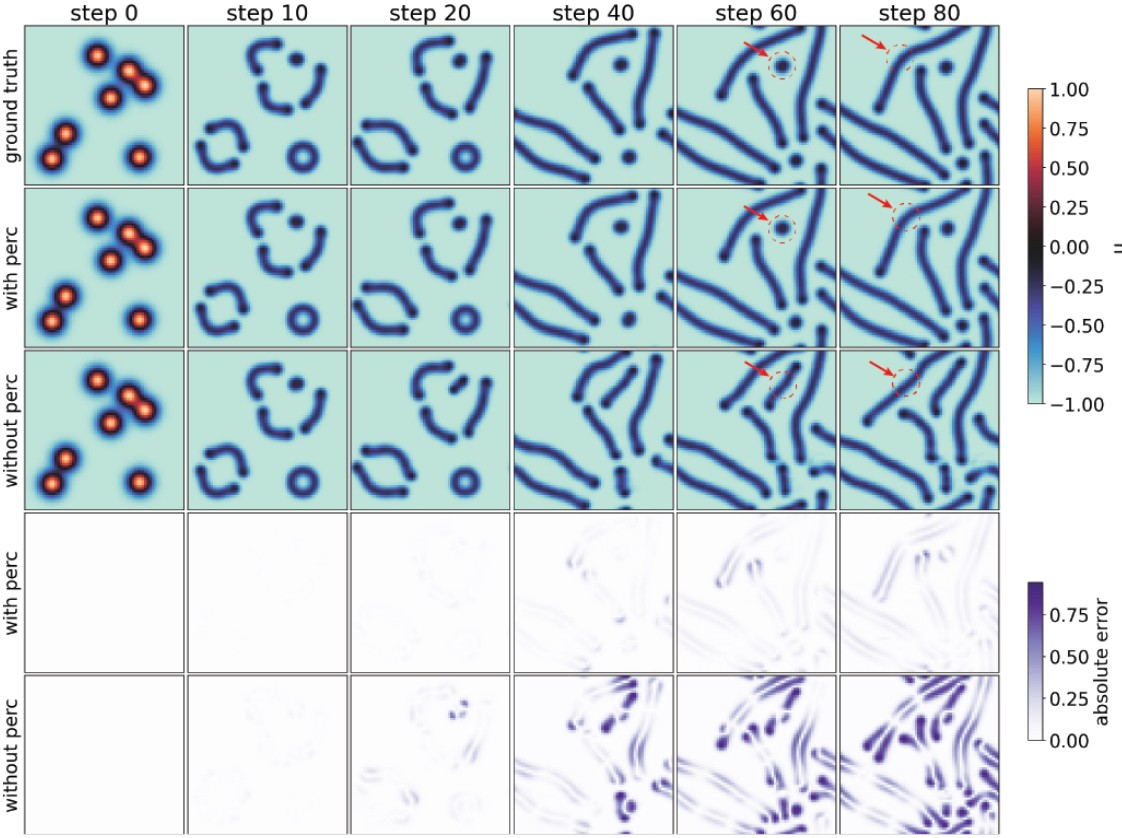

Figure 33: Effect of the perceptual loss on me-UNet rollouts for the Gray–Scott worm dataset (DS-6c). Top three rows: ground truth, prediction with $\mathcal{L}_{\mathrm{pc}}$, prediction without $\mathcal{L}_{\mathrm{pc}}$, at simulation steps $0, 10, 20, 40, 60, 80$. Bottom two rows: pixelwise absolute-error maps for the with- and without-$\mathcal{L}_{\mathrm{pc}}$ predictions, on a shared color scale. Without $\mathcal{L}_{\mathrm{pc}}$, geometric features such as the kink in the upper worm at step 80 are flattened, and errors concentrate along the worm bodies themselves rather than being spatially diffuse. With $\mathcal{L}_{\mathrm{pc}}$, curvature and connectivity of the extended structures are preserved over the full $T = 100$-step rollout.

