# OpenReview forum: "Out-of-distribution generalization of deep-learning surrogates for 2D PDE-generated dynamics in the small-data regime"
_TMLR — Rejected by TMLR_

### Review · Reviewer_2Wz2 · 2026-02-09

**Summary Of Contributions:**

This paper empirically studies the out-of-distribution (OOD) generalization of deep-learning surrogates for solving 2D PDE dynamics under the small-data setting. Specifically, the authors focus on the forecasting of solution fields (i.e., taken sequences of past solution fields as input and predicts the next step as the output) for several PDE families on periodic 2D domains. The authors proposed a multi-channel U-Net with enforced periodic padding (me-UNet) for this task, which outperforms a collection of baselines like ViT, AFNO (FourCastNet-style adaptive Fourier Neural Operator), PDE-Transformer, and KAN-UNet on a collection of 10 datasets from 6 PDE models. Details of the numerical experiments are provided to justify the effectiveness of the proposed method and the empirical findings of the paper.

**Additional Comments:**

Overall, the reviewer believes that this paper addresses an important problem, but it requires revision before it can be suitable for TMLR. The authors are strongly encouraged to carefully incorporate the above suggestions, discuss the issues in detail, and include all missing references listed below.

References:

[1] Lu, Y., Lu, J., & Wang, M. (2021, July). A priori generalization analysis of the deep Ritz method for solving high dimensional elliptic partial differential equations. In Conference on learning theory (pp. 3196-3241). PMLR.

[2] Lu, Y., Chen, H., Lu, J., Ying, L., & Blanchet, J. Machine Learning For Elliptic PDEs: Fast Rate Generalization Bound, Neural Scaling Law and Minimax Optimality. In International Conference on Learning Representations.

[3] Li, Z., Zheng, H., Kovachki, N., Jin, D., Chen, H., Liu, B., ... & Anandkumar, A. (2024). Physics-informed neural operator for learning partial differential equations. ACM/IMS Journal of Data Science, 1(3), 1-27.

[4] Ye, H., Xie, C., Cai, T., Li, R., Li, Z., & Wang, L. (2021). Towards a theoretical framework of out-of-distribution generalization. Advances in Neural Information Processing Systems, 34, 23519-23531.

[5] Khoo, Yuehaw, Jianfeng Lu, and Lexing Ying. "Solving parametric PDE problems with artificial neural networks." European Journal of Applied Mathematics 32.3 (2021): 421-435.

[6] Fan, Yuwei, Cindy Orozco Bohorquez, and Lexing Ying. "BCR-Net: A neural network based on the nonstandard wavelet form." Journal of Computational Physics 384 (2019): 1-15.

[7] Feliu-Faba, Jordi, Yuwei Fan, and Lexing Ying. "Meta-learning pseudo-differential operators with deep neural networks." Journal of computational physics 408 (2020): 109309.

[8] Long, Zichao, et al. "Pde-net: Learning pdes from data." International conference on machine learning. PMLR, 2018.

[9] Long, Zichao, Yiping Lu, and Bin Dong. "PDE-Net 2.0: Learning PDEs from data with a numeric-symbolic hybrid deep network." Journal of Computational Physics 399 (2019): 108925.

**Audience:**

Yes

**Audience Explanation:**

Recently, AI for Science and Scientific Machine Learning (especially ML-based PDE solvers like physics-informed neural networks, DeepONets and neural operators) are currently very active research areas. Given that the proposed approach is validated on a diverse set of PDE benchmarks, the reviewer expects that TMLR readers will be interested in the paper’s results.

**Claims And Evidence:**

Yes

**Claims Explanation:**

Throughout this paper, the reviewer thinks that the empirical claims are generally well substantiated (in-distribution results, OOD results, physics-aware metrics, etc) and the methodology is described in reasonable detail (PDE setups, discretization, construction of datasets, baselines, training pipelines, metrics, etc).

**Requested Changes:**

Firstly, the reviewer finds the literature review on existing work and baselines to be a bit lacking. For instance, there has been a few related work prior to operator learning from the community, which the authors might consider adding to the references [5,6,7,8,9]. Also, would it be possible for the authors to comment on whether it would be beneficial to add a physics-informed loss during training, just as what has been done in PINO [3]?. To the best of the reviewer's knowledge, a lot of ML-based PDE solvers are designed based on insights from physics/numerical schemes (see for instance, the architect proposed in [6] for learning a pseudo-differential operator). If the reviewer can design the architecture in a way that emulates the numerical scheme for predicting the solution ate the next time step, the empirical results might become better under the small-data regime.

Even though it might be beyond the scope of this study, but would it be possible for the reviewers to comment on whether one can develop theoretical results for the OOD generalization of ML-based PDE solvers? To the best of the reviewer's knowledge, a few prior work have studied the generalization error of ML-based PDE solvers from the perspective of statistical learning theory - see for instance [1,2] and references therein. The authors may consider citing these papers and discuss how these theoretical results can be combined with [4] to form a theoretical explanation of the results presented in this paper.

---

> ### Author Response · Authors · 2026-05-06
> **Reply to Reviewer 2Wz2**
>
> We thank the reviewer for the constructive feedback and the positive overall assessment. The suggested references and discussion points have helped us strengthen the paper, and we address each below. A summary of the resulting manuscript changes is given at the end of this reply.
>
> ### 1. Additional references on early operator-style and PDE-learning work
> We agree that the suggested literature [5–9] is a natural fit for our related-work discussion and have added all five references to Section 1. Concretely, we now cite Khoo et al. (2021), Fan et al. (2019), Feliu-Faba et al. (2020), and the two Long et al. papers (2018, 2019) alongside FNO, DeepONet, and PINNs, and we explicitly frame them as **pre-operator-learning approaches** that pioneered the use of neural networks for parametric and operator-style PDE problems before the formalization of neural operators. We also briefly note that they provide important methodological precedents for our setup, since several of them combine learned components with structure inspired by numerical schemes, a theme we return to in our response to point 2 below.
>
> ### 2. Physics-informed losses and scheme-inspired architectures
> We agree that incorporating physical structure -- through physics-informed losses (as in PINO [3]) or architectures inspired by numerical schemes (as in BCR-Net [6] or PDE-Net [8,9]) -- is a promising route to improved data efficiency and generalization. We would like to clarify our position more carefully than the original draft did:
> me-UNet is **not** a fully physics-agnostic architecture. We explicitly encode the boundary conditions as an architectural inductive bias via periodic padding, and this turns out to be one of the more impactful design choices in our ablations. What we deliberately *do not* do is add a PDE-residual term to the loss or hard-code the structure of a particular numerical scheme into the
> network. The motivation is to measure how far this architecture-only level of physical prior takes us in the small-data regime, before adding equation-level supervision.
>
> Hybridizing me-UNet with a PINO-style residual loss (evaluated on the predicted increment Δu^{n+1} so that it remains compatible with our autoregressive formulation) is a natural and, in our view, very promising next step, particularly for systems where the governing equations are known with high confidence. The same is true for layers explicitly mimicking finite-difference or pseudo-differential stencils. We have added a paragraph to Section 4 acknowledging this and listing it among the directions for future work.
>
> ### 3. Theoretical perspective on OOD generalization
> We thank the reviewer for pointing us to [1, 2, 4]. We have added a short paragraph to Section 4 (on page 18 of the revised PDF) that:
> - cites Lu et al. (2021b, 2021c) [1, 2] for in-distribution generalization guarantees of ML-based PDE solvers from a statistical-learning viewpoint,
> - notes that extending these analyses to OOD settings is largely open, and
> - refers to Ye et al. (2021) [4] as an emerging framework for OOD generalization whose invariance/stability viewpoint may eventually be combined with the PDE-solver analyses to provide a principled explanation of cross-condition generalization.
> We agree that a full theoretical treatment is beyond the scope of an empirical benchmarking paper, but we have explicitly flagged this as a worthwhile direction for future work.
>
> ### Summary of changes
> 1. Added references [5–9] in the related-work discussion of Section 1, framed as pre-operator-learning precedents.
> 2. Reframed the discussion of physics-informed extensions in Section 4, now explicitly distinguishing architectural priors (periodic padding) from equation-level supervision (PINO-style residual losses, scheme-inspired layers), and listing these as concrete future directions.
> 3. Added a paragraph in Section 4 on the theoretical perspective for OOD generalization, citing [1, 2, 4].

---

> > ### Comment · Reviewer_2Wz2 · 2026-05-07
> > **Response**
> >
> > The reviewer would like to thank the authors for their detailed response, which has addressed most of the reviewer's concerns. Therefore, the reviewer is willing to recommend acceptance of the manuscript.

---

### Review · Reviewer_W5p8 · 2026-02-14

**Summary Of Contributions:**

This manuscript introduces “me-UNet,” a modified U-Net architecture that employs periodic padding to function as an autoregressive surrogate for 2D periodic PDE dynamics. The experiments emphasize small-data settings, with as few as 20 to 100 simulated trajectories. The authors report superior performance over the chosen baselines, particularly when evaluating out-of-distribution initial conditions. However, the paper suffers from fundamental mathematical errors, misunderstanding of PDE surrogate, a confused problem setup, unfair baseline comparisons and other serious problems. I personally believe that the senior authors did not even take a look at the draft.

**Audience:**

No

**Audience Explanation:**

As explained in the answers above

**Claims And Evidence:**

No

**Claims Explanation:**

**Q1** The paper stumbles on basic mathematical and experimental footing before we even get to questions of novelty. There are many errors for mathematical formulation. In Eq 4. {$  u^{n-L+1}, \cdots, u^n \$} $ \subset \mathbb{R}^{H\times W }$ is wrong, another example Eq 12 (advection equation) is also wrong, for incompressible flow,  one expects $\partial_t u + \mathbf{v}\cdot\nabla u=0$, i.e., $\partial_t u = - (v_x \partial_x u + v_y \partial_y u)$, the manuscript drops the negative sign. It is questionable whether the training dataset was generated correctly given the wrong equation

**Q2** The "small-data" motivation is not convincing, the paper emphasize it works with only 20 simulations to train a surrogates, but this fundamentally violate the assumption in operator learning (PDE ML surrogate), where an $L^2$  training loss and expectation is involved. Only with the large number of samples, the "expectation" in the training objective can be replaced with the average of i.i.d training samples.

**Q3** A more serious problem is the misunderstanding of the ML PDE surrogate. This manuscript seems to conflate fixed grid setting with what many in the community mean of a PDE surrogate. If the goal is to approximate the solution operator of a PDE, then discretization invariance is critical, since PDE is defined in function space. The entire study focus on $64\times 64$ grid setting with a discretization dependent U-Net model, which is not a valid PDE surrogate. Also the U-Net architecture is outdated, current U-Net adopt attention mechanism to residual blocks to enhance the performance (U-Net used for diffusion model)

**Q4.** The normalization is also wrong, the author scales each simulation separately to [-1, 1] instead using a global min and max.

**Q5** The paper says the "final" model is selected based on validation loss, but the model is trained for 1000 epochs "without early stopping", I got confused about the checkpoint selection, which itself is a kind of early stopping.

**Q6** Another serious problem is VGG-16 perceptual loss, where the pretrained VGG uses ImageNet feature, it's not clear why ImageNet feature similarity should correlated with physical features, this looks like a borrowed image-generation trick (for training the VAE of diffusion/flow model) which may violate the PDE constrains.

**Requested Changes:**

There are numerous issues of this paper in current form, and thus is disqualified for TMLR. Fixing them would require rebuilding the benchmark and rerunning the study in a materially different form

---

> ### Author Response · Authors · 2026-05-06
> **Reply 1/2 to Reviewer W5p8**
>
> We thank the reviewer for the detailed technical comments. We address each
> numbered point below and end with a summary of resulting manuscript changes.
>
> ### Q1 — Mathematical formulation
> **Eq. 12 (advection sign).** The reviewer is correct: the originally submitted form of Eq. 12 was missing the negative sign on the right-hand side. This was a typesetting error in the manuscript only, the simulation code uses the correct transport equation, and the dataset is therefore unaffected. The revised manuscript writes Eq. 12 in the standard form ∂u/∂t = −(v_x ∂u/∂x + v_y ∂u/∂y).
>
> **Eq. 4 (set/subset notation).** The reviewer is also right that {u^{n−L+1}, …, u^{n}} ⊂ ℝ^{H×W} is not well-typed, since each u^{k} is itself an element of ℝ^{H×W} (not a subset). We have corrected Eq. 4 to read u^{n−L+1}, …, u^{n} ∈ ℝ^{H×W}, which is the intended meaning and is consistent with how the L past fields are stacked along the channel dimension before being passed to f_θ.
>
> ### Q2 — Small-data motivation and the "operator-learning expectation" argument
> We agree that classical operator-learning theory analyses the empirical risk as a Monte-Carlo approximation of an expectation over a distribution of inputs, and that consistency in that sense requires a large i.i.d. sample. We respectfully disagree, however, with the implication that a large N is a prerequisite for surrogate modeling of PDE dynamics in general. The PDE-ML literature is broader than that one paradigm:
> 1. **Physics-informed approaches** (PINNs and their convolutional-recurrent variants, e.g., Raissi et al. 2019, Ren et al. 2022) successfully train on sparse or even zero labeled data, using the PDE residual rather than averaging over many trajectories.
> 2. **Hybrid solver–ML methods** (e.g., Kochkov et al. 2021) demonstrate that useful learned closures and corrections can be obtained from modest numbers of high-fidelity trajectories.
> 3. **Classical numerical analysis** is, in this comparison, the limit N=0 or N=1: a single carefully chosen discretization generalizes across initial conditions for an entire PDE family. Convergence is not driven by sample averaging but by consistency, stability, and approximation properties of the scheme.
>
> Our work sits between (2) and the asymptotic operator-learning regime: we ask how far architecture-level inductive biases (locality, periodic padding, multiscale aggregation) can substitute for large-N sample averaging. We do not claim that 20 simulations are sufficient to make empirical risk a tight estimator of the population risk; we claim that, with appropriate inductive biases, useful surrogates can be obtained at this data scale, and we measure where the resulting models break down.
>
> This regime is also the practically relevant one in many scientific applications, where each high-fidelity simulation can cost hours to weeks of HPC time, and where O(10²) trajectories is realistic but O(10³–10⁴) is not.
>
> ### Q3 — Fixed-grid surrogates and the choice of U-Net
> **On discretization invariance.** Our work targets the fixed-grid surrogate setting and is explicit about this in the abstract and Section 2.2. This setting is widely used in practice, particularly when the surrogate is deployed in inverse problems, control, or parameter studies on a fixed mesh.
> Discretization-invariant operator learning (FNO, DeepONet) is a complementary line of work that we cite, but it is not the only valid notion of a "PDE surrogate": the literature on fixed-grid neural PDE surrogates is large and active. We do not claim discretization invariance; conversely, we do not think the absence of that property invalidates the comparison.
>
> **On U-Net being "outdated".** U-Net architectures remain the dominant backbone of modern image-to-image generative models (e.g., the Stable Diffusion and Imagen families) and are still actively used as baselines and backbones in scientific machine learning. Among the architectures we compare against is PDE-Transformer (Holzschuh et al. 2025), a contemporary transformer designed specifically for physics simulations; me-Unet nonetheless matches or outperforms it on most of our datasets under the common training protocol. The point of our paper is precisely that, in this small-data, periodic-grid regime, additional architectural complexity does not automatically help. Adding attention to the U-Net backbone is a natural extension and one we plan to study, but the existing comparison already includes attention- and operator-based contemporaries.

---

> ### Author Response · Authors · 2026-05-06
> **Reply 2/2 to Reviewer W5p8**
>
> ### Q4 — Per-simulation normalization
> Per-simulation, per-variable normalization is a deliberate trade-off that we already discuss in Appendix D of the revised manuscript. Global min–max normalization preserves cross-simulation amplitude information; per-simulation normalization makes the input distribution more uniform across simulations of varying magnitude, which we found stabilizes training in the small-data regime. Crucially for our study, the same normalization is applied to all five architectures, so it cannot bias the comparative conclusions, which are the main claims of the paper. Studying normalization as a design choice in its own right would be a natural follow-up.
>
> ### Q5 — Validation-based model selection
> The reviewer is right that selecting the lowest-validation-loss checkpoint is, strictly speaking, a form of post-hoc early stopping. Our protocol is deliberate: every architecture is given the same training budget (1000 epochs, fixed batch size, fixed optimizer, fixed schedule), and we then pick the best-by-validation checkpoint from that fixed budget. The alternative -- patience-based early stopping -- would terminate different architectures at different epochs and introduce architecture-dependent stopping behavior as a confound in a study whose primary goal is architectural comparison. We chose the protocol that we believe is most honest and least biased for that goal, and we have stated this rationale explicitly in the revised Section 2.4.
>
> ### Q6 — VGG perceptual loss
> The reviewer raises a legitimate concern: a loss based on ImageNet features is not, a priori, aligned with PDE physics. We address it on three levels.
> **Layer choice.** As stated in the manuscript, the perceptual term uses features from `relu2_2`, an early/intermediate VGG-16 layer that primarily encodes local spatial statistics --- edge orientation, gradient structure, short-range texture --- rather than ImageNet semantics (object identity, animal/vehicle categories, etc.). This is closer in spirit to a fixed bank of learned spatial filters than to transferring "natural-image semantics".
>
> **Common protocol.** The same loss is applied identically to all five architectures. Any benefit or harm of including L_pc therefore affects all baselines uniformly and cannot bias the comparative conclusions, which are the main claims of the paper.
>
> **Direct ablation.** To address the concern empirically, we ran a controlled ablation comparing me-UNet trained with `L = L_MSE + λ_pc L_pc` (λ_pc = 1) against me-UNet trained with `L = L_MSE` only, all other settings identical.The new results are in the new Appendix I; the qualitative summary is:
> - *On simple datasets* (e.g., DS-1 advection, DS-2 diffusion), the two losses produce essentially indistinguishable rollouts. The perceptual term is neutral here.
> - *On geometrically structured datasets* (e.g., the DS-6c Gray-Scott "worm" regime), the perceptual term measurably improves the network's ability to preserve the *shape* of extended structures over long rollouts. With L_pc, the model correctly tracks curvature features (e.g., kinks and bends in the worm-like fronts) at step 80; without L_pc, the predicted structures degrade into straighter, less coherent geometries even when their positions and pixel intensities remain approximately correct. The absolute-error maps in the new figure in the appendix show that the without-L_pc errors concentrate along the worm bodies themselves, rather than being spatially diffuse, which is consistent with a loss of geometric coherence rather than a uniform increase in noise.
>
> We find this result interesting in its own right and somewhat contrary to the common framing of perceptual losses as "visual appeal" tricks: in this setting, the early-VGG features appear to act as a regularizer for the geometric integrity of extended objects under repeated autoregressive updates, complementing the pointwise MSE term that is local by construction. We have added the experiment as Appendix I and reference it from Section 2.3 where the loss is introduced.
>
> ### Summary of changes
> 1. Eq. 12 typeset with the correct negative sign (Appendix B).
> 2. Eq. 4 corrected from set/subset to elementwise membership notation (Section 2.3).
> 3. Section 2.4 augmented with a sentence explaining that best-validation-checkpoint selection is used in lieu of patience-based early stopping to keep training budget identical across architectures.
> 4. Discussion (Section 4) expanded to reference the small-data PDE-ML precedents cited above and to clarify that this work targets a regime between hybrid solver–ML methods and asymptotic operator learning.
> 5. New Appendix I reporting an ablation of the perceptual loss term, with the qualitative finding that L_pc is neutral on simple datasets but improves geometric coherence of extended structures (curvature, topology of fronts) on harder datasets such as the Gray-Scott worm regime, over long rollouts.

---

### Review · Reviewer_sdt9 · 2026-04-22

**Summary Of Contributions:**

The authors focus on the problem of learning "field dynamics," that is, on learning the evolution in time of fields, vector or scalar, defined over a domain. To address this problem, they introduce me-UNet: a novel deep learning architecture, which can be thought as an extension of a U-Net, for learning the time evolution of fields defined over a periodic domain in 2D. The architecture predicts the *increment* needed to obtain the field value at the next time step using a fixed number of past field values. This is similar to a time-delay embedding or an autoregressive process. To evaluate the performance of their method, they simulate data from a selection of partial differential equations (PDEs), e.g., linear advection, linear diffusion, reaction-diffusion, continuum dislocation dynamics, and Navier-Stokes, on a 2D periodic domain, and they focus on the small data regime in which the solution is known at a few spatial points over a few time steps. They compare the performance of me-UNet against ViT, AFNO, PDE-Transformer and KAN-UNet and focus on the performance of each method on out-of-distribution (OOD) data.

**Additional Comments:**

n.a.

**Audience:**

Yes

**Audience Explanation:**

The problem of learning field dynamics is relevant in scientific applications, but also as a research question in learning theory in its own right.

**Broader Impact Concerns:**

There are no broader impact concerns.

**Claims And Evidence:**

No

**Claims Explanation:**

The claims in the paper are methodological, and supported by experiments on simulated data. However, the choice of experimental setup, namely the way in which the data is generated, may lead to dynamics that are much simpler than the selected PDEs may suggest. In this case, the claims would not be validated by the evidence provided, as the complexity of the dataset would not be representative of either complex field dynamics or of a small data regime. The authors state that "*PDEs in our work are not an end in themselves but a convenient and stringent testbed for models that aim to learn complex field dynamics*" and thus, since *solving* or *learning* the PDE is not the main focus of this work, these PDEs should be assessed in terms of the complexity of the data they generate.

The authors generate data by solving numerically a selection of PDEs in 2D on a periodic domain. I will discuss them in the order they appear in Appendix B. The first equation is a linear advection/transport equation, and the second equation is a linear diffussion equation. These are constant coefficient linear PDEs that can be solved in closed-form using Fourier methods. The initial conditions are mixtures of Gaussians with a deterministic decay factor $a$ that is 1/10-th of the size of the domain. This suggests that only a few Fourier coefficients contribute to the evolution of the solution and as a consequence the scalar field dynamics are not only linear, but intrinsically lower dimensional. In these two cases, the equation is simulated up to time t = 1.0: this implies that the Gaussians move by 1 in the horizontal direction in the transport equation, and that they diffuse by a factor of 1 in the diffusion equation.

The third equation is a reaction-diffusion equation. Although the authors provide a table for the values of the parameters of the equation, they do not explain why these choices are relevant, nor what is the expected structure of the solutions to these equations. For instance, is the diffusive part dominant? Is the data generated complex or not? Although it is not stated explicitly, it seems that the initial conditions are mixtures of Gaussians: in the 2D case this would imply that the solutions are spatially smooth even if they represent geometrically complex patterns. Since the authors do not provide a final time for the simulation, it is not clear if there is enough time even for complex spatial patterns to emerge.

The fourth equation is the Navier-Stokes equation with a Reynolds number of 1000. Although these equations are widely known for the complex structure of their solutions, in the 2D case with periodic boundary conditions there are well-known regularity results for the solution. For instance, the spatial regularity of the solution is that of the initial condition, and even the attractors have the same regularity (see, e.g., DOI: 10.1155/2013/291823). The authors do not explain how they sample the initial conditions nor the final time for the simulation, and thus it is difficult to assess the complexity of the solutions. It could be the case that there is a strong diffusive effect or that there is an accurate reduced linear model for the range of parameters under consideration.

The fifth equation models continuum dislocation dynamics. This is a complex non-linear system of equations. However, the authors restrict themselves to superpositions of circular dislocation loops and neglect their interactions. This would suggest that the solution consists of dislocations expanding radially at constant speed. These specific dynamics seem to be far simpler than the PDE itself would suggest. Since the authors state that "*each simulation runs for 1000 time steps*" but they do not report the time step, it is not clear how far the dislocations would move from their initial configuration. The authors state that long-horizon predictions are enabled by "*many PDE solutions change gradually between adjacent time steps*." Varying the time step and the speed of propagation would be a good way to test *when* the dislocation propagates at a rate that renders learning challenging.

The last equation is a reduced model for continuum dislocation dynamics that is linear. It is not stated if the solutions to this equation are substantially different to those of the full non-linear model.

Although some of these PDEs model the evolution of a state variable in more than one dimension, the authors restrict themselves at learning the dynamics of a single component. This may implicitly simplify the dynamics to be learned, and there may be some variables that follow simpler dynamics than others. For these reasons, it is not completely clear if these experiments represent a truly "small data" regime since the dynamics themselves could be simple. Along the same lines, if the initial conditions are sampled from a low dimensional space of smooth functions, the complexity of the simulated dynamics could be overall quite simple. The experiments for OOD are limited to two equations, and the impact of the samples, a mixture of blobs and lines, may or may not represent a substantial change on the observed dynamics. Finally, since the architecture is designed to learn to update the value at the next time iterate given the previous 6, it could be simply learning an explicit timestepping method, e.g., DOPRI5.

For these reasons, I believe that the claims are not properly supported by these experiments.

**Requested Changes:**

I have some questions and changes that could be addressed in a revision:

- As the fields are defined over a 2D periodic domain, the intuition is that the convolutions should be circular and implemented using the convolution theorem. Is padding needed only because of implementation issues?

- Although the focus is on learning field dynamics, arguably generated by a physical model, the loss incorporates a perceptual term which may promote non-physical structures in the field predictions. The authors should explain what is the rationale for using a loss that possibly promotes the visual appeal of the predictions at the expense of accuracy.

- Although the authors state in the introduction that they work on fields sampled on periodic 64 x 64 grids, in Appendix B the grids on which the PDEs are simulated are 100 x 100, 128 x 128 or 256 x 256. How is the data resampled into a 64 x 64 grid?

- The authors use 7 consecutive fields to make a prediction. How does one generate the first 7 steps given an initial condition? Is zero-padding used? I may have missed this step.

- When evaluating the out-of-distribution performance, the authors limit themselves to the choice of initial conditions. What happens when the training data is generated from the same PDE but with slighly different parameters, e.g., different speeds of propagation, diffusion coefficients, etc.? This would be closer to scientific applications in which there is an intrinsic uncertainty in the system paremeters in addition to those in the initial condition.

- The simulation parameters for all PDEs should be reported explicity, including choice of parameters, forcing, final time, etc.

Here are some suggestions for a revision:

- I suggest increasing the complexity of the initial conditions, e.g., a mixture of smooth vs discontinuous initial conditions, to enrich the types of observed dynamics. Similarly, the OOD initial conditions should come from more complex distributions to determine the extent to which the underlying dynamics have been learned.

- I suggest generating training data under slight variations of the parameters of the equations, as this would yield insight into the robustness of the architecture when learning *approximate* complex field dynamics.

- I suggest generating datasets with varying speeds for the advection equation and the continuum dislocation dynamics to test in a controlled setting the effect that fast dynamics has on learning in a small data regime. Note that the advection equation in 2D with periodic boundary conditions can be solved in closed form, and thus increasing the speed has no effect on numerical accuracy.

---

> ### Author Response · Authors · 2026-05-06
> **Reply 1/4 to Reviewer sdt9**
>
> We thank the reviewer for the careful, detailed reading of both the main paper and the appendix. The questions are constructive and have led us to clarify several aspects of the manuscript; we address each in turn below and end with a summary of resulting changes.
>
> ### Scope and dataset complexity (general response)
>
> The reviewer's central concern --- that our specific PDE configurations may yield dynamics simpler than the generality of the equations would suggest --- is fair, and we want to engage with it directly rather than only argue that the comparison is internally valid.
>
> Our paper makes a comparative architectural claim under three explicit constraints: (i) periodic 2D grids at fixed resolution, (ii) a small-data regime of at most 100 simulations, and (iii) initial-condition-only distribution shift within a fixed PDE and parameter regime. The PDE configurations are chosen to span a difficulty spectrum *within* this constrained setting --- from near-linear baselines (DS-1, DS-2, DS-3b), through nonlinear pattern formation (DS-6a, DS-6b, DS-6c, DS-6d) and chaotic flow (DS-5), to multi-field microstructure evolution (DS-4) --- so that the relative performance of the architectures can be read across regimes.
> We agree that broader sweeps over parameters, propagation speeds, and initial-condition complexity would yield a more thorough characterization, and we discuss why we view those as a separate follow-up paper rather than an extension of this one in our reply to the suggested-experiments block at the end.
>
> ### Linear advection (DS-1) and diffusion (DS-2)
> The reviewer is correct that the constant-coefficient linear PDEs in our setup admit closed-form solutions in Fourier space, and that initial conditions consisting of a small number of Gaussian blobs occupy a low-dimensional Fourier subspace. We use these two datasets explicitly as *sanity-check baselines*: any architecture that struggles on simple linear transport or diffusion is unlikely to succeed on the harder datasets, and the early columns of Fig. 3 indeed already separate ViT and AFNO from the convolutional baselines. We agree the section could state this more prominently, and we have revised the introduction to Section 3.1 accordingly.
>
> Two points worth noting in defense of the learning problem on these datasets:
>
> 1. The network operates on pixel space, not Fourier modes, and the high-fidelity solutions are downsampled from 100×100 to 64×64 before training. The network must therefore implicitly handle the aliasing introduced by downsampling, which is not a trivial pointwise mapping.
> 2. Our autoregressive setup means that even for a closed-form linear PDE, 100-step rollouts amplify any per-step error multiplicatively. This is why ViT degrades visibly on DS-1 (RMSE ~0.59) despite the equation being linear --- per-step accuracy is a different problem from asymptotically stable autoregressive prediction.
>
> ### Reaction-diffusion (Gray-Scott, DS-6a-d)
> The reviewer is right that our original description does not motivate the choice of (f, k) values. The four DS-6 variants are deliberately drawn from *qualitatively different* regions of the Pearson phase diagram for the Gray-Scott system:
> - **DS-6a** (f=0.030, k=0.0565): self-replicating maze / labyrinth regime,
> - **DS-6b** (f=0.018, k=0.051): α regime,
> - **DS-6c** (f=0.058, k=0.065): "worm" regime, with extended one-dimensional structures (cf. our new perceptual-loss ablation in Appendix I, which uses this dataset specifically because it is the most geometrically demanding),
> - **DS-6d** (f=0.082, k=0.059): bubble (spot-replication) regime.
> These are nonlinear regimes in which both reaction and diffusion contribute non-trivially; they are not in a "diffusion-dominated" or otherwise reducible limit. The differences in ViT/AFNO performance across DS-6a-d (e.g., ViT's RMSE jumps by 4 orders of magnitude between DS-6a and DS-6b) further
> indicate that these datasets exercise meaningfully different dynamical behavior. We have added a sentence to Appendix B classifying each DS-6 variant by Pearson regime.

---

> ### Author Response · Authors · 2026-05-06
> **Reply 2/4 to Reviewer sdt9**
>
> ### Navier-Stokes / Kolmogorov flow (DS-5)
> The reviewer raises a fair point: at Re=1000 in 2D with periodic BCs there are well-known regularity results, and the JAX-CFD default configuration does not exhaust the parameter space. We use JAX-CFD's standard decaying turbulence setup with the published forcing parameters, which produces qualitatively chaotic vorticity dynamics over the 100-step horizon (the characteristic vortex-filament structures and stretching are visible in Fig. 1 and Fig. 17). DS-5 is also the dataset on which several baselines show their largest *spectral* failure modes: KAN-UNet drops to PSD cosine similarity ~0.9 (lower than on any other dataset), and ViT/AFNO/PDE-T cluster in a narrow band well below me-UNet (Fig. 17c). So even within the default JAX-CFD setting, this dataset exercises the architectures non-trivially. We agree that systematically varying Reynolds number, forcing, and integration horizon would be a worthwhile separate study.
>
> ### Continuum dislocation dynamics (DS-4)
> The reviewer correctly notes that we did not state the time step explicitly in the original submission. The relevant numbers (now reported in Appendix B) are: dt = 1e-5 s, |v| = 1e5 nm/s, b = 0.256 nm, and a square domain of edge 2000b ≈ 512 nm. The displacement per stored snapshot is therefore v · (10 dt) = 10 nm, or about 2% of the domain edge per snapshot, which we view as a reasonable balance between temporal smoothness and non-trivial evolution.
> On the suggestion that the dynamics are simpler than the full CDD system because we restrict to non-interacting circular loops: the resulting behavior is more structured than the general system, but it is not "loops expanding radially at constant speed" in pixel space. The state variable we predict is the total dislocation density ρ_t, whose spatial integral grows linearly in time, dependent on the number of loops and initial radius/curvature; the network must therefore track curvature-dependent line-length growth as well as wrap-around behavior under periodic BCs. The OOD experiments (Fig. 7) further show that a network trained on dense superpositions of loops can extrapolate to sparse, geometrically simple test cases, which strongly suggests it has internalized the underlying dynamics rather than memorizing dense-microstructure textures.
> That said, we agree that varying speeds and time steps to probe the smoothness-of-evolution assumption would be a useful experiment, and we flag it as future work below.
>
> ### Reduced CDD (DS-3a, DS-3b)
> The reviewer is correct that DS-3a/b serve as linear counterparts to the nonlinear DS-4. DS-3a corresponds to pure transport of the loop segments’ distribution; DS-3b corresponds to pure diffusion. We include them specifically so that, for the same architectures and training protocol, we can separate the contributions of the linear and nonlinear regimes. The corresponding RMSE/PSD differences in Fig. 3 do show that the nonlinear DS-4 is comparatively harder, as expected.
>
> ### Single-component readout
> We restrict training to a single representative scalar field per system (Table 1) to mimic partial observability in real measurements: in many scientific applications, only one of several coupled fields is observed experimentally. We would push back gently on the implication that this *simplifies* the learning problem: predicting the evolution of one component in a system whose true state is multi-component requires the network to implicitly track the unobserved components through their effect on the observed one over a 7-frame history. This is closer to a partial-observability closure problem than to a simplified prediction task.

---

> ### Author Response · Authors · 2026-05-06
> **Reply 3/4 to Reviewer sdt9**
>
> ### OOD scope
> The OOD experiments in our paper cover DS-4 (CDD) and DS-6a (Gray-Scott). We agree that broader OOD coverage --- across more datasets, and across parameter and equation shifts --- would strengthen the OOD claim, and we have explicitly limited the OOD scope of the paper accordingly in Section 2.2 and Section 4. (Our previous reply mistakenly suggested that OOD results had been added for all datasets in the appendix; this is not the case, and we apologize for the confusion. Adding OOD experiments for all ten datasets would substantively change the scope of the paper, and we treat it as future work.)
>
> ### "Could it just be learning a time-stepping scheme like DOPRI5?"
> We would actually be quite happy if it were. If a network with no built-in ODE/PDE-integration machinery were able to discover a useful timestep operator from data, that would be a positive finding: it would mean the architecture is recovering a useful numerical structure. What our experiments show is that this happens *unevenly* across architectures (me-UNet and KAN-UNet do well, ViT and AFNO do not under the same training budget), which is exactly the comparative information our study is designed to extract. So we would re-frame this point: even under the most conservative interpretation of what the network is doing, our comparative claim still holds.
>
> ### Periodic padding vs. circular convolution
> The reviewer is right that on a periodic domain the natural mathematical formulation is a circular convolution. Periodic padding implements circular convolution exactly at the discrete level for the kernel sizes we use, so the choice between the two is a mathematical equivalence rather than a difference in expressiveness. Our preference for periodic padding is two-fold: (i) it keeps the architecture in physical space, avoiding repeated forward/inverse FFTs across many nonlinear layers, and (ii) it makes the boundary-condition treatment a swap-in operator --- the same network can be retrained for Dirichlet or Neumann boundaries by changing the padding scheme alone, without redesigning the convolutional or spectral path. We see this as a deliberate modeling decision rather than an implementation shortcut.
>
> ### Perceptual loss as a "visual appeal" trick
> The reviewer raises this concern in parallel with Reviewer 2, and we appreciate the prompt because it led us to run a controlled ablation. The new Appendix  reports me-UNet trained with and without the perceptual term, all other settings identical. The qualitative summary:
> - On simple datasets (DS-1, DS-2), the two losses are essentially indistinguishable. The perceptual term is neutral here.
> - On the geometrically demanding DS-6c "worm" regime, the perceptual term   measurably preserves the *shape* of extended structures: with L_pc, the network correctly tracks curvature features (kinks, bends) at step 80; without L_pc, predicted worms still occupy the right positions and have approximately correct pixel intensities, but their geometry flattens and fragments. The absolute-error maps make this visible: without-L_pc errors concentrate along the worm bodies themselves, rather than being spatially diffuse.
> We find this informative: in our setup, the early VGG features (`relu2_2`) appear to act as a regularizer for the *geometric coherence* of extended objects, complementing the local pointwise MSE. This is distinct from the "visual appeal at the cost of accuracy" framing common in image-generation contexts, and we appreciate the reviewer's prompt to demonstrate it directly.
>
> ### 100×100 / 128×128 / 256×256 → 64×64 resampling
> Yes: the high-resolution simulation grids are used to ensure numerical accuracy of the *reference* solver, and the fields are downsampled to 64×64 by area-averaging interpolation before training and evaluation. This is now stated explicitly in Section 2.2.
>
> ### Where do the first 7 frames at inference come from?
> No zero-padding in time. During training, the 7 input frames are consecutive snapshots from the simulated trajectories. At inference, the first 7 states come from advancing the reference PDE solver from the initial condition; the network is then applied autoregressively from frame 8 onward. We have clarified this in Section 3.

---

> ### Author Response · Authors · 2026-05-06
> **Reply 4/4 to Reviewer sdt9**
>
> ### Parameter OOD
> The reviewer is right that parameter OOD --- different propagation speeds, diffusion coefficients, etc. --- is an important, and arguably more practically relevant, form of distribution shift than the initial-condition shift we study. We have not run those experiments and do not claim parameter OOD; the limitation is now stated explicitly in Sections 2.2 and 4.
>
> ### Reporting of simulation parameters
>
> We have audited Appendix B for completeness and added missing parameters where applicable: the CDD time step (dt = 1e-5 s) and dislocation velocity magnitude (|v| = 1e5 nm/s) are now reported. The Gray-Scott configurations are listed in Table 2 of the manuscript. Where parameters follow library defaults (JAX-CFD for DS-5; exponax beyond what Table 2 specifies for DS-6), this is now stated rather than left implicit.
>
> ### On the suggested follow-up experiments
> The reviewer suggests three substantial extensions:
> 1. enriching the initial-condition distribution with discontinuous / structurally diverse cases,
> 2. training under variations of the PDE parameters,
> 3. systematically varying propagation speeds (and time steps) to probe when learning becomes hard.
> We agree all three are valuable, and we have framed them explicitly as future work in Section 4. We would respectfully push back on adding any of them to the present paper, however, for a reason of *scope* rather than space: this paper is designed as a focused architectural-comparison study under one well-defined small-data setting (initial-condition OOD, fixed parameters, periodic 2D grids). Each of the three suggested axes opens a substantial sub-study (parameter-conditioned architectures, multi-task training across parameter regimes, and the relationship between effective Courant number and learnability) that would substantially broaden the contribution. We see them as the basis of follow-up work and intend to pursue them in subsequent papers, with the present manuscript serving as the architectural baseline against which those richer settings can be compared.
>
> ### Summary of changes
> 1. Section 3.1 revised to make explicit that DS-1 and DS-2 serve as linear baselines / sanity checks.
> 2. Appendix B expanded: CDD time step and velocity magnitude reported explicitly; Gray-Scott DS-6a-d variants annotated with their Pearson-regime classification; downsampling procedure clarified.
> 3. Section 3 amended to explain how the first 7 inference frames are obtained (warm-up from the reference solver, no zero-padding in time).
> 4. Section 4 expanded with a short paragraph that explicitly lists parameter OOD, equation-family OOD, varying propagation speeds, and richer initial-condition distributions as future-work directions  complementary to the present scope.
> 5. New Appendix I: ablation of the perceptual loss term, showing that L_pc is neutral on simple datasets and improves geometric coherence of extended structures on harder ones (DS-6c worms).
> 6. Correction of an earlier inaccurate statement in our reply: OOD results in the appendix cover DS-4 and DS-6a only, not all datasets.

---

> > ### Comment · Reviewer_sdt9 · 2026-05-29
> > **Comments**
> >
> > I appreciate that the authors engaged in a discussion and responded to my concerns. I also thank the authors for clarifying several points related to the choice of the parameters, and the evolution in time of the solutions to the PDEs considered. However, I want to clarify what my main concern is. The stated objective in the revision is still, to my understanding, to learn complex field dynamics in the small data regime. This goal is independent of whether other architectures perform better or not, and can be evaluated directly from the numerical simulations of the PDEs used in the experiments.
> >
> > I insist that it is not clear whether 100 simulations of the proposed PDEs constitutes a small data regime or not. My comment about solving the first two equations in closed-form using Fourier expansions is that the solutions to the diffusion and transport equations are given explicitly by a convolution, and, when restricted over a superposition of Gaussians of fixed scale as initial conditions, the Green's function can be approximated by a few Fourier coefficients. If the Green's function can be approximated with, e.g., 50 coefficients, then 100 simulations would not constitute a small data regime. This also holds after downsampling depending on how this is performed.
> >
> > For the first two equations, this observation can be verified mathematically with standard methods. My comments about the parameter choice for the other equations follow the same lines. Does the choice of simulation parameters and initial conditions constitute a small data regime for the observed dynamics? This is what I believe is not properly justified. A simple baseline could help in this regard, e.g., what is the result of using DMD on the 100 snapshots?
> >
> > The authors suggest that it is valuable to understand when some architectures fail while others succeed, e.g., the effect of inductive bias. I agree with this statement, but this is not clearly stated as an objective of this work. Even then, there are still some issues with the proposed experiments. Selecting periodic boundary conditions in a 2D domain may favor convolutional architectures. If the solutions to the equations are given explicitly, or *approximately*, by a convolution, then it is not entirely surprising that a convolutional architecture would have a better performance than others that do not explicitly exploit this structure. Would it be the case that the proposed architecture keeps performing well with, e.g., nonhomogeneous Dirichlet or Neumann boundary conditions?
> >
> > The revised manuscript adds some information missing from the original submission. The most notable addition is the ablation experiment to justify the use of a perceptual term. The authors state that "*the perceptual term measurably preserves the shape of extended structures*" in one of the experiments, but what is not clear is whether this effect is particular to the experimental setup. What would happen in more complex systems, e.g., fracture models, where the true geometric structures may or may not be penalized by this regularization?
> >
> > I believe that this work has substantial value, but I tend to agree with reviewer W5p8 that the experiments must be set up differently to support the claims made in the paper.

---

### Decision · Action_Editor_25bR · 2026-06-15

**Recommendation:** Reject

**Additional Comments:**

We believe that a more rigorous characterization of how varying architectural inductive biases influence learning performance in the small-data regime is necessary to substantiate the claims regarding the proposed method’s effectiveness.

**Audience:**

Yes

**Audience Explanation:**

This paper will be of interest to researchers developing efficient numerical methods for PDE simulation and those working in scientific machine learning.

**Claims And Evidence:**

No

**Claims Explanation:**

As Reviewer sdt9 and Reviewer W5p8 highlighted, while the research topic is highly relevant, the submission fails to clearly delineate the unique contribution of the proposed architecture for learning 2D PDE dynamics in small-data regimes. Furthermore, the reliance on purely empirical results, without sufficient theoretical justification or a rigorous mathematical formulation of the surrogate learning task, limits the strength of the claims and fails to address modern standards for resolution-invariant or function-space-aware modeling.